JCB Journal of Cell Biology

# Assembly and phosphoregulatory mechanisms of the budding yeast outer kinetochore KMN complex

Noah N. Turner[1], Ziguo Zhang[1], Jing Yang[1], Kyle W. Muir[1], Stephen H. McLaughlin[1], Tomos Morgan[1], and David Barford[1]

During mitosis and meiosis, kinetochores mediate interactions between chromosomes and spindle microtubules. Kinetochores are multi-megadalton protein complexes essential for chromosome segregation; however, recent structural, functional, and evolutionary studies have revealed divergent mechanisms of kinetochore assembly. Here, we use cryo-EM to understand the structural mechanisms by which the budding yeast microtubule-binding outer kinetochore KMN complex assembles, and how its interactions with the centromere-binding inner kinetochore are regulated. The KMN complex comprises three subcomplexes: Knl1c, Mis12c$^{Mtw1c}$, and Ndc80c. We show how C-terminal motifs of the Mis12c$^{Mtw1c}$ subunits Dsn1, Mis12$^{Mtw1}$, and Nnf1 bind Knl1c and Ndc80c. At the opposite end of the Mis12c$^{Mtw1c}$ stalk, an N-terminal auto-inhibitory segment of Dsn1 (Dsn1$^{AI}$) folds into two α-helices that engage the Mis12c$^{Mtw1c}$ head 1 domain, thereby occluding binding sites for the inner kinetochore subunits CENP-C$^{Mif2}$ and CENP-U$^{Ame1}$, reducing their affinity for Mis12c$^{Mtw1}$. Our structure reveals how Aurora B$^{Ipl1}$ phosphorylation of Dsn1$^{AI}$ would release this auto-inhibition to substantially strengthen preexisting connections between the inner and outer kinetochore.

## Introduction

Kinetochores are large nucleoprotein complexes that mediate faithful chromosome segregation during cell division (Ariyoshi and Fukagawa, 2023; McAinsh and Marston, 2022; Musacchio and Desai, 2017). Kinetochores assemble at centromeres to create an essential load-bearing linkage between chromosomes and microtubules of the mitotic spindle (Akiyoshi et al., 2010; Lang et al., 2018). The point centromeres of budding yeast *Saccharomyces cerevisiae* comprise genetically defined ~118-bp *CEN* sequences with three centromere DNA elements (CDEI to III) (Clarke and Carbon, 1980; Clarke and Carbon, 1983; Fitzgerald-Hayes et al., 1982; Hyman and Sorger, 1995). The *CEN* sequence and CENP-A$^{Cse4}$ nucleosome provide a foundation upon which the rest of the kinetochore assembles (Lang et al., 2018). The CENP-C$^{Mif2}$ protein mediates specific interactions between the CENP-A nucleosome and the inner kinetochore constitutive centromere–associated network (CCAN) complex, and also interacts with the outer kinetochore KMN complex (Ariyoshi et al., 2021; Dimitrova et al., 2016; Hornung et al., 2014; Kato et al., 2013; Przewloka et al., 2011; Screpanti et al., 2011; Xiao et al., 2017; Yan et al., 2019). CCAN complexes entrap centromeric DNA within topological chambers (Dendooven et al., 2023; Yatskevich et al., 2022).

The KMN complex bridges the inner kinetochore with the microtubule (Cheeseman et al., 2006; Deluca et al., 2006). Mis12c/Mis12c$^{Mtw1c}$ (human/budding yeast) interacts with Knl1c and Ndc80c to assemble the KMN complex (Fig. 1 A and Fig. S1 A) and mediates recruitment of the entire complex to kinetochores (Dimitrova et al., 2016; Ghodgaonkar-Steger et al., 2020; Petrovic et al., 2014; Petrovic et al., 2010; Polley et al., 2024; Yatskevich et al., 2024). Ndc80c also associates with CENP-T independently of Mis12c$^{Mtw1c}$ and Knl1c (Malvezzi et al., 2013; Nishino et al., 2013). Mis12c engages Knl1c and Ndc80c at the apex of a central α-helical stalk (Petrovic et al., 2014). The base of the stalk bifurcates to form the head 1 and head 2 domains that mediate interactions with the inner kinetochore (Akiyoshi et al., 2013; Dimitrova et al., 2016; Emanuele et al., 2008; Kim and Yu, 2015; Petrovic et al., 2016; Rago et al., 2015; Walstein et al., 2021). Head 1 interacts with N-terminal motifs of CENP-C$^{Mif2}$ and CENP-U$^{Ame1}$ in budding yeast, and CENP-C$^{Mif2}$ and CENP-T in vertebrates (Dimitrova et al., 2016; Hornung et al., 2014; Huis In 'T Veld et al., 2016; Killinger et al., 2020; Petrovic et al., 2016). The mutually exclusive binding of inner kinetochore subunits to Mis12c$^{Mtw1c}$ (Huis In 'T Veld et al., 2016; Killinger et al., 2020) recruits multiple copies of KMN to the inner kinetochore

[1]MRC Laboratory of Molecular Biology, Cambridge, UK.

Correspondence to David Barford: dbarford@mrc-lmb.cam.ac.uk; Noah N. Turner: noah.turner@jic.ac.uk

N.N. Turner current affiliation is John Innes Centre, Norwich, UK. K.W. Muir current affiliation is Institute of Cell Biology, University of Edinburgh, Edinburgh, UK.

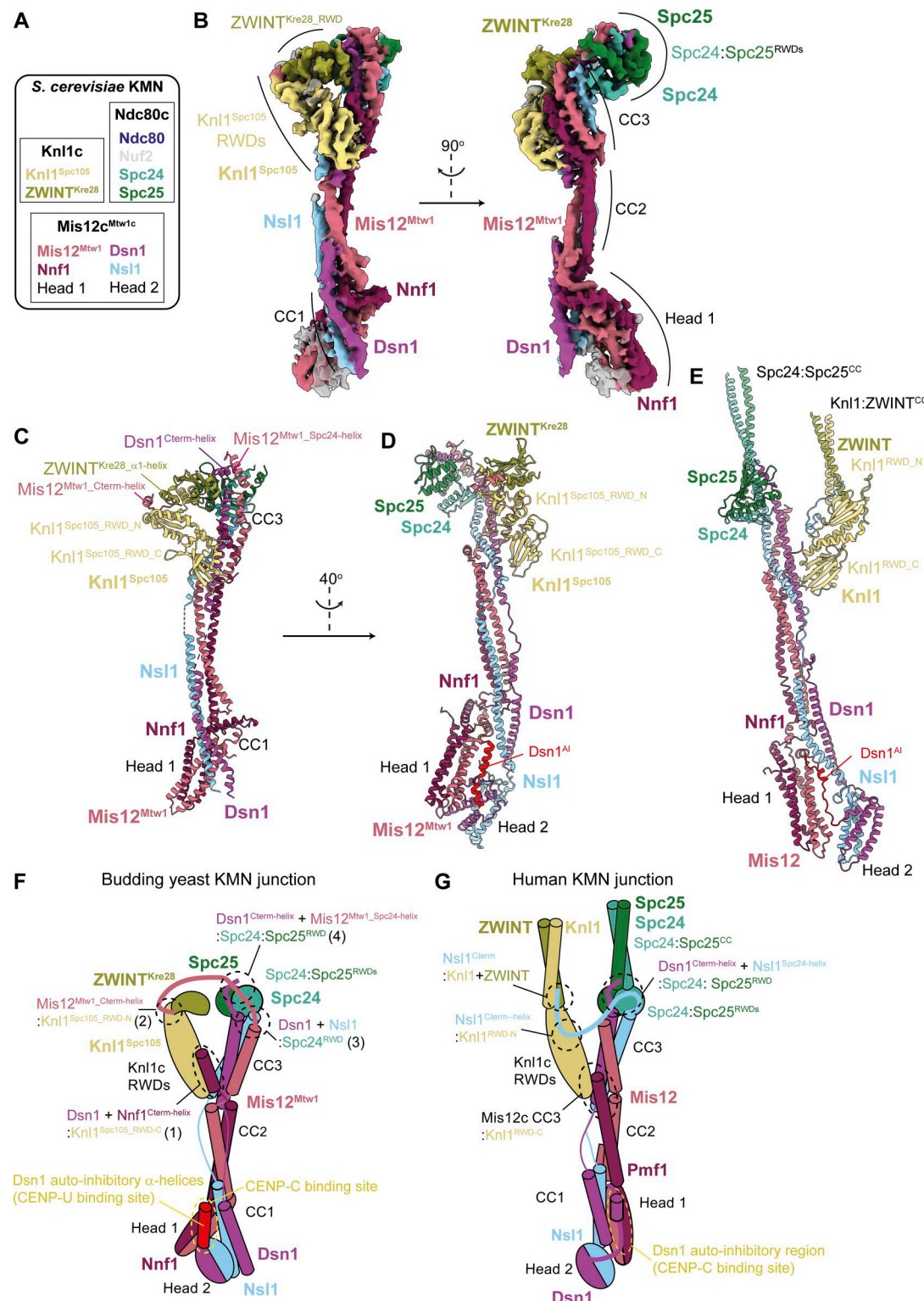

Figure 1. **Overall organization of the *S. cerevisiae* KMN junction complex. (A)** Subcomplex and subunit composition of the *S. cerevisiae* KMN complex. **(B)** Composite cryo-EM density map of the *S. cerevisiae* KMN junction complex derived from docking the apex and base bodies (Fig. S2 Diii) into the consensus density map. **(C)** Model of the KMN junction complex, highlighting the motifs that mediate the interactions between Mis12c[Mtw1c] and Knl1c or Ndc80c. **(D and E)** Comparison of cryo-EM structures of the inactive *S. cerevisiae* KMN junction complex with head 2 and Dsn1[AI] defined (this study) (D) with the inactive human KMN junction complex (PDB: 8PPR) from Yatskevich et al. (2024) (E). Structures shown in D and E were superimposed on the stalk segment (CC2, CC3) of the Mis12/Mis12[Mtw1] chain. **(F)** Cartoon schematic of the *S. cerevisiae* KMN junction complex, based on the structures determined in this study. The interactions between Mis12[Mtw1_Cterm-helix] and Knl1[Spc105_RWD-N]; Nnf1[Cterm-helix] and Knl1[Spc105_RWD-C]; and Dsn1[Cterm-helix] and Mtw1[Spc24-helix] and the Spc24 and 25 RWD domains, and Mis12c[Mtw1c] and the two auto-inhibitory α-helices in Dsn1[AI], and CENP-C[Mif2] and CENP-Q[Ame1]–binding sites are highlighted. The four interfaces discussed in the text are numbered in parenthesis. **(G)** Cartoon schematic of the human KMN junction complex, based on the structures determined in Yatskevich et al. (2024) (PDB: 8PPR). Interactions between Nsl1[Cterm] and Knl1c; Mis12c stalk CC3 region and Knl1[RWD-C]; and Dsn1[Cterm-helix] and Nsl1[Spc24-helix] and the Spc24 and 25 RWD domains, and Mis12c head 1 and Dsn1[AI], and CENP-C–binding site are highlighted.

(Cieslinski et al., 2023; Dhatchinamoorthy et al., 2017; Joglekar et al., 2006; Johnston et al., 2010). These interactions are auto-inhibited by a region within the intrinsically disordered N terminus of the Mis12c$^{Mtw1c}$ subunit Dsn1 in a manner that is relieved by Aurora B$^{Ipl1}$ phosphorylation at mitotic kinetochores (Dimitrova et al., 2016; Petrovic et al., 2016; Polley et al., 2024; Takenoshita et al., 2024; Walstein et al., 2021; Yatskevich et al., 2024).

Ndc80c interacts with microtubules through N-terminal domains of its constituent Ndc80 and Nuf2 subunits (Alushin et al., 2012; Alushin et al., 2010; Ciferri et al., 2008; Lampert et al., 2013; Muir et al., 2023; Wei et al., 2007), augmented through interactions with the unrelated Astrin/SKAP and Ska complexes in human and the Dam1 complex in budding yeast (Helgeson et al., 2018; Huis In 'T Veld et al., 2019; Kern et al., 2017; Lampert et al., 2010; Muir et al., 2023; Tien et al., 2010; Van Hooff et al., 2017). The Ndc80 and Nuf2 microtubule-binding calponin homology (CH) domains are at the opposite end of the 570 Å-long Ndc80 complex to RWD domains in the C termini of the Spc24 and Spc25 subunits (Spc24$^{RWD}$:Spc25$^{RWD}$) that interact with Mis12c$^{Mtw1c}$ (Wang et al., 2008; Wei et al., 2006; Wei et al., 2005). The long Ndc80c is formed by the end-on interactions of the Ndc80:Nuf2 and Spc24:Spc25 coiled coils (Ciferri et al., 2005; Valverde et al., 2016). Kinetochore-associated Knl1c contributes to spindle assembly checkpoint (SAC) and error correction signaling (reviewed in Fischer, 2023; McAinsh and Kops, 2023).

Although the general mechanisms of KMN complex assembly and regulation are conserved from budding yeast to human, specific details of how Mis12c$^{Mtw1c}$ interacts with both Ndc80c and Knl1c differ (Fig. 1, F and G). The Spc24:Spc25 RWD domains of Ndc80c interact with α-helical motifs in C-terminal extensions of the rodlike stalk domain of human Mis12c and budding yeast Mis12c$^{Mtw1c}$. These include a conserved C-terminal α-helix in the Dsn1 subunit (Dimitrova et al., 2016; Polley et al., 2024; Yatskevich et al., 2024), and, additionally in humans, an α-helix in the Mis12c subunit Nsl1 (Polley et al., 2024; Yatskevich et al., 2024). No equivalent motif exists in budding yeast Nsl1 (Dimitrova et al., 2016). Instead, an α-helix in the budding yeast Mis12$^{Mtw1}$ binds to the Spc24 and Spc25 RWD domains cooperatively with the Dsn1 α-helix (Ghodgaonkar-Steger et al., 2020). In humans, motifs at the C terminus of Nsl1 mediate binding of Mis12c to the tandem RWD domains of the human Knl1c subunit Knl1 (Kiyomitsu et al., 2007; Petrovic et al., 2016; Petrovic et al., 2014; Polley et al., 2024; Yatskevich et al., 2024). The orthologous budding yeast protein Knl1$^{Spc105}$ also uses tandem RWD domains to interact with Mis12c$^{Mtw1c}$ (Maskell et al., 2010), although the regions of Mis12c$^{Mtw1c}$ that interact with yeast Knl1c are largely uncharacterized. Likewise, due to sequence differences between human and budding yeast Dsn1, the structural basis of Mis12c$^{Mtw1c}$ auto-inhibition by an N-terminal region of Dsn1, and its relief by Aurora B$^{Ipl1}$ phosphorylation (Akiyoshi et al., 2013; Kim and Yu, 2015; Yang et al., 2008) are unclear. To understand these differences, we determined cryo-EM structures of the *S. cerevisiae* KMN complex. The structure reveals that interactions between the three KMN subcomplexes are organized by α-helical motifs in the C termini

of the Mis12c$^{Mtw1c}$ subunits Dsn1, Mis12$^{Mtw1}$, and Nnf1, which engage interfaces on Knl1c and Ndc80c. Our work shows that a combination of conserved and plastic interfaces, divergent at the sequence and subunit level, underpins a comparable KMN complex architecture between point and regional centromeres. Furthermore, we show that Aurora B$^{Ipl1}$ overcomes auto-inhibition of human Mis12c and budding yeast Mis12c$^{Mtw1c}$ through similar structural mechanisms.

## Results

### Structural features of the budding yeast KMN complex

To determine the cryo-EM structure of the *S. cerevisiae* KMN complex, we purified Ndc80c and a Knl1c:Mis12c$^{Mtw1c}$ complex (K$^{HB-RWD}$M) from insect cells (Fig. S1, A and B). K$^{HB-RWD}$M comprised a truncated Knl1$^{Spc105}$ subunit (Knl1$^{Spc105\_HB-RWD}$) that incorporated its helical bundle and tandem RWD domains but had a deleted N-terminal disordered region (retaining residues 445–917). Protein band intensities on SDS-PAGE gels (Fig. S1 B) indicated that the purified K$^{HB-RWD}$M contained one subunit of each of Knl1$^{Spc105\_HB-RWD}$ and ZWINT$^{Kre28}$, a stoichiometry consistent with the number of Knl1$^{Spc105}$ and ZWINT$^{Kre28}$ proteins at kinetochores in cells defined using fluorescence microscopy (Roy et al., 2022).

We pursued numerous strategies to reconstitute and stabilize KMN on cryo-EM grids, with the best results obtained using the GraFix methodology (Kastner et al., 2008). The KMN complex (Fig. 1 A) was reconstituted by incubating K$^{HB-RWD}$M and Ndc80c (Fig. S1, C and D) at an equimolar ratio. K$^{HB-RWD}$MN complex formation was indicated by the comigration in a glycerol gradient of its 10 subunits, separate from K$^{HB-RWD}$M and Ndc80c (Fig. S1 E). The reconstituted K$^{HB-RWD}$MN (henceforth KMN) was crosslinked and purified using a 10–30% glycerol gradient with 0.0–0.2% glutaraldehyde (Fig. S1 F). We collected a large cryo-EM dataset on this sample. Cryo-electron micrographs and 2D class averages revealed elongated particles (Fig. S2, A and B) that corresponded closely with the previously observed structures of isolated Mis12c$^{Mtw1c}$, Mis12c, and Ndc80c, the entire human KMN complex, and negative-stain EM of the budding yeast KMN complex (Ciferri et al., 2008; Dudziak et al., 2025; Hornung et al., 2011; Maskell et al., 2010; Petrovic et al., 2014; Petrovic et al., 2010; Polley et al., 2024; Wang et al., 2008; Wei et al., 2005; Yatskevich et al., 2024). We determined medium-resolution cryo-EM maps of a KMN junction complex that contained the stalk and head 1 domains of Mis12c$^{Mtw1c}$, and the RWD domains of Knl1c and Ndc80c (Fig. S2, C–F and Table S1).

Overall, the *S. cerevisiae* KMN junction complex architecture (Fig. 1, B and C; and Video 1) (schematic in Fig. 1 F) resembles the human KMN complex (Polley et al., 2024; Yatskevich et al., 2024) (Fig. 1, E and G). The Mis12c$^{Mtw1c}$ structure is dominated by an α-helical bundle assembled from its four subunits that form a central stalk. This stalk domain comprises three coiled-coil regions (CC1, CC2, and CC3) (Dimitrova et al., 2016). Similar to human KMN, the Knl1c and Spc24:Spc25 RWD domains are bound at the apex of the Mis12c$^{Mtw1c}$ stalk, and, although in close proximity, are not in direct contact. The Mis12c$^{Mtw1c}$ head 1 domain, comprising a short four α-helical bundle formed from the

Mis12$^{Mtw1}$ and Nnf1 subunits, is attached to the base of the stalk through loops, docking against Dsn1 and Nsl1 of the CC1 region (Fig. 1, B and C). In the composite cryo-EM map of the *S. cerevisiae* KMN junction complex, no density was observed for portions of Ndc80c comprising the Spc24:Spc25 coiled coils and the entire Ndc80:Nuf2 subcomplex. Likewise, for Knl1c, all regions preceding the C-terminal RWD domains of Knl1$^{Spc105}$ and ZWINT$^{Kre28}$ were not visible. Disorder of both the Ndc80c and Knl1c subcomplexes in *S. cerevisiae* KMN, indicative of their conformational flexibility, was also observed in the human KMN structure (Polley et al., 2024; Yatskevich et al., 2024).

Our structure of *S. cerevisiae* KMN also revealed important differences with the human KMN complex, specifically in how Mis12c$^{Mtw1c}$ interacts with both Knl1c and Ndc80c, and a C-terminal RWD domain in ZWINT$^{Kre28}$ (ZWINT$^{Kre28\_RWD}$) that is absent from the human ortholog (Fig. 1, C–G). An RWD domain in *S. cerevisiae* ZWINT$^{Kre28}$ had previously been predicted through phylogenetic analyses and AlphaFold2 (AF2) models (Polley et al., 2024; Tromer et al., 2019). In *S. cerevisiae* KMN, ZWINT$^{Kre28\_RWD}$ interacts with the N-terminal RWD domain of Knl1$^{Spc105}$, a role played by the ZWINT C-terminal α-helix in human Knl1 (Yatskevich et al., 2024). Lastly, in this reconstruction of the *S. cerevisiae* KMN junction complex, cryo-EM density for the Mis12c$^{Mtw1c}$ head 2 domain is not visible due to the subtraction of head 2 cryo-EM density for cryo-EM data processing.

### Interactions of Mis12c$^{Mtw1c}$ with Knl1c and Ndc80c and comparison with human KMN

#### Mis12c$^{Mtw1c}$ contacts Knl1c through the Dsn1:Nnf1 CC3 and a peptidic extension of Mis12$^{Mtw1}$

The Dsn1, Mis12$^{Mtw1}$, and Nnf1 subunits mediate Mis12c$^{Mtw1c}$ interactions at two contact sites on Knl1c. First, conserved aliphatic residues in Dsn1 and the C-terminal α-helix of Nnf1 (Nnf1$^{Cterm-helix}$) (Fig. 2 C), part of Mis12c$^{Mtw1c}$ CC3, engage a hydrophobic patch on the C-terminal RWD domain of Knl1$^{Spc105}$ (Knl1$^{Spc105\_RWD-C}$) (interface [1]) (Fig. 1, C and F; Fig. 2, A–C; Fig. 3 A; and Video 1). In general, this *S. cerevisiae* Mis12c$^{Mtw1c}$-Knl1c interface resembles its counterpart in human KMN (Fig. 2, E and F) (Polley et al., 2024; Yatskevich et al., 2024).

At the second Mis12c$^{Mtw1c}$-Knl1c contact site, which differs from human KMN (Polley et al., 2024; Yatskevich et al., 2024), the extreme C-terminal α-helix of Mis12$^{Mtw1}$ (residues 282–289: Mis12$^{Mtw1\_Cterm-helix}$) engages a channel formed at the interface of the Knl1$^{Spc105}$ RWD-N domain (Knl1$^{Spc105\_RWD-N}$) and an α-helix of ZWINT$^{Kre28}$ (ZWINT$^{Kre28\_α1-helix}$) (interface [2]) (Fig. 1, C and F; Fig. 2, A, B, and D–F; and Fig. 3, B and C). The ZWINT$^{Kre28\_α1-helix}$ combines with Knl1$^{Spc105\_RWD-N}$ to create a hydrophobic channel that is ideally suited to bind the mixed hydrophobic and acidic Mis12$^{Mtw1\_Cterm-helix}$ (Fig. 2 D). Immediately preceding Mis12$^{Mtw1\_Cterm-helix}$, a β-strand from Mis12$^{Mtw1}$ augments the β-sheet of Knl1$^{Spc105\_RWD-N}$ (Fig. 2 B). Consistent with the observation that the ZWINT$^{Kre28\_RWD}$ domain is not essential for *S. cerevisiae* viability (Dudziak et al., 2025), our structure shows that although ZWINT$^{Kre28\_RWD}$ may indirectly stabilize ZWINT$^{Kre28\_α1-helix}$ through its contact with Knl1$^{Spc105}$ (Fig. 2 B), ZWINT$^{Kre28\_RWD}$ does not contribute

directly to contacting Mis12c$^{Mtw1c}$ subunits. In human KMN, ZWINT lacks an RWD domain, and Mis12 does not contact Knl1c. Instead, the C terminus of Nsl1 contacts the Knl1$^{RWD-N}$:ZWINT interface and a shallow groove on Knl1$^{RWD-N}$ (Petrovic et al., 2014; Polley et al., 2024; Yatskevich et al., 2024) (Fig. 1 G and Fig. 2 F). Our experimental structure of the Mis12$^{Mtw1}$-Knl1c interface agrees with AF2 predictions of the Knl1c:Mis12c$^{Mtw1c}$ complex (Fig. 3, C–E), supporting our assignment of cryo-EM densities to Mis12$^{Mtw1\_Cterm-helix}$ and Nnf1$^{Cterm-helix}$ (Fig. 3, A and B).

Consistent with our structure, deletion of residues comprising and surrounding Mis12$^{Mtw1\_Cterm-helix}$ mildly reduced levels of Knl1$^{Spc105}$ and ZWINT$^{Kre28}$ that copurified with Mis12c$^{Mtw1c}$ isolated from yeast (Ghodgaonkar-Steger et al., 2020). Indeed, we found that deletion of either Mis12$^{Mtw1\_Cterm-helix}$ or Nnf1$^{Cterm-helix}$ reduced the copurification of Mis12c$^{Mtw1c}$ with Knl1c from insect cells (Fig. 3 F). These deletions did not affect the stability nor assembly of Mis12c$^{Mtw1c}$ (Fig. S3, A and B) nor its interaction with Ndc80c (Fig. S3, C and D), suggesting that the specific functions of Mis12$^{Mtw1\_Cterm-helix}$ and Nnf1$^{Cterm-helix}$ are to mediate the interaction between Mis12c$^{Mtw1c}$ and Knl1c.

To investigate the function of Mis12$^{Mtw1\_Cterm-helix}$ and Nnf1$^{Cterm-helix}$ *in vivo*, we used the auxin-inducible degron (AID) system (Tanaka et al., 2015) to deplete the endogenous *MTW1* and *NNF1* gene products and tested whether mutant *mtw1* and *nnf1* proteins with their C-terminal α-helices deleted (*mtw1*$^{ΔC}$ and *nnf1*$^{ΔC}$, respectively) rescued depletion of the endogenous gene product. Western blotting confirmed that the mAID$_3$-FLAG$_5$–tagged gene products of the endogenous MTW1 and NNF1 loci were depleted after addition of indole-3-acetic acid (IAA) (Fig. S1, G and H), and that the rescue alleles were expressed from the exogenous loci (Fig. S1, I and J).

Consistent with the essentiality of Mis12$^{Mtw1}$ and Nnf1 (Euskirchen, 2002), strains with an empty cassette incorporated at the exogenous locus were unable to grow on plates containing IAA (Fig. S3 E). The expression of wild-type *MTW1* and *NNF1* rescue alleles rescued this lethality (Fig. S3 E). The mutant *mtw1*$^{ΔC}$ and *nnf1*$^{ΔC}$ alleles also rescued lethality caused by depletion of endogenous *MTW1* and *NNF1* at 30°C (Fig. S3 E). These data are consistent with our *in vitro* results that deletion of either Mis12$^{Mtw1\_Cterm-helix}$ or Nnf1$^{Cterm-helix}$ helices alone reduces, but does not abolish, copurification of Mis12c$^{Mtw1c}$ with Knl1c (Fig. 3 F). We were unable to test the consequence to cell growth of deleting Mis12$^{Mtw1\_Cterm-helix}$ and Nnf1$^{Cterm-helix}$ simultaneously because strains combining both *mtw1*$^{ΔC}$ and *nnf1*$^{ΔC}$ alleles with AID-tagged *MTW1* and *NNF1* could not be isolated, suggestive of lethality.

#### Mis12c$^{Mtw1c}$ contacts Ndc80c through the Dsn1:Nsl1 CC3 and peptidic extensions of Dsn1 and Mis12$^{Mtw1}$

In the *S. cerevisiae* KMN junction complex, Mis12c$^{Mtw1c}$ engages Ndc80c through multiple contacts involving the RWD domains of Spc24 and Spc25 (Spc24$^{RWD}$ and Spc25$^{RWD}$). These interactions comprise the CC3 region of the Mis12c$^{Mtw1c}$ stalk, and peptidic C-terminal extensions of Dsn1 and Mis12$^{Mtw1}$ (Fig. 1, C and F; Fig. 4, A–E; Fig. 5, A and B; and Video 1). First, as similarly observed in human KMN (Polley et al., 2024; Yatskevich et al.,

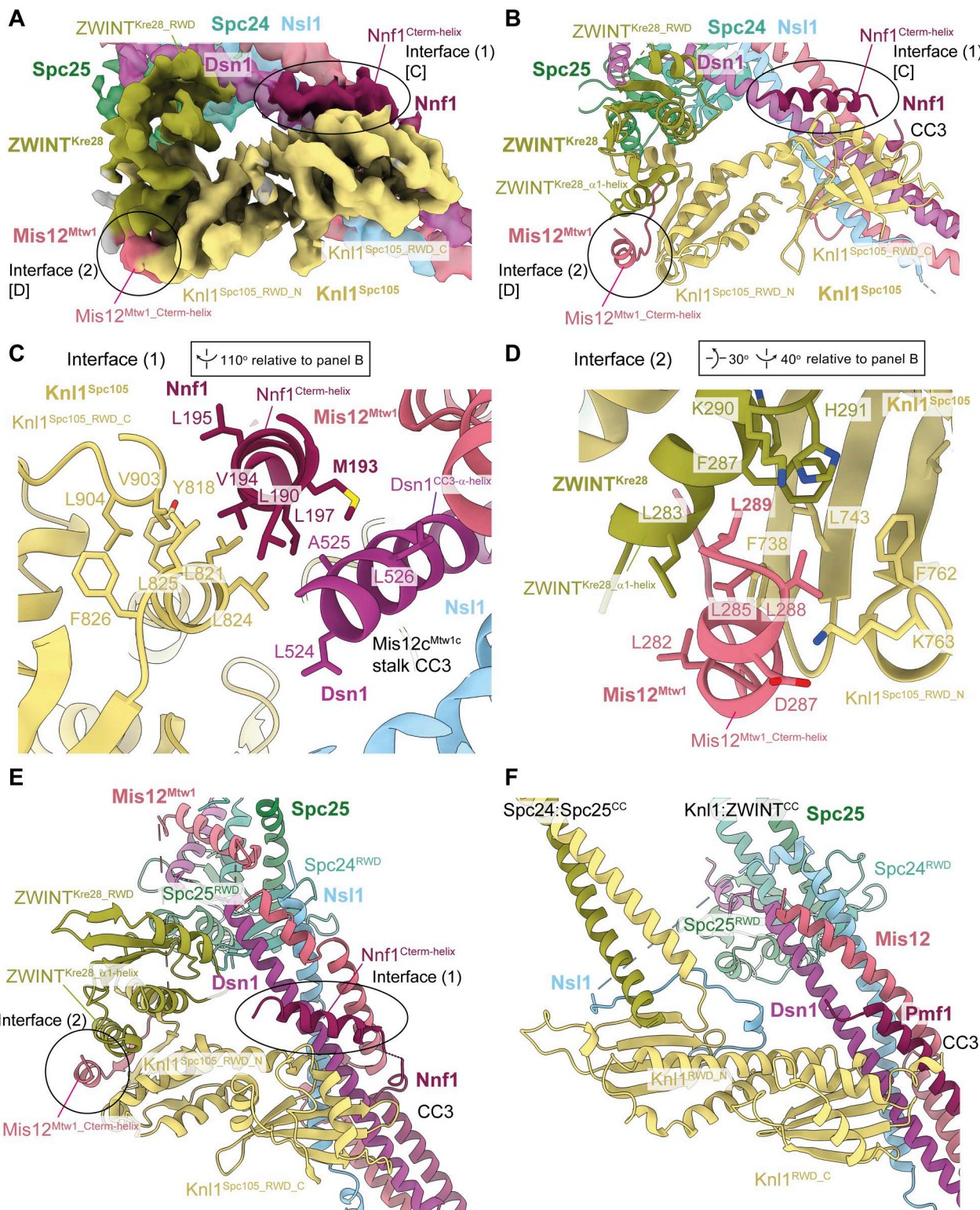

Figure 2. **Structural details of the interactions between Mis12c^Mtw1c and Knl1c. (A)** Cryo-EM density map (Fig. S2 Diii) highlighting the interactions between Mis12c^Mtw1c and Knl1c. [C] and [D] highlight regions of the map that are shown close-up in C and D, for interface 1 and interface 2, respectively. **(B)** Model of Knl1c engaging in interactions with motifs from Mis12c^Mtw1c, highlighting the Mis12^Mtw1_Cterm-helix and Nnf1^Cterm-helix motifs from Mis12c^Mtw1c that interact with the Knl1^Spc105_ RWD-N and Knl1^Spc105_RWD-C domains, respectively. The molecule is rotated 90° anti-clockwise related to Fig. 1 C. **(C)** Structural details of the interaction between the Nnf1^Cterm-helix helix, the hydrophobic patch on the Knl1^Spc105_RWD-C domain, and the Dsn1^CC3_α-helix at interface (1). **(D)** Structural details of the interaction between the Mis12^Mtw1_Cterm-helix and the interface formed by the ZWINT^Kre28_RWD and Knl1^Spc105_ RWD-N domains at interface (2). **(E and F)** Comparison of *S. cerevisiae* (E) and human (F) KMN junction complexes at the Mis12c interface with Knl1:Zwint. Human structure PDB: 8PPR from Yatskevich et al. (2024). Structures were superimposed on the Knl1^RWD domains. The comparison highlights the interchangeable roles played by the C termini of Mis12^Mtw1 in *S. cerevisiae* and Nsl1 in human in contacting the Knl1^RWD-C domain. Views in E and F are similar to (B). Source data are available for this figure: SourceData F2.

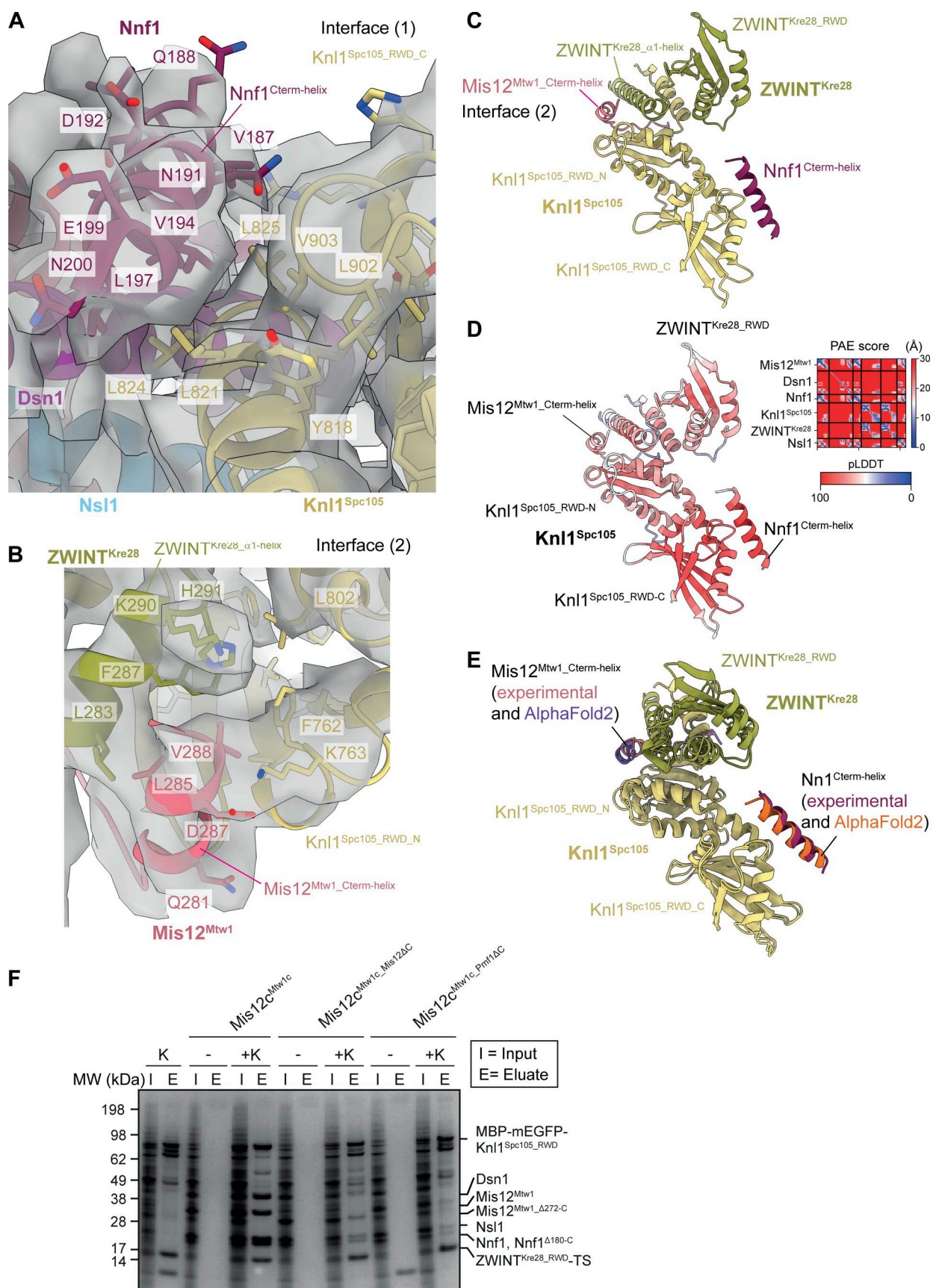

Figure 3. **Modeling of interactions between Mis12cMtw1c and Knl1c. (A)** Molecular model of the interaction between Nnf1Cterm-helix and Knl1c docked into the cryo-EM density map, showing density for amino acid side chains relevant to the interaction. View is similar to Fig. 3 C. **(B)** Molecular model of the

interaction between Mis12$^{Mtw1\_Cterm-helix}$ at interface (2) and Knl1c docked into the cryo-EM density map, showing density for amino acid side chains relevant to the interaction. View is similar to Fig. 3 D. **(C)** AF2 structure prediction of the Knl1c:Mis12c$^{Mtw1c}$ interaction. The prediction was performed using full-length protein sequences apart from Knl1$^{Spc105}$, for which residues 1–444 were excluded. The structure prediction is colored by chain. **(D)** AF2 structure prediction presented in C with the model colored by residue pLDDT score and with the PAE score shown as an inset in the top right corner of the panel. **(E)** Structural alignment of the Knl1c:Mis12c$^{Mtw1c}$ AF2 model with the Mis12c$^{Mtw1c}$ and Knl1c experimental molecular model generated in this study. The predicted and experimentally modeled Knl1$^{Spc105}$ and ZWINT$^{Kre28}$ subunits are colored identically, whereas the Nnf1$^{Cterm-helix}$ and Mis12$^{Mtw1\_Cterm-helix}$ are colored distinctly according to whether they derive from the AF2 model or the experimental molecular model. **(F)** Pull-down experiment to investigate the role of the Mis12$^{Mtw1\_Cterm-helix}$ and Nnf1$^{Cterm-helix}$ helical motifs in mediating the interaction between Knl1c and Mis12c$^{Mtw1c}$. Knl1c composed of Knl1$^{Spc105\_RWD}$ and ZWINT$^{Kre28\_RWD}$ containing a C-terminal TS (K$^{RWD}$) was coexpressed with Mis12c$^{Mtw1c}$ in insect cells. I = input; E = eluate; K = K$^{RWD}$ complex present; - = K$^{RWD}$ complex absent; Mis12$^{Mtw1\_\Delta C}$ = Mis12$^{Mtw1\_\Delta272-C}$; Nnf1$^{\Delta C}$ = Nnf1$^{\Delta180-C}$. Experimental details are provided in Materials and methods. **(E and F)** Structures superimposed on Knl1$^{RWD}$ domains. PAE, predicted alignment error.

2024), the hydrophobic patch of the Spc24$^{RWD}$ β-sheet interacts with hydrophobic residues of Dsn1 and Nsl1 in CC3 (interface 3) (Fig. 1, C and F; Fig. 4, B, C, E, and F; and Fig. 5 A). Second, differing from human KMN, an α-helical segment of Mis12$^{Mtw1}$ (Mis12$^{Mtw1\_Spc24-helix}$) buttresses Spc24$^{RWD}$ and the Spc24:Spc25 coiled coil (interface 4) (Fig. 1, C and F; Fig. 4, B, D–F; and Fig. 5 B), before Mis12$^{Mtw1}$ crosses to contact Knl1c through Mis12$^{Mtw1\_Cterm-helix}$ (Fig. 1, C and F). The interaction of Mis12$^{Mtw1\_Spc24-helix}$ residues Tyr236 and Arg240 with Spc24$^{RWD}$ and Spc25$^{RWD}$ is supported by well-resolved cryo-EM density (Fig. 5 B). These interactions likely rationalize why Asp substitution of Mis12$^{Mtw1}$ residues Arg233 and Tyr236 reduced levels of Spc24:Spc25 associated with Mis12c$^{Mtw1c}$ in an in vitro assay, and sensitizes yeast to benomyl (Ghodgaonkar-Steger et al., 2020). In human KMN, S. cerevisiae Mis12$^{Mtw1\_Spc24-helix}$ is functionally replaced by a short α-helix of Nsl1 (Nsl1$^{Spc24-helix}$) that buttresses Spc24$^{RWD}$ and the Spc24:Spc25 coiled coil, before Nsl1 also crosses to contact Knl1c (Nsl1$^{Cterm}$) (Polley et al., 2024; Yatskevich et al., 2024) (Fig. 1, F and G; and Fig. 4, E and F).

Thirdly, in another interaction that is conserved with human KMN (Polley et al., 2024; Yatskevich et al., 2024) the C-terminal Dsn1 α-helix (residues 560–574: Dsn1$^{Cterm-helix}$) docks into a deep channel at the interface of Spc24$^{RWD}$ and Spc25$^{RWD}$ (interface 4: Fig. 1, C and F; Fig. 4, B, D–F; and Fig. 5 B). However, the budding yeast KMN complex differs from human in that the Mis12$^{Mtw1\_Spc24-helix}$ stabilizes this interaction. The four-turn Mis12$^{Mtw1\_Spc24-helix}$ bridges both Dsn1$^{Cterm-helix}$ and Spc24$^{RWD}$ through interactions with Val244 and Leu245 of Mis12$^{Mtw1\_Spc24-helix}$ (Fig. 4 D and Fig. 5 B). The shorter human Nsl1$^{Spc24-helix}$ does not extend to contact the Dsn1$^{Cterm-helix}$ (Polley et al., 2024; Yatskevich et al., 2024). This interaction of budding yeast Dsn1$^{Cterm-helix}$ with Mis12$^{Mtw1\_Spc24-helix}$ might be the basis by which Dsn1$^{Cterm-helix}$ and Mis12$^{Mtw1\_Spc24-helix}$ bind to Spc24:Spc25 cooperatively. This interface could also explain how Asp substitutions of Mis12$^{Mtw1}$ Val244, together with Leu248, a residue not resolved in our structure, strongly impaired binding of Spc24:Spc25 and Mis12c$^{Mtw1c}$ in vitro, and confers benomyl sensitivity in vivo (Ghodgaonkar-Steger et al., 2020).

Lastly, our structure of Mis12$^{Mtw1\_Spc24-helix}$ and Dsn1$^{Cterm-helix}$ bound to Spc24$^{RWD}$:Spc25$^{RWD}$ agrees with a crystal structure of the Spc24$^{RWD}$:Spc25$^{RWD}$:Dsn1$^{Cterm-helix}$ complex (Fig. 5 C) (Dimitrova et al., 2016), and an AF2 prediction of Mis12c$^{Mtw1c}$ with Spc24 and Spc25 (Fig. 5, D and E).

## Mechanism of Mis12c$^{Mtw1c}$ auto-inhibition and activation by Aurora B$^{Ipl1}$ kinase

### An auto-inhibitory segment of Dsn1 engages the head 1 domain of Mis12c$^{Mtw1c}$

The Dsn1 N-terminal intrinsically disordered region (N-IDR), situated immediately N-terminal of head 2, auto-inhibits the interaction of Mis12c$^{Mtw1c}$ with the inner kinetochore (Dimitrova et al., 2016; Petrovic et al., 2016; Walstein et al., 2021; Yatskevich et al., 2024), whereas Aurora B$^{Ipl1}$ phosphorylation of this region strengthens inner and outer kinetochore interactions (Akiyoshi et al., 2013; Kim and Yu, 2015; Yang et al., 2008). Due to signal subtraction of the Mis12c$^{Mtw1c}$ head 2 domain during cryo-EM processing, our model of the KMN junction complex did not reveal the structural basis of Mis12c$^{Mtw1c}$ auto-inhibition. However, we observed cryo-EM density on the surface of head 1 not accounted for in our model (Fig. S2 Dii). We reasoned this density might correspond to either a region of head 2 or a region of Dsn1$^{N-IDR}$ involved in auto-inhibition (Dimitrova et al., 2016). To investigate these possibilities, we performed focused 3D classification of the non–signal-subtracted KMN particles using a mask encompassing the base of the Mis12c$^{Mtw1c}$ stalk, head 1, and putative head 2 domain (Fig. S4 A). Using this approach, we obtained a subset of particles that resulted in 3D reconstructions with defined cryo-EM density for the Mis12c$^{Mtw1c}$ head 1 and head 2 domains (Fig. S4, A–D and Table S1).

The small number of particles (18,160) exhibiting head 2 associated with head 1 limited the global resolution of the reconstruction to 6.5 Å (Fig. S4 B), with a local resolution of 6.0 Å in regions of the map closest to the head 1 central region (Fig. S4 C). The overall conformations of the head 1 and Mis12c$^{Mtw1c}$ stalk domains are unchanged compared with the composite cryo-EM map of the KMN junction complex (Fig. S5, A and B). To interpret the putative head 2 density (Fig. S5A), we used an AF2 prediction of S. cerevisiae Spc24:Spc25:Mis12c$^{Mtw1c}$ (Fig. S5, C and D). This accounted for the helical bundle of head 2 docked onto head 1, but not an additional α-helical-like density contacting head 1, nor continuous density between CC1 of the Mis12c$^{Mtw1c}$ stalk and the head 2 domain (Fig. S5, A and B). Using crosslinking mass spectrometry (CL-MS) on the K$^{HB-RWD}$M complex, we identified crosslinks between residues in head 1 and head 2, in agreement with the docked configuration of head 2 (Fig. 6 A).

The AF2 model of S. cerevisiae Mis12c$^{Mtw1c}$ (ScMis12c$^{Mtw1c}$) was not consistent with the conformation of head 2 in the ScMis12c$^{Mtw1c}$ cryo-EM map with improved head 2 density (Fig. S5, E and F). In contrast, an AF2 prediction of Mis12c$^{Mtw1c}$ from

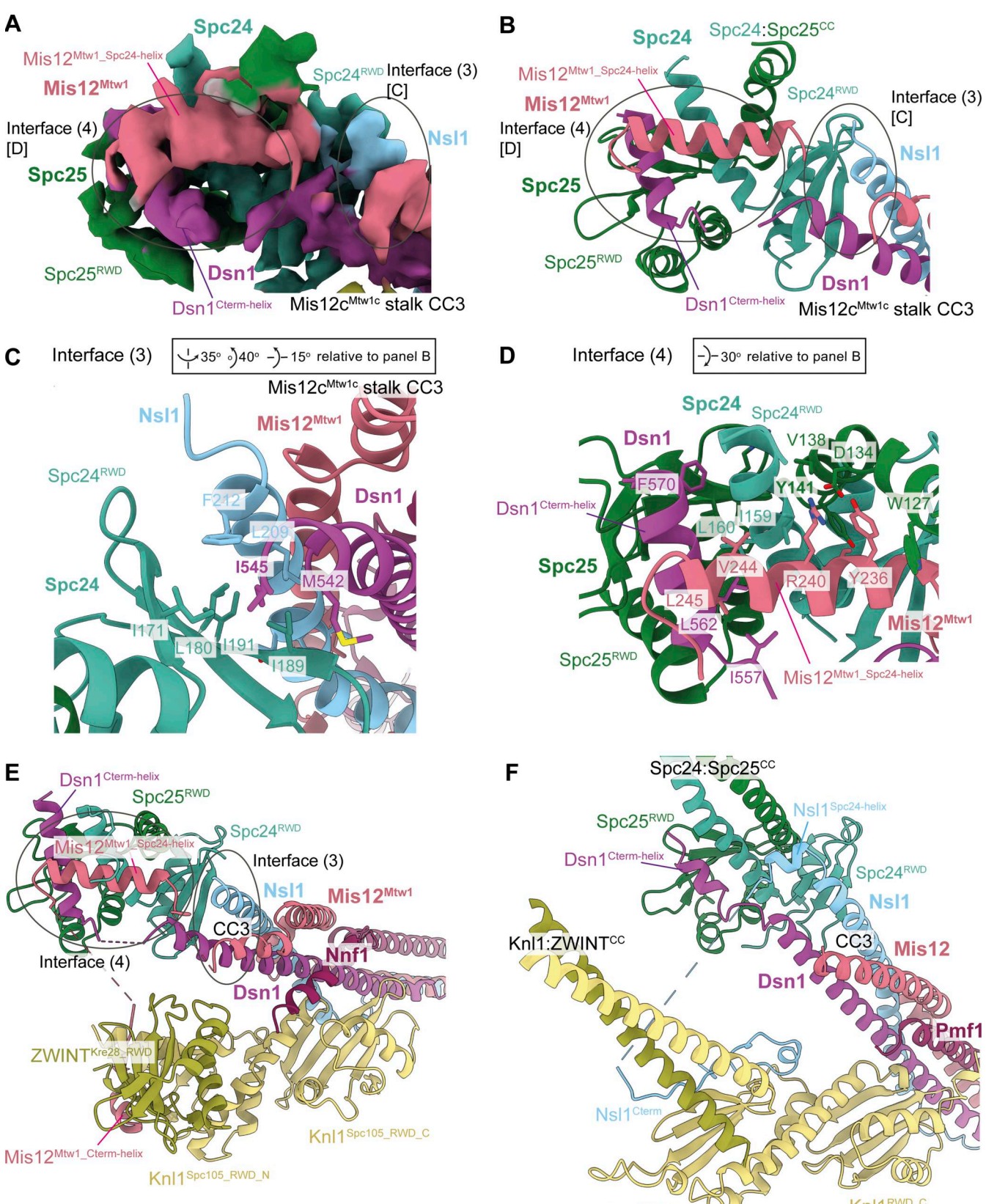

Figure 4. **Structural details of interactions between Ndc80c and Mis12c^Mtw1c. (A)** Cryo-EM density map (Fig. S2 Diii) highlighting the interactions between Mis12c^Mtw1c and Spc24 and Spc25 RWD domains. [C] and [D] highlight regions of the map that are shown close-up in C and D for interface 3 and interface 4, respectively. **(B)** Model of Spc24 and Spc25 RWD domains interacting with motifs from Mis12c^Mtw1c, highlighting the Mis12^Mtw1_Spc24-helix and Dsn1^Cterm-helix helical motifs from Mis12c^Mtw1c that interact with Spc24 and Spc25 RWD domains at interface (3). The interaction between the CC3 region of the Mis12c^Mtw1c

stalk domain and the Spc24 RWD domain is also featured. **(C)** Structural details of the interactions between the CC3 region of the Mis12c$^{Mtw1c}$ stalk domain and the solvent-exposed hydrophobic patch on the Spc24 RWD domain at interface (4). **(D)** Structural details of the interactions between Dsn1$^{Cterm-helix}$ and Mis12$^{Mtw1\_Spc24-helix}$ motifs and the Spc24 and Spc25 RWD domains. **(E and F)** Comparison of *S. cerevisiae* (E) and human (F) KMN junction complexes at the Mis12c interface with Spc24:Spc25$^{RWD}$ domains. In both *S. cerevisiae* and human KMN junction complexes, the Dsn1$^{Cterm-helix}$ contacts Spc24:Spc25$^{RWD}$ interface. The comparison highlights the interchangeable roles played by the C termini of Mis12$^{Mtw1}$ in *S. cerevisiae* (Mis12$^{Mtw1\_Spc24\ helix}$) and Nsl1 in human in contacting the Spc24$^{RWD}$. In the *S. cerevisiae* KMN junction complex, the longer Mis12$^{Mtw1\_Spc24\ helix}$ contacts Dsn1$^{Cterm-helix}$. Structures superimposed on the Spc24$^{RWD}$ domain. Human KMN junction (PDB: 8PPR) from Yatskevich et al. (2024). Views in E and F are similar to (B).

the related budding yeast *Kluyveromyces lactis* (*Kl*Mis12c$^{Mtw1c}$) better recapitulated the contacts between head 1 and 2 observed in the density map. In the *Kl*Mis12c$^{Mtw1c}$ model, the head 2 domain is confidently predicted to contact head 1 (Fig. S5, G and H) and matched our docking of *S. cerevisiae* head 2 into cryo-EM density (Fig. S5, B and G). With this prediction of *Kl*Mis12c$^{Mtw1c}$, we built a model of *Sc*Mis12c$^{Mtw1c}$, which yielded an excellent fit into the density map (Fig. 6, B and C). This model rationalizes all cryo-EM density including that which could not be assigned using AF2 predictions of *Sc*Mis12c$^{Mtw1c}$ head 2 (gray density in Fig. S5 A). A linker within Nsl1 connecting CC1 and head 2 docks into the continuous density between CC1 and head 2 (Fig. 6, B and C), consistent with observed CL-MS crosslinks between the Nsl1 linker and CC1 residues Dsn1$^{K436}$, Nsl1$^{K105}$, and Nsl1$^{Y106}$ (Fig. 6 D). Within the N-IDR of *K. lactis* Dsn1, residues 201–230 (*Kl*Dsn1$^{201-230}$) are predicted to fold as two α-helices joined by a short linker (Fig. S5, G and H). Indeed, this region of *Kl*Dsn1 mediates association of the *Kl*Mis12c$^{Mtw1c}$ head 1 and head 2 domains *in vitro* and auto-inhibits the interaction of *Kl*Mis12c$^{Mtw1c}$ with *Kl*CENP-C$^{Mif2}$ and *Kl*CENP-U$^{Ame1}$ (Dimitrova et al., 2016). This region of Dsn1 is referred to as the Dsn1 auto-inhibitory segment (Dsn1$^{AI}$). Based on the sequence similarity of *Sc*Dsn1$^{228-257}$ and *Kl*Dsn1$^{201-230}$ (Fig. 6 F), we built Dsn1$^{AI}$ of *Sc*Mis12c$^{Mtw1c}$ into the α-helical-like density as *Sc*Dsn1$^{229-255}$ (Fig. 6, B and C) using the *Kl*Mis12$^{Mtw1c}$ AF2 model. This assignment is supported by a CL-MS crosslink between *Sc*Mis12$^{Mtw1}$ Glu84 in head 1 and *Sc*Dsn1$^{229-255}$ Ser240 (Fig. 6 E). Importantly, *Sc*Dsn1$^{229-255}$ incorporates the Aurora B$^{Ipl1}$ consensus sites Ser240 and Ser250 whose replacement with Asp enhanced the association of *Sc*Dsn1 with inner kinetochore proteins (Akiyoshi et al., 2013). In agreement with this, phosphomimetic mutants of the equivalent residues in *Kl*Dsn1$^{201-230}$ (Ser213 and Ser223) substantially increased the affinity of *Kl*Mis12c$^{Mtw1c}$ for N-terminal fragments of CENP-C$^{Mif2}$ and CENP-U$^{Ame1}$ (Dimitrova et al., 2016).

Consistent with our structure, an isothermal titration calorimetry (ITC) experiment showed that a peptide of Dsn1 residues 228–254 (Dsn1$^{228-254}$) bound to K$^{HB-RWD}$M with the entire Dsn1$^{N-IDR}$ (residues 1–257) deleted (K$^{HB-RWD}$M$^{Dsn1Δ257}$) with a $K_D$ of 1.3 μM (Fig. 6 Gi). In contrast, the Dsn1$^{228-254}$ peptide did not bind to K$^{HB-RWD}$M with both the Dsn1$^{N-IDR}$ and head 1 deleted (K$^{HB-RWD}$M$^{Dsn1Δ257\_Δhead1}$) (Fig. 6 Gii). Dsn1$^{229-255}$ is basic and projects Lys238, Arg245, and Arg248 toward a cluster of negatively charged residues in head 1 (Fig. 6 E). That the Dsn1$^{228-254}$ peptide–binding site corresponds to the site on head 1 associated with the α-helical-like density was also consistent with a threefold-increased $K_D$ for its association with the K$^{HB-RWD}$M$^{Dsn1Δ257\_Mis12-3A}$ mutant, which has a disrupted negatively charged patch on head 1 (Ala substitutions of Mis12$^{Mtw1}$ Glu69, Glu73, and Asp77) (Fig. 6, E and Giii).

### Phosphoregulation of CENP-C$^{Mif2}$ and CENP-U$^{Ame1}$ binding to Mis12c$^{Mtw1c}$

*K. lactis* Dsn1 residues, equivalent to those of *S. cerevisiae* Dsn1 that form the negatively charged patch of head 1 (Fig. 6 E), were implicated in binding the N termini of CENP-C$^{Mif2}$ and CENP-U$^{Ame1}$ (Dimitrova et al., 2016; Killinger et al., 2020). To understand the mechanism of Mis12c$^{Mtw1c}$ auto-inhibition, we superimposed the crystal structure of the *Kl*Mis12c$^{Mtw1c}$ head 1 bound to CENP-C$^{Mif2}$ (Dimitrova et al., 2016) onto our model of the KMN junction complex (Fig. 7 A). This indicated that Dsn1$^{AI}$ (Dsn1$^{229-255}$) overlaps with CENP-C$^{Mif2}$ bound to head 1, supporting a mechanism for Mis12c$^{Mtw1c}$ auto-inhibition in which Dsn1$^{AI}$ partially occludes the binding site of CENP-C$^{Mif2}$.

Phosphomimetic mutants of Aurora B$^{Ipl1}$ consensus sites in *Sc*Dsn1$^{229-255}$ and *Kl*Dsn1$^{201-230}$ relieve Mis12c$^{Mtw1c}$ auto-inhibition and increased CENP-C$^{Mif2}$ and CENP-U$^{Ame1}$ affinity for the complex (Dimitrova et al., 2016; Lang et al., 2018). *Sc*Dsn1$^{S250}$, an Aurora B$^{Ipl1}$ kinase substrate (Maskell et al., 2010; Westermann et al., 2003), projects toward the negatively charged patch of Mis12c$^{Mtw1c}$ head 1 (Fig. 6 E). Phosphorylation of *Sc*Dsn1$^{S250}$, and the equivalent residue in *Kl*Dsn1 (Ser223), would cause electrostatic repulsion between the phosphoserine and this negatively charged patch, and likely clash with Dsn1 residues of head 2. This might relieve Mis12c$^{Mtw1c}$ auto-inhibition by reducing the binding affinity of Dsn1$^{AI}$ for head 1, releasing it from the CENP-C$^{Mif2}$–binding site. *Sc*Dsn1$^{S240}$ and its equivalent residue, *Kl*Dsn1$^{S213}$, conform to the Aurora B$^{Ipl1}$ substrate consensus sequence. Although not reported to be phosphorylated by Aurora B$^{Ipl1}$, *Sc*Dsn1$^{S240}$ is phosphorylated *in vivo* (Lanz et al., 2021). *Sc*Dsn1$^{S240}$ projects away from head 1 (Fig. 6 E); however, phosphorylation of *Sc*Dsn1$^{S240}$ and *Kl*Dsn1$^{S213}$ could establish intramolecular interactions with positively charged residues *Sc*Dsn1$^{R237}$ and *Sc*Dsn1$^{R247}$ (conserved in *K. lactis*, Fig. 6 F) to relieve Mis12c$^{Mtw1c}$ auto-inhibition by destabilizing the Dsn1$^{AI}$ α-helices. Consistent with these proposals, an ITC experiment demonstrated that an *S. cerevisiae* Dsn1$^{228-254}$ peptide with phosphomimetic mutations at the Dsn1$^{S240}$ and Dsn1$^{S250}$ Aurora B$^{Ipl1}$ sites bound to K$^{HB-RWD}$M$^{Dsn1Δ257}$ with a $K_D$ of 6.0 μM (Fig. 6 Giv), a fivefold lower affinity than the wild-type peptide (Fig. 6 Gi).

To test the model that Dsn1$^{AI}$ auto-inhibits *Sc*Mis12c$^{Mtw1c}$, we used analytical SEC to assess binding of an N-terminal fragment of CENP-C$^{Mif2}$ comprising residues 1–63 (CENP-C$^{Mif2\_1-63}$) to K$^{HB-RWD}$M complexes containing Dsn1 N-terminal truncations. CENP-C$^{Mif2\_1-63}$ did not bind K$^{HB-RWD}$M containing full-length Dsn1 (Fig. 7, B and Cii) but did interact with K$^{HB-RWD}$M$^{Dsn1Δ257}$ (Fig. 7, B and Ciii), similar to previous reports for *Kl*Mis12c$^{Mtw1c}$ (Dimitrova et al., 2016). This indicated that the region mediating Mis12c$^{Mtw1c}$ auto-inhibition is within Dsn1$^{N-IDR}$. CENP-C$^{Mif2\_1-63}$ did not bind K$^{HB-RWD}$M$^{Dsn1Δ226}$, which lacked the N-terminal 226

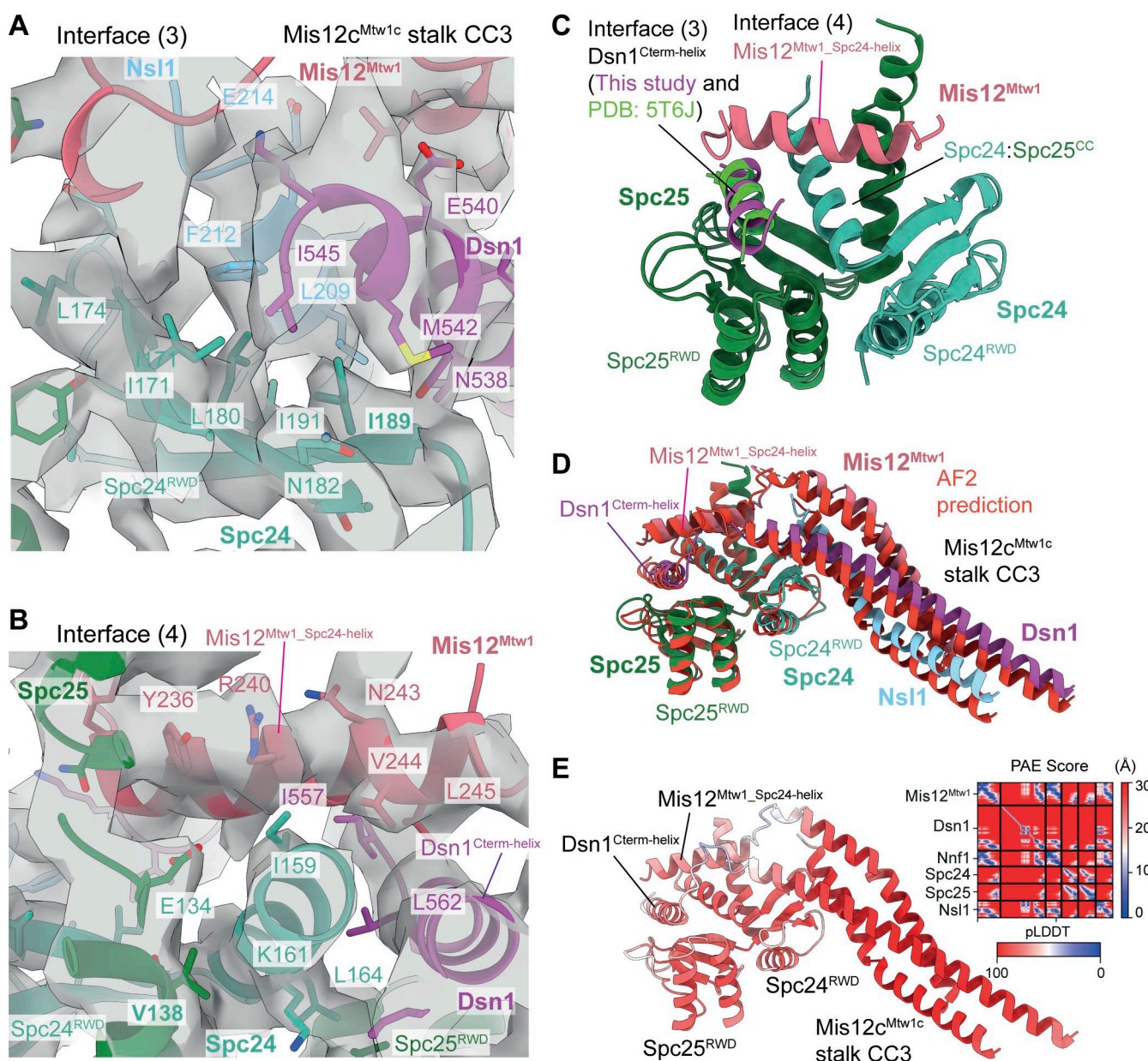

Figure 5. **Modeling of interactions between Mis12c^Mtw1c and Ndc80c. (A)** Molecular model of the interactions between the CC3 region of the Mis12c^Mtw1c stalk domain and the Spc24 and Spc25 RWD domains at interface (3) docked into the cryo-EM density map, showing density for select amino acid side chains relevant to the interaction. View is similar to Fig. 4 C. **(B)** Molecular model of the interactions between Mis12^Mtw1_Spc24-helix and Dsn1^Cterm-helix motifs with the Spc24 and Spc25 RWD domains at interface (4) docked into the cryo-EM density map, showing density for select amino acid side chains relevant to the interaction. View is similar to Fig. 4 D. **(C)** Structural alignment of the crystal structure of Spc24 and Spc25 RWD domains bound to Dsn1^560-576 peptide (PDB: 5T6J) (Dimitrova et al., 2016) onto the Mis12c^Mtw1c and Spc24 and Spc25 experimental molecular models (this study). The Spc24 and Spc25 chains derived from 5T6J and this study are colored identically, while Dsn1^Cterm-helix is colored distinctly according to whether it derives from 5T6J or this study. **(D)** AF2 model of the interaction between Spc24, Spc25, and Mis12c^Mtw1c (colored in red), structurally aligned against the Spc24, Spc25, and Mis12c^Mtw1c molecular models produced in this study (colored by chain). The prediction was performed using full-length and wild-type protein sequences for Spc24, Spc25, and Mis12c^Mtw1c subunits. **(E)** AF2 structure prediction presented in D colored by residue pLDDT score, with the PAE score plot shown in inset in the top right corner. PAE, predicted alignment error.

amino acids of Dsn1 (Fig. 7, B and Civ), demonstrating that residues 227–257 of Dsn1, corresponding to Dsn1^AI, are sufficient to confer Mis12c^Mtw1c auto-inhibition. Aurora B^Ipl1 Dsn1^S240E and Dsn1^S250E phosphomimetic mutants within Dsn1^AI in either full-length Dsn1 (Fig. 7, B and Cv) or Dsn1^Δ1–226 (Fig. 7, B and Cvi) were sufficient to abolish Mis12c^Mtw1c auto-inhibition. Together, these

results indicate that Dsn1^AI auto-inhibits Mis12c^Mtw1c by binding to head 1, and that either Aurora B^Ipl1 phosphorylation of Ser240 and Ser250 or deletion of the N-terminal 257 amino acids of Dsn1 relieves Mis12c^Mtw1c auto-inhibition.

We further investigated the effect of Dsn1^AI on the interaction between K^HB-RWDM (which has wild-type unphosphorylated

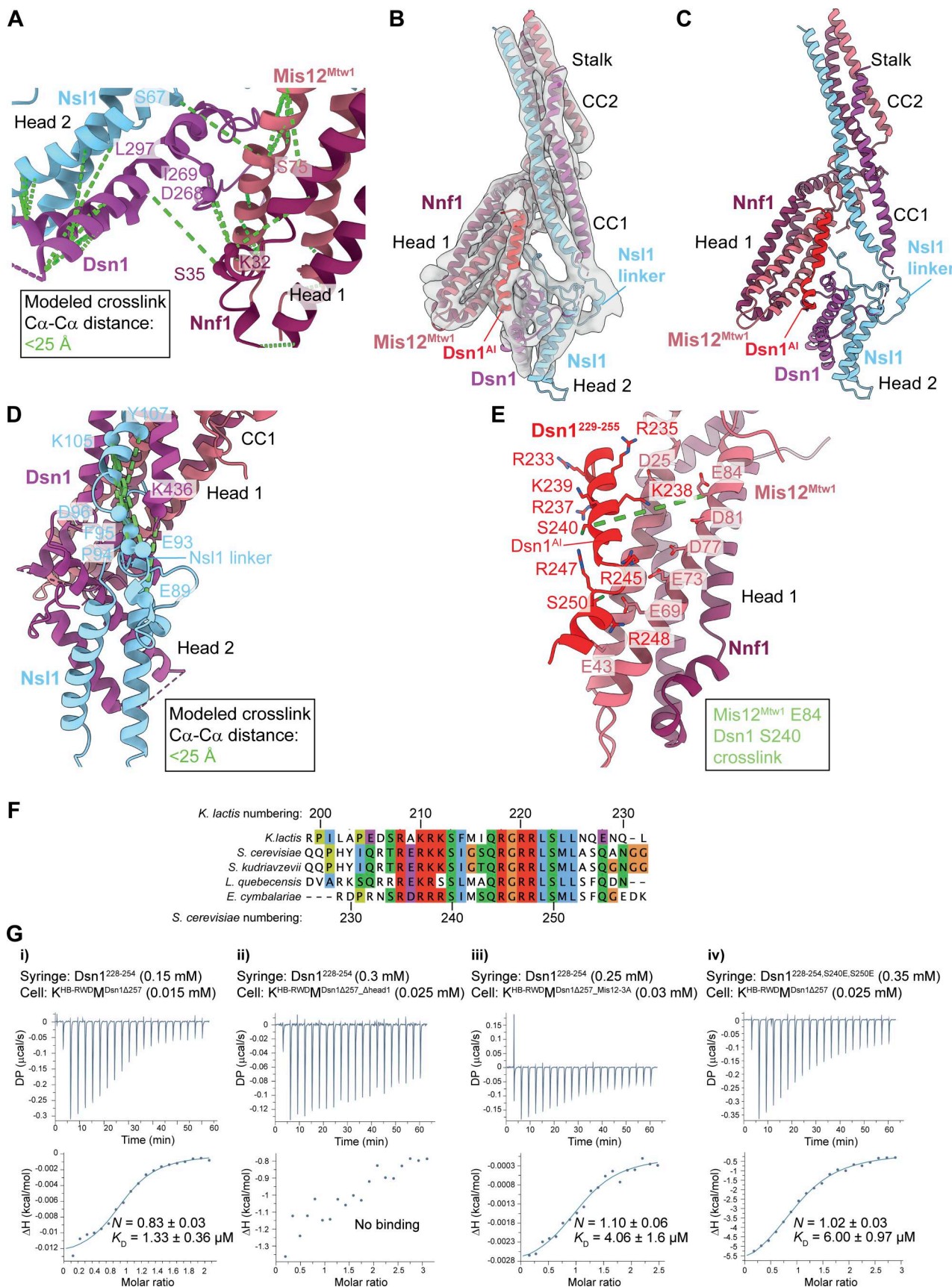

Figure 6. **Interactions between the Mis12c^Mtw1c head 1 and head 2 domains. (A)** Observed CL-MS crosslinks on the K^HB-RWDM complex, visualized using the molecular model from Fig. S5 B. Inter-subunit crosslinks between residues in the Mis12c^Mtw1c head 1 and head 2 domains are highlighted, and relevant

Cα atoms of crosslinked residues are shown as spheres. Crosslinks shown are between residues that satisfy the physical distance constraint of the chemical crosslinker. Crosslinks between two residues are only displayed if both residues are visible in the panel. **(B)** Cryo-EM density map with improved occupancy for the Mis12c$^{Mtw1c}$ head 2 domain (Fig. S4), interpreted and color-coded according to the experimental model of ScMis12c$^{Mtw1c}$. **(C)** Experimental cryo-EM model of ScMis12c$^{Mtw1c}$ head 1 and head 2 domains with Dsn1 auto-inhibitory region (Dsn1$^{AI}$) at the base of the ScMis12c$^{Mtw1c}$ stalk domain. **(D)** CL-MS crosslinks from the ScMis12c$^{Mtw1}$ dataset that satisfy the physical distance constraint of the chemical crosslinker, indicated in green, mapped onto ScMis12c$^{Mtw1c}$ head 1, head 2, and stalk domains and Dsn1$^{AI}$. **(E)** Molecular model showing the interaction between the ScMis12c$^{Mtw1c}$ head 1 domain and Dsn1$^{AI}$ (residues 229–255). ScMis12c$^{Mtw1c}$ head 2 domain is hidden. **(F)** Multiple sequence alignment of Dsn1 protein from distantly related budding yeast demonstrates that the KlDsn1$^{201-230}$ segment of the Dsn1 N-IDR is conserved. **(G)** ITC experiments to determine the identity of the unassigned α-helical-like density. (i) Titration of Dsn1$^{228-254}$ peptide to Knl1c: Mis12c$^{Mtw1c}$ complex (K$^{HB-RWD}$M$^{Dsn1Δ257}$) with Knl1$^{Spc105}$ residues 1–444 and Dsn1 residues 1–257 deleted. (ii) Titration of Dsn1$^{228-254}$ peptide to a head 1–deleted Knl1c:Mis12c$^{Mtw1c}$ complex (K$^{HB-RWD}$M$^{Dsn1Δ257\_Δhead1}$): Knl1$^{Spc105}$ residues 1–444, Dsn1 residues 1–257, Mis12$^{Mtw1}$ residues 1–106, and Nnf1 residues 1–104 deleted. (iii) Titration of Dsn1$^{228-254}$ peptide to K$^{HB-RWD}$M$^{Dsn1Δ257}$ complex (K$^{HB-RWD}$M$^{Dsn1Δ257\_Mis12-3A}$) containing mutant Mis12$^{Mtw1\_E69A,E73A,E77A}$. (iv) ITC data for phosphomimetic mutations at Aurora B sites in Dsn1$^{228-254}$. Titration of Dsn1$^{228-254}$ peptide with S240E and S250E phosphomimetic mutations (Dsn1$^{228-254,S240E,S250E}$) to Knl1c:Mis12c$^{Mtw1c}$ complex (K$^{HB-RWD}$M$^{Dsn1Δ257}$) with Knl1$^{Spc105}$ residues 1–444 and Dsn1 residues 1–257 deleted. $K_D$ and $n$ values are an average from three i) and two ii), iii), and iv) experiments (technical repeats).

Dsn1$^{AI}$) and a peptide modeled on the N-terminal 38 residues of CENP-C$^{Mif2}$ (CENP-C$^{Mif2}$_1–38). Using ITC, we did not detect an interaction between CENP-C$^{Mif2}$_1–38 and K$^{HB-RWD}$M (Fig. 7 Di). In contrast, CENP-C$^{Mif2}$_1–38 associated with K$^{HB-RWD}$M$^{2E}$ (incorporating Dsn1$^{S240E}$ and Dsn1$^{S250E}$ phosphomimetic mutants) with a $K_D$ of 0.98 µM (Fig. 7 Dii), an affinity similar to CENP-C$^{Mif2}$_1–41 binding to KlMis12c$^{Mtw1c}$ with phosphomimetic mutations of Dsn1$^{AI}$ ($K_D$ of 1.75 µM) (Dimitrova et al., 2016). These data are consistent with Aurora B$^{Ipl1}$ phosphorylation of Dsn1 overcoming auto-inhibition of Mis12c$^{Mtw1c}$ stimulating its association with CENP-C$^{Mif2}$.

Budding yeast Mis12c$^{Mtw1c}$ binds an N-terminal segment of CENP-U$^{Ame1}$; however, there are conflicting data concerning whether Mis12c$^{Mtw1c}$ auto-inhibition affects this interaction (Dimitrova et al., 2016; Hornung et al., 2014). A complex of *S. cerevisiae* CENP-U$^{Ame1}$ and CENP-Q$^{Okp1}$ (CENP-QU) binds to ScMis12c$^{Mtw1c}$ with wild-type Dsn1$^{AI}$ when both are at micromolar concentrations, as assessed by analytic SEC (Hornung et al., 2014). However, another study using fluorescence polarization showed that at lower concentrations (50 nM), the N-terminal 25 amino acids of *K. lactis* CENP-U$^{Ame1}$ did not interact with KlMis12c$^{Mtw1c}$ containing an intact Dsn1$^{AI}$ (Dimitrova et al., 2016). To investigate these apparent discrepancies and to understand how Dsn1$^{AI}$ regulates Mis12c$^{Mtw1c}$-CENP-U$^{Ame1}$ interactions, we superimposed an AF2 structure prediction of *S. cerevisiae* Mis12c$^{Mtw1c}$ bound to CENP-U$^{Ame1}$ (Fig. 8 A; and Fig. 9, C and D) onto our model of the KMN junction complex (Fig. 8 B). The AF2-predicted structure of CENP-U$^{Ame1}$ residues 1–33 (CENP-U$^{Ame1}$_1–33) bound to head 1 clashed with Dsn1$^{AI}$ (Fig. 8 B) more extensively than CENP-C$^{Mif2}$ bound to the Mis12c$^{Mtw1c}$ head 1 domain (Fig. 7 A; and Fig. 9, A and B). The mode of CENP-U$^{Ame1}$ binding to head 1, involving a basic α-helix lying along the acidic patch of head 1, is strikingly similar to how Dsn1$^{AI}$ interacts with head 1 (Fig. 6, C and E; Fig. 8, B and C; and Fig. 9 C). This suggests that Dsn1$^{AI}$ also inhibits the interaction between Mis12c$^{Mtw1c}$ and CENP-U$^{Ame1}$ by occluding the CENP-U$^{Ame1}$–binding site on head 1. The AF2 model predicts that CENP-U$^{Ame1}$ binds to a site on head 1 that is mainly distinct from the CENP-C$^{Mif2}$–binding site (Dimitrova et al., 2016), but that both ligands nevertheless share a small common binding surface (Fig. 8 D). This may explain why CENP-C$^{Mif2}$ and CENP-U$^{Ame1}$ associate with Mis12c$^{Mtw1c}$ mutually exclusively (Killinger et al., 2020).

Although analytical SEC showed that CENP-QU bound to K$^{HB-RWD}$M with full-length wild-type Dsn1 with partial dissociation (Fig. 8, E and F), consistent with previous results (Hornung et al., 2014), in an ITC experiment using 16 µM K$^{HB-RWD}$M, we did not detect K$^{HB-RWD}$M binding to 400 µM CENP-U$^{Ame1}$_1–33 (residues 1–33 of CENP-U$^{Ame1}$) (Fig. 8 Gi). Consistent with the model that Aurora B$^{Ipl1}$ kinase phosphorylation of Dsn1 strengthens CENP-U binding to KMN, KM$^{2E}$ with Dsn1$^{S240E}$ and Dsn1$^{S250E}$ phosphomimetic mutations bound CENP-U$^{Ame1}$_1–33 with a $K_D$ of 0.24 µM (Fig. 8 Gii). This binding affinity is sixfold higher than that observed for residues 1–25 of *K. lactis* CENP-U$^{Ame1}$_1-25 to KlMis12c$^{Mtw1c}$ containing phosphomimetic mutations ($K_D$ of 1.48 µM) (Dimitrova et al., 2016), an increase likely explained by the predicted interactions of CENP-U$^{Ame1}$ residues 27–33 with Mis12c$^{Mtw1c}$ head 1 (Fig. 8 A). In agreement with this, substituting Ala residues in CENP-U$^{Ame1}$_1–33 to disrupt interactions of CENP-U$^{Ame1}$ residues 27–33 with KM$^{2E}$ reduced CENP-U$^{Ame1}$_1-33-Mut binding affinity to a $K_D$ of 1.31 µM (Fig. 8 Giii). The increased affinity of CENP-U$^{Ame1}$_1–33 for KM$^{2E}$ relative to CENP-C$^{Mif2}$_1–38 may explain the binding of CENP-QU and not CENP-C$^{Mif2}$_1–63 to K$^{HB-RWD}$M as assessed by SEC (Fig. 7 C; and Fig. 8, E and F).

Human Mis12c binds CENP-C more tightly than its budding yeast counterpart, with nonphosphorylated Mis12c also capable of binding CENP-C (Petrovic et al., 2016; Yatskevich et al., 2024). One explanation for this difference is an N-terminal 13-residue extension in human CENP-C that interacts with Mis12c head 1 (Petrovic et al., 2016), reminiscent of how ScCENP-U$^{Ame1}$ is predicted to bind ScMis12c$^{Mtw1}$ (Fig. 9, C and E).

### Model for Aurora B$^{Ipl1}$-dependent activation mediated by conformational changes of head 2

Relative to our cryo-EM structure, head 2 in the AF2 ScMis12c$^{Mtw1c}$ model is rotated ∼90° and displaced from head 1, with Dsn1$^{AI}$ adopting an extended conformation (Fig. 6 C and Fig. S5 E). The separated configuration of head 1 and head 2 in the AF2 ScMis12c$^{Mtw1c}$ model is reminiscent of crystal structures of human Mis12c bound to CENP-C (Petrovic et al., 2016) and KlMis12c$^{Mtw1c}$ from which the auto-inhibitory segment was absent (Dimitrova et al., 2016) (Fig. 9, E and F). Additionally, the orientation of head 2 in the KlMis12c$^{Mtw1c}$ crystal structure closely matches that of the AF2 ScMis12c$^{Mtw1c}$ model (Fig. 9, F; and Fig. S5, E and F). This suggests that the AF2 ScMis12c$^{Mtw1c}$

**Turner et al.**
Cryo-EM structure of the budding yeast KMN complex

**Journal of Cell Biology**  12 of 28

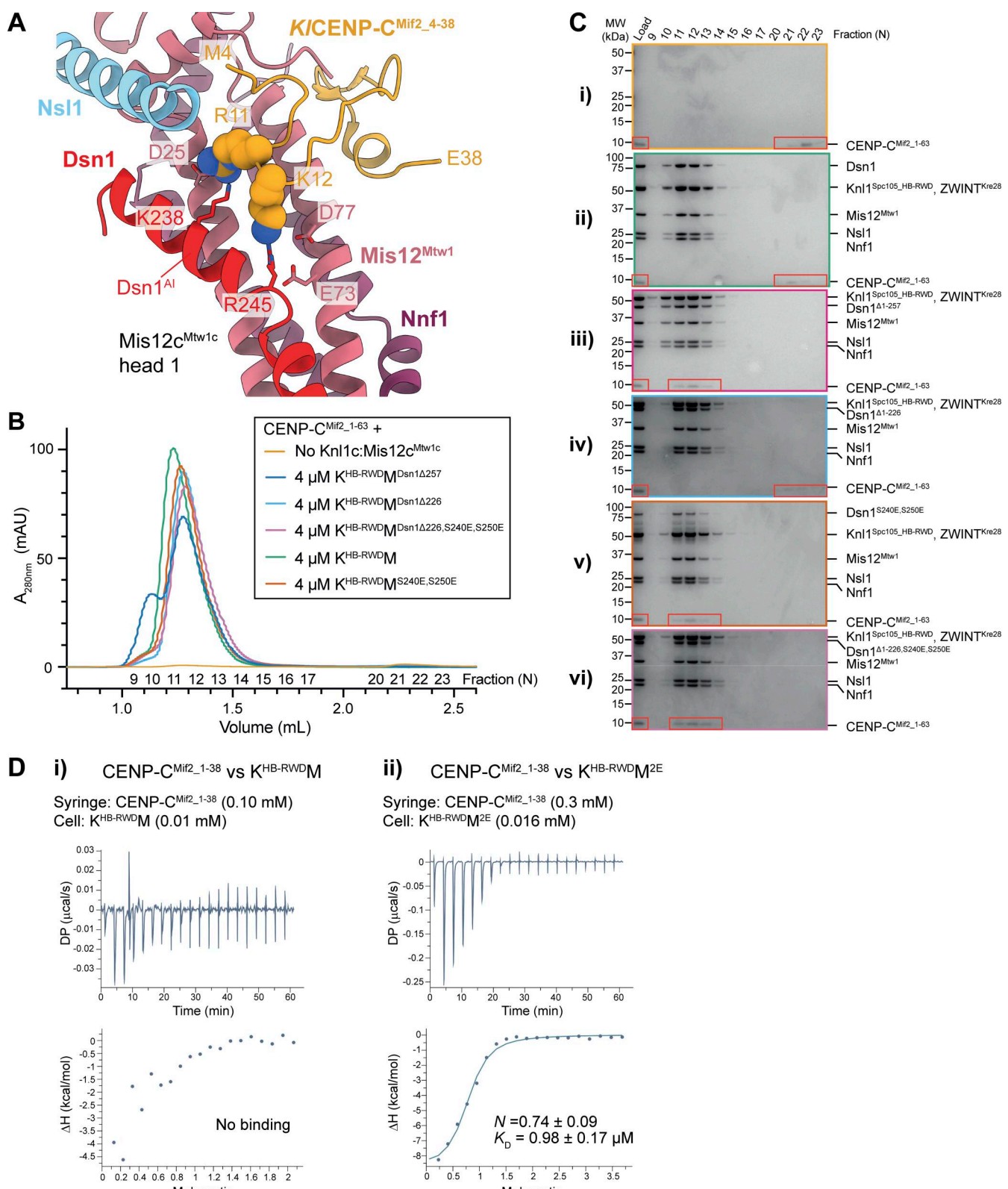

Figure 7. **Regulation of Mis12c$^{Mtw1c}$ auto-inhibition by Aurora B kinase and its effect on the interaction with CENP-C$^{Mif2}$. (A)** Structural alignment of the crystal structure of *K. lactis* CENP-C$^{Mif2}$ residues 4–36 (*Kl*CENP-C$^{Mif2\_4-36}$) as bound to the *Kl*Mis12c$^{Mtw1c}$ head 1 domain (PDB: 5T59) (Dimitrova et al., 2016) to the experimental model of *Sc*Mis12c$^{Mtw1c}$ (this study). The models are colored by chain; Mis12$^{Mtw1}$ and Nnf1 are colored identically in 5T59 and our experimentally determined model. *Kl*CENP-C$^{Mif2\_4-36}$ is shown in space filling representation to highlight lack of steric clash, and overlapping binding site, with Dsn1$^{AI}$. Only *Kl*CENP-C$^{Mif2\_4-36}$ is shown. **(B)** SEC elution chromatograms of attempts to reconstitute the interaction between Knl1c:Mis12c$^{Mtw1c}$ complexes containing different Dsn1 N-terminal truncation and/or phosphomimetic mutants and CENP-C$^{Mif2}$ residues 1–63 (CENP-C$^{Mif2\_1-63}$) indicated within red boxes.

**(C)** Coomassie brilliant blue–stained SDS-PAGE gels of the chromatograms shown in B. **(B and C)** $K^{HB-RWD}M$ = full-length Dsn1. $K^{HB-RWD}M^{2E}$ = full-length Dsn1 containing Dsn1$^{S240E}$ and Dsn1$^{S250E}$ phosphomimetic mutations. $K^{HB-RWD}M^{Dsn1\Delta257}$ = Dsn1 residues 1–257 are deleted (Dsn1$^{\Delta1-257}$). $K^{HB-RWD}M^{Dsn1\Delta226}$ = Dsn1 residues 1–226 are deleted (Dsn1$^{\Delta1-226}$). $K^{HB-RWD}M^{Dsn1\Delta226-2E}$ = Dsn1 residues 1–226 are deleted and Dsn1$^{S240E}$ and Dsn1$^{S250E}$ phosphomimetic mutations (Dsn1$^{\Delta1-226,S240E,S250E}$). CENP-C$^{Mif2}$_1–63 is indicated within a red box. **(D)** ITC data for (i) CENP-C$^{Mif2}$_1–38 interactions with KM, and (ii) CENP-C$^{Mif2}$_1–38 interactions with KM$^{2E}$. $K_D$ and $n$ values are an average from three experiments (technical repeats). Source data are available for this figure: SourceData F7.

model represents a plausible prediction of the Aurora B$^{Ipl1}$-phosphorylated, activated state of $Sc$Mis12c$^{Mtw1c}$. Consistent with this proposal, binding sites for CENP-C$^{Mif2}$ and CENP-U$^{Ame1}$ on head 1 are also not occluded by Dsn1$^{AI}$ in this conformation (Fig. 9, A and C; and Video 2). Dsn1$^{AI}$-mediated auto-inhibition of $Sc$Mis12c$^{Mtw1c}$ resembles mechanisms proposed for human Mis12c in which Dsn1$^{AI}$ bridges the two head domains, and, together with head 2, occludes the inner kinetochore–binding sites on head 1 (Fig. 9, G and H) (Yatskevich et al., 2024).

#### Model of the entire *S. cerevisiae* KMN complex

We were unable to identify particles in our cryo-EM datasets that contained signal for the predicted Knl1c helical bundle domain (Fig. S1 A, and Fig. 10, A and B), nor for the Spc24:Spc25 coiled coils and Ndc80:Nuf2 subunits (Fig. S1 A, and Fig. 10, A and C). AF2 generated a structure prediction for the Knl1c helical bundle (residues 492–632 of Knl1$^{Spc105}$ and 7–194 of ZWINT$^{Kre28}$) with high confidence (Fig. 10, A and B). This helical bundle is connected to the Knl1c RWD domains by two segments in Knl1$^{Spc105}$ and ZWINT$^{Kre28}$ ("linker 1" and "linker 2") that were predicted to adopt extended conformations with lower pLDDT scores (Fig. 10, A and B). Flexibility of linker 2 would explain why we were unable to resolve Knl1c domains beyond the RWD domains. An equivalent linker does not interrupt the coiled coil of human Knl1c, possibly explaining why its cryo-EM density was visible (Yatskevich et al., 2024). We also performed AF2 structure predictions of Ndc80c. We observed a bent conformation that brings the Ndc80c microtubule-binding elements (Ndc80c CH domains and Ndc80 N-tail) into proximity with an intermediate section of the Ndc80:Nuf2 coiled coil (Fig. 10, A and C). Although isolated Ndc80c has been observed to exhibit modest bending by negative-stain EM (Wang et al., 2008), such a highly bent conformation is unlikely to be adopted in the presence of microtubules or Dam1c rings (Aravamudhan et al., 2014; Muir et al., 2023). In the structure predicted by AF2, the Spc24:Spc25 coiled coils are connected to their respective RWD domains by a linker ("Spc24:Spc25 linker") (Fig. 10 C). We expect that we were unable to resolve domains beyond the Spc24:Spc25 linker in our cryo-EM analysis because it flexibly tethers the Ndc80c coiled coils and microtubule-binding elements to the KMN junction. FRET measurements performed in the absence of both microtubules and Dam1c showed that Ndc80c bending is dynamic, and that more tightly bent Ndc80c conformations than those predicted by AF2 are possible (Scarborough et al., 2019). However, these tightly bent conformations are adopted with low propensity in complex with Mis12c$^{Mtw1c}$ (Scarborough et al., 2019).

We generated a model of the *S. cerevisiae* KMN complex by superimposing the AF2 predictions of Knl1c and Ndc80c onto their corresponding RWD domains in the KMN junction complex

structure (Fig. 10 A). The close proximity of the Knl1c helical bundle to the Spc24:Spc25 coiled coils in the hypothetical model is consistent with previous CL-MS analysis of the budding yeast KMN complex (Ghodgaonkar-Steger et al., 2020), although the exact positions of the domains are probably not correct. The Knl1c helical bundle is also close to the Ndc80c microtubule-binding elements.

### Discussion

Our cryo-EM structure and biochemical data of the *S. cerevisiae* KMN junction complex revealed both similarities and differences in how budding yeast Mis12c$^{Mtw1c}$ and human Mis12c interact with Knl1c and Ndc80c, and the mechanism of Aurora B$^{Ipl1}$-mediated regulation of KMN assembly onto the inner kinetochore. For both species, Mis12c$^{Mtw1c}$/Mis12c interacts with Knl1c and Ndc80c through two distinct interaction modes. In a conserved mode of interaction, the coiled-coil (CC3) α-helices of the Mis12c$^{Mtw1c}$/Mis12c stalk create a rigid interface for docking with Knl1c and Ndc80c (Fig. 1, F and G). Specifically, Dsn1:Nsl1 contact Ndc80c, whereas Dsn1:Nnf1 contact Knl1c. The other mode of interaction is mediated by C-terminal extensions of Mis12c$^{Mtw1c}$/Mis12c subunits that interact with sites on Knl1c and Ndc80c. Conserved in both human and budding yeast KMN is the interaction of Dsn1$^{Cterm-helix}$ at the interface of the Spc24:Spc25 RWD domains of Ndc80c (Fig. 1, F and G), a site that also binds the Ndc80c-binding α-helix of CENP-T (Malvezzi et al., 2013; Nishino et al., 2013). The most striking difference between budding yeast and human KMN is the contrasting functions of the Mis12$^{Mtw1}$ and Nsl1 subunits to contact Knl1c and Ndc80c. In human KMN, an α-helix of Nsl1 (Nsl1$^{Spc24-helix}$) contacts the Ndc80c Spc24:Spc25 RWD domains (Fig. 1 G). Nsl1 then engages Knl1c at Knl1$^{RWD-N}$, and at the Knl1$^{RWD-N}$:ZWINT interface. In budding yeast, these functions are substituted by Mis12$^{Mtw1}$. Mis12$^{Mtw1\_Spc24-helix}$ contacts the Spc24:Spc25 RWD domains, which also stabilizes the Dsn1$^{Cterm-helix}$:Spc24$^{RWD}$:Spc25$^{RWD}$ interaction (Fig. 1 F). The Mis12$^{Mtw1}$ chain then interacts with the Knl1$^{Spc105\_RWD-N}$:Kre28$^{ZWINT}$ interface of Knl1c.

Our hypothetical model of the entire *S. cerevisiae* KMN complex including the flexible tethering of other Knl1c and Ndc80c domains to the RWD domains of the KMN junction could have functional consequences for SAC signaling (Fig. 10). Experiments in humans and budding yeast are consistent with a model in which SAC signaling is activated at kinetochores that are not attached to microtubules (Rieder et al., 1995; Wells and Murray, 1996), and that therefore do not incorporate Dam1c rings (Li et al., 2002; Tanaka et al., 2005, 2007). Under these conditions, Ndc80c might adopt a conformation in which its microtubule-binding elements localize near the Knl1c helical bundle domain and the N-terminal disordered region of

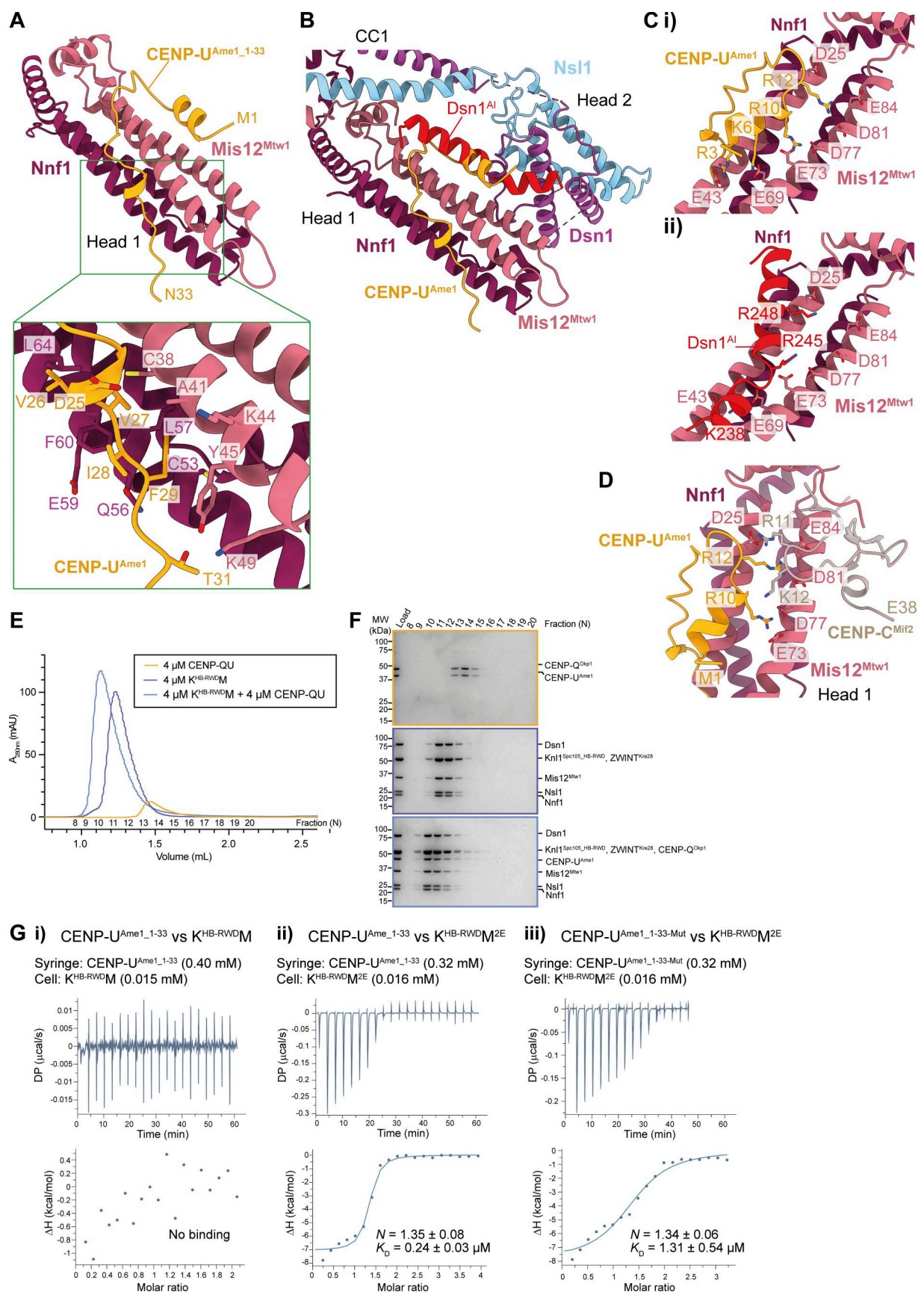

Figure 8. **Effect of Mis12c$^{Mtw1c}$ auto-inhibition on the interaction with CENP-U$^{Ame1}$. (A)** AF2 structure prediction of the interaction between *S. cerevisiae* Mis12c$^{Mtw1c}$ and CENP-U$^{Ame1}$ colored by chain, showing only the Mis12c$^{Mtw1c}$ head 1 domain and residues 1–33 of CENP-U$^{Ame1}$ predicted to interact with

Mis12c[Mtw1c]. The structure prediction was performed using full-length and wild-type proteins apart from Dsn1, from which Dsn1 residues 1–257 were excluded to avoid Mis12c[Mtw1c] auto-inhibition. Top panel: Overview of the complex. Lower panel: Details of CENP-U[Ame1] residues Asp25 to Thr31 interactions with head 1. Val26, Val27, Ile28, and Phe29 of CENP-U[Ame1] engage a hydrophobic pocket on head 1, augmented by hydrogen bonds from Asp25 and Thr31. **(B)** Superimposition of the AF2 model presented in Fig. 8 A onto the experimental model of ScMis12c[Mtw1c] (this study). The subunits are colored by chain; Mis12[Mtw1] and Nnf1 are colored identically in both the AF2 model and the experimentally determined model. The apparent steric clash between CENP-U[Ame1] residues 1–24 and Dsn1[AI] is highlighted. **(C)** Comparison of the AF2 model of CENP-U[Ame1] bound to ScMis12c[Mtw1c] (top panel) and Dsn1[AI] of ScMis12c[Mtw1c] (bottom panel). Both CENP-U[Ame1] and Dsn1[AI] form basic α-helices that engage the acid patch of head 1. **(D)** CENP-C[Mif2] and CENP-U[Ame1]–binding sites overlap. **(E)** SEC elution chromatograms of CENP-QU alone, Knl1c:Mis12c[Mtw1c] complexes that lack residues 1–444 of Knl1[Spc105] alone (K[HB-RWD]M), or after reconstitution of the interaction between CENP-QU and K[HB-RWD]M. **(F)** Coomassie brilliant blue–stained SDS-PAGE gels of the experiments in E. **(G)** ITC data for (i) CENP-U[Ame1_1-33] interactions with KM, (ii) CENP-U[Ame1_1-33] interactions with KM[2E], and (iii) the CENP-U[Ame1_1-33] mutant with Ala substitutions of D25, V27, I28, F29, T31 predicted to interact with Mis12c[Mtw1c] (Fig. 8, C and D) to KM[2E]. $K_D$ and $n$ values are an average from three experiments (technical repeats). Source data are available for this figure: SourceData F8.

Knl1[Spc105]. This may facilitate Mps1, bound proximal to the Ndc80c CH domains (Parnell et al., 2024; Pleuger et al., 2024; Zahm and Harrison, 2024), in phosphorylating MELT motifs in the Knl1[Spc105] N-terminal disordered region to stimulate SAC signaling.

We determined a lower resolution cryo-EM reconstruction of the KMN junction complex in which head 2 of Mis12c[Mtw1c] was resolved. The auto-inhibitory region Dsn1[AI] suppressed binding of Mis12c[Mtw1c] to the N-terminal segments of both CENP-C[Mif2] and CENP-U[Ame1]. Phosphomimetic mutations of Aurora B[Ipl1] sites within a peptide modeled on Dsn1[AI] reduced its affinity for the K[HB-RWD]M[Dsn1Δ257] complex. Additionally, the same phosphomimetic mutations in Dsn1[AI] activated K[HB-RWD]M binding of both CENP-C[Mif2] and CENP-U[Ame1]. These results suggest that the intramolecular association of Dsn1[AI] with Mis12c[Mtw1c] head 1 is responsible for Mis12c[Mtw1c] auto-inhibition, and that Aurora B[Ipl1] phosphorylation of serine residues within this region relieves Mis12c[Mtw1c] auto-inhibition by reducing the binding affinity of Dsn1[AI] for head 1. Cryo-EM structures and AF2 models suggest that this is because the Dsn1[AI] and CENP-U[Ame1] would sterically clash if both bound to head 1 (Fig. 8 C). Such a steric clash does not occur between CENP-C[Mif2] and Dsn1[AI]; instead, the two elements compete for contacts with the same amino side chains within head 1 (Fig. 7 A). The higher affinity of CENP-U[Ame1] for KMN relative to CENP-C[Mif2], as evidenced by our ITC data, and the binding of CENP-QU, but not CENP-C[Mif2_1-63], to unphosphorylated K[HB-RWD]M on SEC (this study and Hornung et al., 2014), has implications for understanding observations that CENP-U[Ame1] is essential for yeast viability (Hornung et al., 2014; Schmitzberger et al., 2017), whereas the N terminus of CENP-C[Mif2] is dispensable (Hornung et al., 2014). Aurora B[Ipl1] localization to CCAN is necessary for the correction of erroneous kinetochore microtubule attachments (Fischböck-Halwachs et al., 2019; García-Rodríguez et al., 2019; Li et al., 2023). Our findings therefore suggest a mechanism by which attachments of mitotic chromosomes to spindle microtubules, mediated by the connection of KMN complexes to centromere-bound CCAN through either CENP-C[Mif2] or CENP-U[Ame1], are strengthened at kinetochores that are able to correct errors in these attachments.

Our work has revealed that the mechanisms mediating assembly and auto-inhibition of the KMN complex are conserved at the structural level in humans and budding yeast. Furthermore, we showed that Aurora B[Ipl1]-mediated phosphorylation overcomes auto-inhibition of human Mis12c and budding yeast

Mis12c[Mtw1c] through similar structural mechanisms. In humans, Aurora B regulation of Mis12c auto-inhibition probably restricts KMN complex recruitment to mitotic kinetochores that are competent to correct erroneous kinetochore–microtubule attachments through error correction signaling (Yatskevich et al., 2024). This and a previous study (Hornung et al., 2014) revealed that in budding yeast the inner kinetochore subunit CENP-U[Ame1], when in complex with CENP-Q[Okp1], binds to unphosphorylated Mis12c[Mtw1c]. In contrast to humans, the KMN complex is attached to the centromere throughout almost the entire cell cycle in budding yeast, with only transient disassembly of kinetochores during S-phase (Kitamura et al., 2007). Furthermore, Aurora B[Ipl1] is recruited to kinetochores from G1 to metaphase, being depleted from centromeres under tension at metaphase and relocalizing to the spindle midzone (Buvelot et al., 2003; Campbell and Desai, 2013; Edgerton et al., 2023; Nerusheva et al., 2014; Shimogawa et al., 2009). Consequently, relief of Mis12c[Mtw1c] auto-inhibition from G1-phase to metaphase through phosphorylation by Aurora B[Ipl1] most likely strengthens preexisting connections between the inner and outer kinetochore, allowing bioriented kinetochores to withstand the pulling and pushing forces exerted by spindle microtubules. However, the diminished, but not abolished, affinity of both CENP-U[Ame1] for unphosphorylated Mis12c[Mtw1c] is possibly still able to support kinetochore–microtubule attachment during anaphase after Aurora B[Ipl1] has localized away from kinetochores.

## Materials and methods

### Cloning of S. cerevisiae CENP-C[Mif2] and S. cerevisiae KMN complex proteins

Genes encoding S. cerevisiae Ndc80, Nuf2, Spc24, Spc25, Dsn1, Mis12[Mtw1], Nnf1, Nsl1, residues 1–444 of Knl1[Spc105], and ZWINT[Kre28] were each amplified by polymerase chain reaction (PCR) from S. cerevisiae strain S288c genomic DNA (Zhang et al., 2016). Details of the DNA oligonucleotide primer used in this study are listed in Table S3.

Genes encoding Ndc80 and Nuf2 were cloned into the same pU1 plasmid (Zhang et al., 2016). Genes encoding Spc24 and Spc25 were cloned into the same pF1 plasmid (Zhang et al., 2016). Genes encoding Knl1[Spc105] residues 445–917 (Knl1[Spc105_HB-RWD]) and ZWINT[Kre28] were cloned into the same modified pF1 plasmid. Genes encoding Knl1[Spc105] residues 710–917 (Knl1[Spc105_RWD]) and

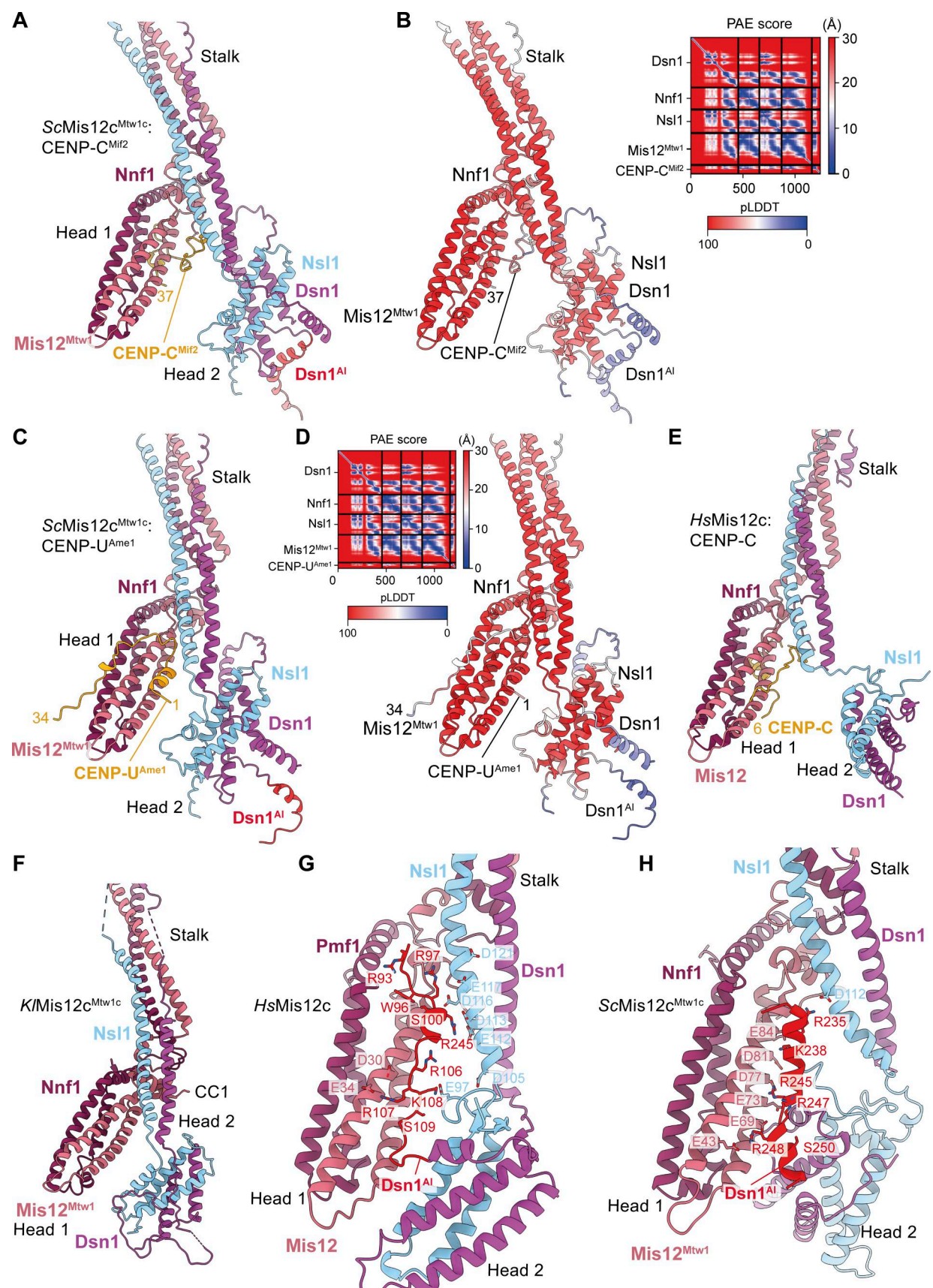

Figure 9. **Comparison of cryo-EM structure of *Sc*Mis12c^Mtw1c with AF2 models of *Sc*Mis12c^Mtw1c in complex with CENP-C^Mif2 and CENP-U^Ame1 and *Kl*Mis12c and human Mis12c. (A)** AF2 model of *Sc*Mis12c^Mtw1c in complex with CENP-C^Mif2 (residues of 1–55). **(B)** Model colored by residue pLDDT score. The

PAE score plot is shown to the right. **(C)** AF2 model of *Sc*Mis12c$^{Mtw1c}$ in complex with CENP-U$^{Ame1}$ (residues 1–34 of CENP-U$^{Ame1}$ shown). **(D)** Model colored by the residue pLDDT score. The PAE score plot is shown to the right. In both AF2 models, head 2 is predicted to rotate by 180° about two orthogonal axes relative to head 2 of the inactive *Sc*Mis12c$^{Mtw1c}$ structure in Fig. 6 C. **(E)** Crystal structure of the heads and stalk region of activated human Mis12c in complex with CENP-C (residues 1–48) (PDB: 5LSK) (Petrovic et al., 2016). **(F)** Crystal structure of the heads and stalk region of *K. lactis* Mis12c$^{Mtw1c}$ (PDB: 5T58) (Dimitrova et al., 2016). Head 2 adopts a similar conformation to head 2 of the AF2 model of *Sc*Mis12c$^{Mtw1c}$ (Fig. S5 E). **(G and H)** Comparison of the cryo-EM structures of the heads and stalk region of human Mis12c in the inactive unphosphorylated conformation (PDB: 8PPR) (Yatskevich et al., 2024) (G) with the equivalent regions of *S. cerevisiae* Mis12c$^{Mtw1c}$ (this study) (H). The structures shown in G and H were superimposed on the Mis12 chain of head 1. PAE, predicted alignment error.

ZWINT$^{Kre28}$ residues 268–385 (ZWINT$^{Kre28\_RWD}$) were cloned into the same modified *pF1* plasmid. TEV and HRV-3C cleavable Twin-Strep-tag II (TS) tags were introduced into the C termini of Nuf2 and ZWINT$^{Kre28}$, respectively, and a tandem tag comprising maltose-binding protein (MBP) fused to monomeric enhanced green fluorescent protein (mEGFP) was introduced into the N terminus of Knl1$^{Spc105\_RWD}$. Genes encoding Dsn1, Mis12$^{Mtw1}$, Nsl1, and Nnf1 were cloned into the same modified *pF1* plasmid for coexpression with Knl1$^{Spc105}$ and ZWINT$^{Kre28}$. For the expression of Mis12c$^{Mtw1c}$ and Mis12c$^{Mtw1c}$ mutants, Dsn1 and Nsl1 were cloned into the same modified *pF1* plasmid and TS was introduced into the N terminus of Dsn1. This plasmid was used to generate a virus for coexpression with a second virus generated from a another modified *pF1* plasmid containing a Mis12$^{Mtw1}$-Nnf1 pair. Constructs for Dsn1$^{Δ1–226}$, Dsn1$^{Δ1–226,S240E,S250E}$, Dsn1$^{Δ1–257}$, Dsn1$^{S240E,S250E}$, Mis12$^{Mtw1\_E69A,E73A,E77A}$, Mis12$^{Mtw1\_Δ272-C}$, and Nnf1$^{Δ180-C}$ were created using primers supplied from Merck and the KLD reaction kit supplied by New England Biolabs and cloned into plasmids containing all other Mtw1c subunits for coexpression with Knl1$^{Spc105\_HB-RWD}$ and ZWINT$^{Kre28}$. Constructs for Mis12$^{Mtw1\_Δ272-C}$ and Nnf1$^{Δ\_180-C}$ were additionally cloned into the plasmid containing the Mis12$^{Mtw1}$-Nnf1 pair and expressed together with a virus made from the plasmid containing a Dsn1-Nsl1 pair. To express Knl1:Mis12c$^{Mtw1c}$ with the Mis12c head 1 domain deleted, constructs for Mis12$^{Mtw1\_Δ1–106}$ and Nnf1$^{Δ1–104}$ were cloned with Dsn1$^{Δ1–257}$ and wild-type Nsl1 into the same modified *pF1* plasmid and coexpressed with Knl1$^{Spc105}$ and ZWINT$^{Kre28}$. To express Knl1:Mis12c$^{Mtw1c}$ with mutated Dsn1, constructs for Mis12$^{Mtw1}$, Nnf1, Nsl1, and mutant Dsn1 were cloned into the same modified *pF1* plasmid and coexpressed with Knl1$^{Spc105}$ and ZWINT$^{Kre28}$.

The gene encoding *S. cerevisiae* CENP-C$^{Mif2}$ residues 1–63 (CENP-C$^{Mif2\_1–63}$) was amplified from *S. cerevisiae* strain S288c genomic DNA and cloned into a pET28 plasmid carrying a Kanamycin selectable marker. A 6xHis-SUMO-tag was introduced into the N terminus of the protein.

To construct plasmids for yeast strain construction, the promoter, coding regions, and terminator regions defined for the *MTW1* and *NNF1* genes in the yeast genome (Euskirchen, 2002) were amplified from *S. cerevisiae* strain S288c genomic DNA. The *MTW1* fragments were cloned into a modified *pRS304* plasmid (Frigola et al., 2013), and a HA$_3$ tag was introduced into the N terminus of the protein in a single step using the NEBuilder HiFi DNA assembly reaction kit supplied by New England Biolabs. The *NNF1* fragments were cloned into a modified *pRS305* plasmid (Frigola et al., 2013), and a V5$_3$ tag was introduced into the N terminus of the protein in a single step using the NEBuilder HiFi DNA assembly reaction kit. A *pRS304* series plasmid and a *pRS305* series plasmid encoding HA$_3$-Mis12$^{Mtw1\_Δ272-C}$ and V5$_3$-Nnf1$^{Δ180-C}$ were constructed from the plasmids described

above using PCR with primers supplied by Merck followed by processing with the KLD reaction kit supplied by New England Biolabs.

## Expression and purification of KMN subcomplexes, and CENP-C$^{Mif2\_1-63}$ and CENP-QU complexes

Baculoviruses for the expression of KMN subcomplexes were generated from the plasmids described above using methods as described previously (Zhang et al., 2016). Ndc80c was expressed from two baculoviruses encoding a Ndc80-Nuf2 gene pair and a Spc24-Spc25 gene pair in High-5 insect cells. Six-subunit Knl1c:Mis12c$^{Mtw1c}$ complexes were expressed from two baculoviruses encoding a Knl1$^{Spc105\_HB-RWD}$-ZWINT$^{Kre28}$ gene pair and a Dsn1-Nsl1-Mis12$^{Mtw1}$-Nnf1 gene quartet in High-5 insect cells. Isolated Mis12c$^{Mtw1c}$ was expressed from two baculoviruses encoding Mis12$^{Mtw1}$-Nnf1 and Dsn1-Nsl1 gene pairs. The Sf9 and High-5 (BTI-TN-5B1-4) insect cell lines used for baculovirus generation and protein complex expression were not tested for *Mycoplasma* contamination or authenticated. 2-liter roller bottles containing 500 ml suspension cultures of High-5 cells were typically infected with 2.5% vol/vol of P3 Sf9 suspension cell culture, and the High-5 expression cultures were subsequently harvested 42–70 h after infection when the cell viability had dropped to 75–85%.

All purification steps were carried out at 4°C or on ice. Cells expressing Ndc80c, Knl1c:Mis12c$^{Mtw1c}$, or Mis12c$^{Mtw1c}$ were lysed by sonication in lysis buffer (50 mM Tris.HCl, pH 8.5, 8 mM benzamidine, 1 mM EDTA, 1 mM TCEP, 0.2 mM PMSF, cOmplete EDTA-free protease inhibitor tablets) containing Benzonase and either 300 mM NaCl and 7.5% glycerol (Ndc80c), 250 mM NaCl and 7.5% glycerol (Mis12c$^{Mtw1c}$), or 200 mM NaCl and 5% glycerol (Knl1c:Mis12c$^{Mtw1c}$). Lysates were cleared by ultracentrifugation at 38,000×*g* for one h and loaded onto Strep-Tactin superflow plus cartridges (Qiagen). Immobilized complexes were washed with lysis buffer after adjusting to pH 8.0. Immobilized Knl1c:Mis12c$^{Mtw1c}$ complexes were subjected to further washes with a buffer containing ATP (50 mM Tris.HCl, pH 8.0, 175 mM KCl, 10 mM MgCl$_2$, 2.5 mM ATP sodium salt, 1 mM TCEP, 2.5% glycerol) followed by a buffer containing 50 mM Tris.HCl, pH 8.0, 200 mM NaCl, 2.5% glycerol. Bound complexes were then eluted using a buffer containing 50 mM Tris.HCl, pH 8.0, 2.5 mM d-desthiobiotin, 1 mM TCEP, and either 100 mM NaCl (Ndc80c and Mis12c$^{Mtw1c}$) or 200 mM NaCl (Knl1c:Mis12c$^{Mtw1c}$).

Ndc80c was further purified by loading the Strep-Tactin eluate onto a Resource Q anion exchange column (Cytiva). Bound complex was eluted with buffer containing 50 mM Tris.HCl, pH 8.0, 1 mM TCEP, 2.5% glycerol, and a gradient from 100 to 1,000 mM NaCl over 15–20 column volumes. Peak fractions were

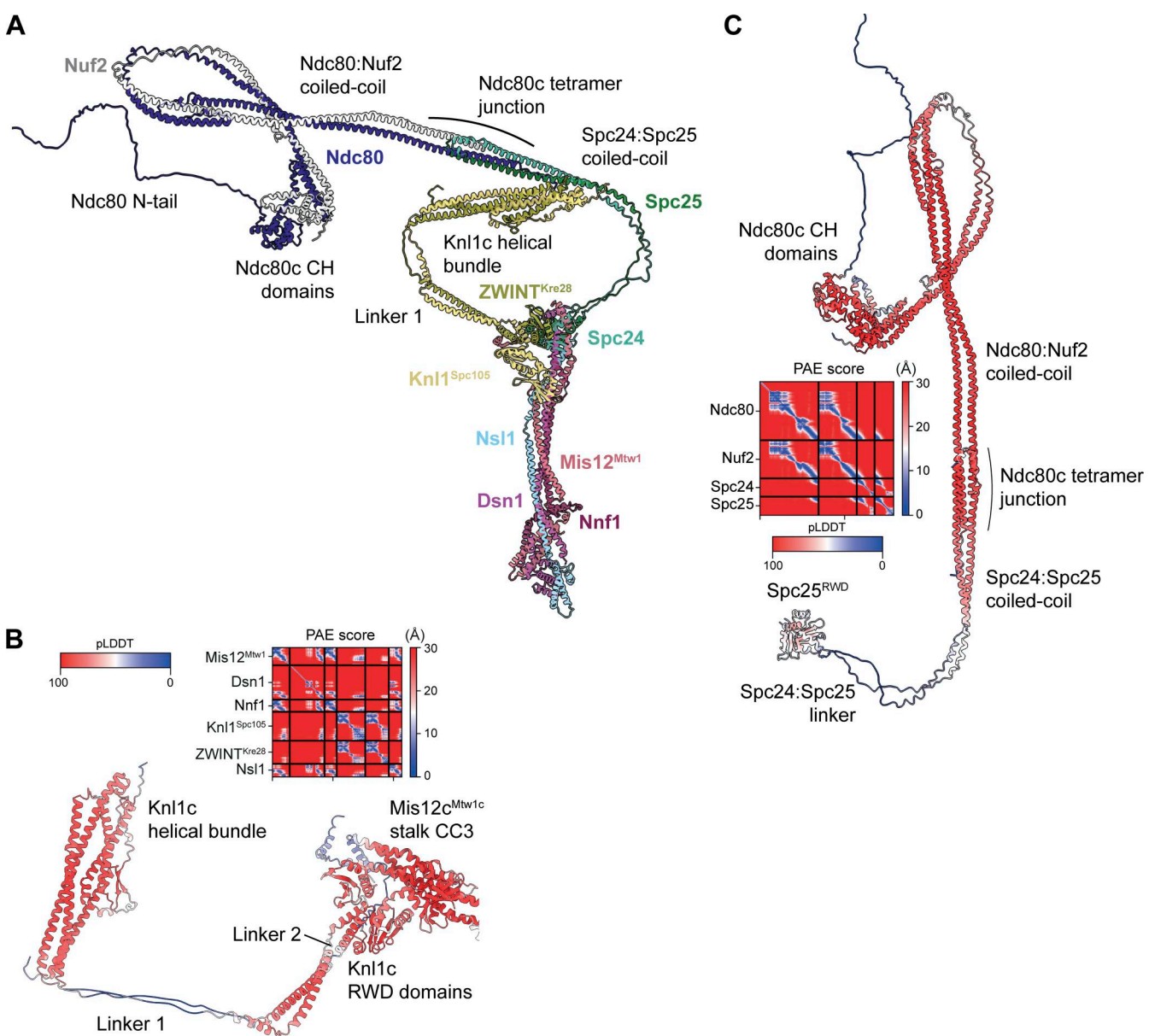

Figure 10. **Hypothetical model of the entire _S. cerevisiae_ KMN complex and comparison with humans. (A)** Model of the entire _S. cerevisiae_ KMN complex based on the structures determined experimentally in this study and AF2 models of Knl1c and Ndc80c. Flexibility of Ndc80c has been demonstrated (Scarborough et al., 2019; Wang et al., 2008), and flexibility of the Knl1c helical bundle is inferred from AF2 models. CH: calponin homology. N-tail: N-terminal tail. **(B)** AF2 structure prediction of Knl1c:Mis12c[Mtw1c] showing the Knl1c helical bundle domain, Knl1c RWD domains, and Mis12c[Mtw1c] stalk CC3 region. The structure was predicted using full-length, wild-type protein sequences colored by the residue pLDDT score with the PAE score plot shown above. PAE, predicted alignment error. **(C)** AF2 structure prediction of Ndc80c, colored by residue pLDDT score. The structure was predicted using full-length, wild-type protein sequences. The PAE score plot of the AF2 Ndc80c model is shown on the left of the panel.

pooled and further purified using a Superose 6 10/300 Increase GL column (Cytiva) equilibrated in KMN buffer (20 mM HEPES, pH 7.5, 150 mM NaCl, 1 mM TCEP). Peak Ndc80c fractions were pooled, concentrated, and flash-frozen using liquid nitrogen.

Mis12c[Mtw1c] complexes and Knl1c:Mis12c[Mtw1c] complexes were further purified by concentrating the Strep-Tactin eluate and applying it to a Superose 6 Increase 10/300 Gl column equilibrated in KMN buffer. The peak protein complex fractions were concentrated and flash-frozen using liquid nitrogen.

CENP-C[Mif2_1-63] with a N-terminal 6xHis-SUMO-tag was expressed in _E. coli_ BL-21 CodonPlus-RIL cells by induction with

0.5 mM IPTG at a cell culture $OD_{600nm}$ = 0.8 followed by incubation of cells for a further 18 h at 18°C with shaking. Cells were harvested by centrifugation and lysed by sonication in lysis buffer (30 mM Tris.HCl, pH 8.0, 300 mM NaCl, 1 mM EDTA, 1 mM TCEP, 0.2 mM PMSF) containing Benzonase and 10 mM imidazole. The lysate was cleared by ultracentrifugation at 38,000×$g$ in a JA25.50 rotor for 1 h, and the clarified lysate was applied to a HisTrap Excel column (Cytiva). Immobilized proteins were washed with lysis buffer and were eluted by successive applications of lysis buffer containing 50, 100, 150, 300, and 500 mM imidazole. The eluates obtained after application of

150 and 300 mM imidazole were pooled and applied to HiPrep 26/10 desalting columns (Cytiva) that were equilibrated in buffer containing 20 mM HEPES, pH 7.5, 300 mM NaCl, 1 mM EDTA, 1 mM TCEP to reduce the imidazole content.

The 6xHis-SUMO tag was removed from CENP-C$^{Mif2\_1-63}$ by incubation for 16 h with 0.279 mg of Ulp1 protease carrying a hexa-His tag at 4°C. Cleaved protein was isolated from the protease by a reverse IMAC purification step. After diluting the salt concentration to 100 mM, the reverse IMAC flow-through was applied to a Resource S cation exchange column (Cytiva) for further purification. CENP-C$^{Mif2\_1-63}$ was eluted using buffer containing 20 mM HEPES, pH 7.5, 300 mM NaCl, 1 mM TCEP. CENP-C$^{Mif2\_1-63}$ was concentrated and further purified by application to a Superdex 75 16/60 column (Cytiva) equilibrated in KMN buffer. Peak CENP-C$^{Mif2\_1-63}$ fractions were pooled, concentrated, and flash-frozen using liquid nitrogen.

The recombinant hexa-His-tag-Ulp1 protease (residues 403–621) was expressed in *E. coli* Novagen TUNER(DE3) cells (Thermo Fisher Scientific) from a pET30a plasmid (gift from George Ghanim and Kelly Nguyen) and subsequently purified on a HisTrap HP 5-ml column (Cytiva). The protein was applied to the column in buffer containing 25 mM HEPES, pH 8.0, 350 mM NaCl, 50 mM imidazole, 2 mM β-mercaptoethanol, 1 mM PMSF, 20% sucrose, and 0.1% NP-40. The column was washed with buffer containing 25 mM HEPES, pH 8.0, 1,000 mM NaCl, 50 mM imidazole, 2 mM β-mercaptoethanol, and 1 mM PMSF. The protein was then eluted in a buffer containing 25 mM HEPES, pH 8.0, 350 mM NaCl, 2 mM β-mercaptoethanol, and 1 mM PMSF with a 50- to 500-mM imidazole gradient over 10 column volumes. Protein-containing fractions identified by SDS-PAGE were pooled and dialyzed overnight into buffer containing 25 mM HEPES, pH 8, 100 mM NaCl, 0.5 mM EDTA, 1 mM DTT, and 0.5 mM PMSF. The dialyzed protein was further purified using a Poros 50 HE heparin column (Thermo Fisher Scientific). The bound protein was eluted in a buffer containing 25 mM HEPES, pH 8.0, 0.5 mM EDTA, and 1 mM DTT using a 100- to 1,000-mM NaCl gradient over 20 column volumes. The purified protein was supplemented with 50% glycerol (vol/vol) and stored at –80°C.

CENP-QU was expressed and purified according to the method described in Dendooven et al. (2023).

### Reconstitution and GraFix crosslinking the *S. cerevisiae* KMN complex for cryo-EM

The *S. cerevisiae* KMN complex was reconstituted and crosslinked for cryo-EM using a GraFix protocol (Stark, 2010) (Kastner et al., 2008). Two buffers were used to prepare 10–30% continuous glycerol gradients (4 ml total volume) with an additional 0.0–0.2% glutaraldehyde component using a gradient master (Biocomp Instruments): buffer A contained 20 mM HEPES, pH 7.5, 150 mM NaCl, 10% glycerol, 1 mM TCEP, and buffer B contained 20 mM HEPES, pH 7.5, 150 mM NaCl, 30% glycerol, 1 mM TCEP, 0.2% glutaraldehyde. Stocks of Ndc80c and a Knl1c:Mis12c$^{Mtw1c}$ complex that contained Knl1$^{Spc105\_HB-RWD}$ (K$^{HB-RWD}$M) were thawed on ice, and diluted and mixed at a 4 μM equimolar (1:1) ratio in KMN buffer. The mixture was incubated for 30 min at 4°C to reconstitute the KMN complex.

200 pmol (50 μl volume) of reconstituted K$^{HB-RWD}$MN complex was applied to each glycerol gradient, and gradients were subsequently subjected to centrifugation at 336,840×*g* in SW 60 Ti Swinging-Bucket Rotor (Beckman Coulter) for 16 h. Glycerol gradients were manually fractionated into 250 μl fractions, and unreacted glutaraldehyde was quenched by pipetting the fractions over a solution of 1 M Tris.HCl, pH 7.5, such that the final concentration of Tris.HCl in each fraction was 20 mM. Fractions encompassing the gradient volume 2.00–2.75 ml were pooled, glycerol content was reduced by application to Zeba spin desalting columns equilibrated in KMN buffer, and the protein complex was concentrated to a final concentration of 0.8 mg/ml.

### Cryo-EM grid preparation

Cryo-EM grids were glow-discharged in an Edwards S150B glow discharger for 60 s at setting 6, 20–25 Ma, 1.2 kV, and 0.2 mBar (0.15 Torr) immediately prior to application of the sample. 3 μl of reconstituted and crosslinked K$^{HB-RWD}$MN complex was applied to freshly glow-discharged UltrAuFoil 300 mesh gold R1.2/1.3 grids (Quantifoil Micro Tools). Grids were mounted in a FEI Vitrobot Mark III humidity chamber (maintained at 4°C, 100% humidity), blotted with Whatman filter paper I (2 s blotting time, –7 blotting force), and vitrified in liquid ethane.

### Cryo-EM data acquisition

Cryo-EM movies were collected on a Thermo Fisher Scientific Titan Krios 1 microscope operating at 300 keV using a Gatan K3 camera equipped with a Gatan energy filter. Movies were collected at ×105,000 magnification (pixel size 0.825 Å px$^{-1}$) with a 1.75-s exposure and 15.6 e$^-$/px/s fluence, providing a total dose of 40 e$^-$/Å$^2$, and were split into 40 frames. A series of movies were collected over a range of defoci from –1.4 μm to –3.0 μm, at an interval of 0.4 μm. Data acquisition was performed using aberration-free image shifts as employed in the Thermo Fisher Scientific "EPU" automated data acquisition software.

### Cryo-EM data processing

Alignment of micrograph raw movie frames and correction to a single micrograph image were performed using UCSF MotionCor2 as implemented in RELION (Kimanius et al., 2021; Zheng et al., 2017). The contrast transfer function of each motion-corrected micrograph was estimated using CTFFIND-4.1 as implemented in RELION (Kimanius et al., 2021; Rohou and Grigorieff, 2015). Manual particle picking of 10 micrographs was used to train a Topaz model that was used to pick the entire dataset of 26,390 micrographs (Bepler et al., 2019) (Fig. S2 and Table S1). The particles picked by the first Topaz model were curated by two successive rounds of 2D classification in CryoSPARC (Punjani et al., 2017). 2D classification in CryoSPARC was always performed using batch sizes of 400 and 40 online-em iterations. Particles classified into class averages that resembled KMN were selected and used to train a second Topaz model in RELION that was once again used to pick all 26,390 micrographs.

Particles were separated into six distinct sets, based on the appearance of 2D class averages, using two successive rounds of 2D classification in CryoSPARC. Each set of particles was used to

generate an *ab initio* model in CryoSPARC, yielding one KMN model, one additional protein model, and four decoy noise models. All six *ab initio* models were used as references for a noise decoy classification step using the heterogeneous refinement job type in CryoSPARC. Particles corresponding to the model that resembled the KMN complex were subjected to two further successive rounds of 2D classification, yielding 321,078 particles that corresponded to 2D class averages that resembled KMN. This set of particles was further processed in RELION for all subsequent steps. All 3D refinement and 3D classification jobs in RELION used Blush regularization, as implemented in RELION 5.0 (Kimanius et al., 2021; Kimanius et al., 2024).

The particles were used to generate an *ab initio* model, against which they were refined (successive unmasked and masked refinements) to reconstruct a cryo-EM density map with a resolution of 7.5 Å and discontinuous density for the Spc24 and Spc25 RWD domains and the $Mis12c^{Mtw1c}$ head 2 domain.

To better resolve the Spc24 and Spc25 RWD domains, particles were subjected to focused 3D classification without alignment using a mask encompassing the CC3 region of the $Mis12c^{Mtw1c}$ stalk domain, the Knl1c RWD domains, and the Spc24 and Spc25 RWD domains, and a *T* value of 25. A set of 113,688 particles corresponding to two 3D classes with improved continuity of the Spc24 and Spc25 RWD domain density were refined to reconstruct a cryo-EM density map with a resolution of 6.5 Å and discontinuous density for the $Mis12c^{Mtw1c}$ head 2 domain. Signal corresponding to the $Mis12c^{Mtw1c}$ head 2 domain was subtracted from the particle images, and the subtracted particles were refined to reconstruct a cryo-EM density map with a resolution of 6.0 Å. Two nonoverlapping masks were generated for multibody refinement: one encompassed part of the CC2 region of the $Mis12c^{Mtw1c}$ stalk domain, the CC3 region of the $Mis12c^{Mtw1c}$ stalk domain, the Knl1c RWD domains, and the Spc24 and Spc25 RWD domains ("apex mask 1"), while the other encompassed the $Mis12c^{Mtw1c}$ head 1 domains, the CC1 region of the $Mis12c^{Mtw1c}$ stalk domain, and the remaining segment of the CC2 region of the $Mis12c^{Mtw1c}$ stalk domain ("base mask 1"). Multibody refinement using these masks yielded two cryo-EM density maps with global resolutions (GS-FSC) of 4.8 Å ("apex body 1") and 4.9 Å ("base body 1"), respectively. The local resolutions of the reconstructions reached 4.2 Å and 4.5 Å in the central regions of the apex and base density maps, respectively (Fig. S2, E and F), which were sufficient to interpret with an AF2-predicted model (Jumper et al., 2021). To obtain a composite map of the KMN junction complex, the cryo-EM maps of the apex and base multibodies were docked into the consensus cryo-EM map (Fig. 1 B). This composite map was used to build a model of the *S. cerevisiae* KMN junction complex using AF2 structure predictions and Isolde model refinement (Croll, 2018; Jumper et al., 2021) as detailed below in "Cryo-EM molecular model building."

To better resolve the $Mis12c^{Mtw1c}$ head 2 domain, particles were subjected to focused 3D classification without alignment using a mask encompassing the $Mis12c^{Mtw1c}$ head 1 and head 2 domains and the CC1 region of the $Mis12c^{Mtw1c}$ stalk domain (Fig. S4 and Table S1). A set of 18,160 particles corresponding to a 3D class with continuous density for the $Mis12c^{Mtw1c}$ head 2 domain were refined to reconstruct a cryo-EM density map with a

resolution of 7.5 Å that retained discontinuous density for the Spc24 and Spc25 RWD domains. Signal corresponding to the Spc24 and Spc25 RWD domains was subtracted from the particle images, and the subtracted particle images were used to reconstruct a cryo-EM density map with a resolution of 7.2 Å. Two nonoverlapping masks were generated for multibody refinement: one encompassed part of the CC2 region of the $Mis12c^{Mtw1c}$ stalk domain, the CC3 region of the $Mis12c^{Mtw1c}$ stalk domain, and the Knl1c RWD domains ("apex mask 2"), while the other encompassed the $Mis12c^{Mtw1c}$ head 1 and head 2 domains, the CC1 region of the $Mis12c^{Mtw1c}$ stalk domain, and the remaining segment of the CC2 region of the $Mis12c^{Mtw1c}$ stalk domain ("base mask 2"). Multibody refinement using these masks reconstructed a cryo-EM density map for the $Mis12c^{Mtw1c}$ head 1 and head 2 domains and relevant sections of the stalk domain with a global resolution (GS-FSC) of 6.5 Å ("base body 2").

**Cryo-EM molecular model building**

AF2 multimer structure predictions of $Mis12c^{Mtw1c}$:Spc24:Spc25 and Knl1c:$Mis12c^{Mtw1c}$, performed using full-length, wild-type proteins apart from $Knl1^{Spc105}$ (from which residues 1–444 were excluded) and Dsn1 (from which residues 1–278 were excluded), were used to build a model for the KMN junction complex. A composite map of the KMN junction complex was obtained by docking apex body 1 and base body 1 into the cryo-EM density map reconstructed by the preceding refinement and used for model building.

The AF2 structure predictions were divided into five regions for model building. "Region one," corresponding to Spc24 and Spc25 RWD domains interacting with $Mis12^{Mtw1\_Spc24-helix}$ and $Dsn1^{Cterm-helix}$ helical motifs, comprised $Spc24^{143-213}$, $Spc25^{124-221}$, $Mis12^{Mtw1\_228-252}$, and $Dsn1^{556-576}$. "Region two," corresponding to the RWD domains of $ZWINT^{Kre28}$ and $Knl1^{Spc105}$ interacting with the $Mis12^{Mtw1\_Cterm-helix}$ and $Nnf1^{Cterm-helix}$ helical motifs, comprised $Knl1^{Spc105\_719-917}$, $ZWINT^{Kre28\_280-368}$, $Mis12^{Mtw1\_274-289}$, and $Nnf1^{184-201}$. "Region three," corresponding to all of the CC3 and most of the CC2 regions of the $Mis12c^{Mtw1c}$ stalk domain, comprised $Dsn1^{483-547}$, $Nnf1^{139-180}$, $Nsl1^{143-216}$, and $Mis12^{Mtw1\_132-222}$. "Region four," corresponding to the remainder of the CC2 and all of the CC1 regions of the $Mis12c^{Mtw1c}$ stalk domains, comprised $Dsn1^{431-482}$, $Nnf1^{115-138}$, $Nsl1^{97-142}$, and $Mis12^{Mtw1\_116-131}$. "Region five," comprising the $Mis12c^{Mtw1c}$ head 1 domain, comprised $Nnf1^{1-114}$ and $Mis12^{Mtw1\_1-115}$. Isolde, as implemented in ChimeraX, was used to remodel and refine these five regions of the AF2 structure predictions into the composite map of the KMN junction complex (Croll, 2018; Goddard et al., 2018).

C-terminal residues of Nnf1, $Mis12^{Mtw1}$, Dsn1, and Nsl1 in region five and region four were manually reconnected with the N-terminal residue of the corresponding chain in region four and region three, respectively, and refined into the density in Coot (Emsley et al., 2010) to yield "region tff." The model comprising region one, region two, and region tff was subsequently refined and remodeled into the composite map using Isolde to yield the final model of the KMN junction complex.

To build the experimental model of the *S. cerevisiae* KMN junction complex with head 2, we docked the KMN junction complex (without head 2) and the AF2 prediction of the *S. cerevisiae*

Mtw1[Mis12] head 2 domain (Dsn1 residues 259–353 and Nsl1 residues 1–84), and the AF2 prediction of *Kl*Mtw1[Mis12c], into the 6.5 Å KMN junction base body 2 cryo-EM map (Fig. S4 C and Table S1). The *Kl*Mtw1[Mis12c] model accounted for nearly all density in the cryo-EM map, whereas for the *S. cerevisiae* model of head 2, some density regions were not assigned, mainly a helical-like density associated with Dsn1 head 1 (Dsn1[AI], *Sc* residues 229–255), the Dsn1 CC1–head 2 linker (*Sc* residues 354–365), and the Nsl1 CC1–head 2 linker (*Sc* residues 85–96) (Fig. S5, A, B, and E). We used the *Kl*Mtw1[Mis12c] model (Fig. S5, G and H) to guide complete building of an experimental model of *Sc*Mtw1[Mis12c] (Fig. 6, B and C).

### Model refinement
Refinement was performed using the Phenix package (Liebschner et al., 2019) (Table S1).

### AF2 structure predictions
Local installations of AF2 (Jumper et al., 2021) were used for all protein structure predictions in this study. The Colabfold implementation (Mirdita et al., 2022) of AF2 was used for all structure predictions other than for the *K. lactis* Mis12c[Mtw1c] (*Kl*Mis12c[Mtw1c]), which used the Colabfold 3 version. Unless otherwise stated, wild-type, full-length protein sequences deposited in UniProt were used for all protein structure predictions.

### ITC experiments
Dsn1[228–254] and Dsn1[228–254,S240E,S250E] peptides were synthesized by Alta BioScience with N-terminal acetylation and C-terminal GSAW extension and amidation. The final sequence of Dsn1[228–254] was PHYIQRTRERKKSIGSQRGRRLSMLASGSAW, and Dsn1[228–254,S240E,S250E] was PHYIQRTRERKKEIGSQRGRRLEMLASGSAW. All protein complexes and peptides used in a given experiment were dialyzed for a minimum of 16 h and/or buffer-exchanged by size-exclusion chromatography into the same batch of ITC buffer (25 mM HEPES, pH 7.5, 100 mM NaCl, 1 mM TCEP). All experiments were performed using an Auto-iTC200 instrument (Malvern Instruments) at 25°C. In each experiment, 360 µl of protein complex was pipetted into the calorimeter cell. The protein complex loaded into the calorimeter cell contained Knl1c:Mis12c[Mtw1c] with Knl1[Spc105_HB-RWD] and Dsn1[Δ1–257] and either wild-type Mis12[Mtw1] (K[HB-RWD]M[Dsn1Δ257]) or Mis12[Mtw1_E69A,E73A,E77A] mutant (K[HB-RWD]M[Dsn1Δ257_Mis12-3A]). Wild-type constructs of all other Knl1c and Mis12c[Mtw1c] subunits were included. Titration of either the Dsn1[228–254] or Dsn1[228–254,S240E,S250E] peptide into the cell was then performed by an initial injection of 0.5 µl, followed by 19 further injections of 2.0 µl each. Heat arising from the initial injection was discarded, and the MicroCal PEAQ-ITC analysis software 1.0.0.1258 (Malvern Instruments) was subsequently used to fit the change in the amount of heat released, integrated over the entire titration, to a single-site–binding model. Concentrations of the protein complex and peptide used for each experiment are provided in the relevant figure panel. Titrations were performed in duplicate or triplicate, and reported $K_D$ and $n$ values were calculated as an average from all replicate results.

CENP-U[Amel_1-33] (MDRDTKLAFRLRGSHSRRTDDIDDDVIVFK TPNW-amide), CENP-U[Amel_1-33-Mut] (MDRDTKLAFRLRGSHSR RTDDIDDAAAVAKAPNW-amide), and CENP-C[Mif2_1-38] (MDYM KLGLKSRKTGIDVKQDIPKDEYSMENIDDFFKDD-amide) peptides were synthesized by Alta BioScience. Peptides and proteins (K[HB-RWD]M and K[HB-RWD]M[2E]) were dialyzed for 16 h at 4°C against a buffer of 25 mM HEPES (pH 7.5), 100 mM NaCl, 0.5 mM TCEP. The ITC experiments were performed in triplicate as described above except at 20°C. Experiments were performed either two or three times (technical repeats).

### Investigation of the function of Mis12c[Mtw1c] motifs that form interfaces with Knl1c
To test the effect of deleting the Mis12c[Mtw1c] motifs that form interfaces with Knl1c on Mis12c[Mtw1c] assembly and interaction with Ndc80c, 4 µM concentration of wild-type Mis12c[Mtw1c] or one of the three Mtw1c mutants (Mis12c[Mtw1c_Mis12ΔC], contained Mis12[Mtw1_Δ272-C]; Mis12c[Mtw1c_Nnf1ΔC], contained Nnf1[Δ180-C]; Mis12c[Mtw1c_Mis12ΔC-Nnf1ΔC], contained Mis12[Mtw1_Δ272-C] and Nnf1[Δ180-C]) was either individually diluted into KMN buffer, or mixed with 4 µM concentration of wild-type Ndc80c in KMN buffer. Mixtures of Ndc80c and wild-type or mutant Mis12c[Mtw1c] were incubated on ice for 30 min to reconstitute the complex. 100 µl of individual protein complexes or reconstitution mixtures was loaded onto a Superose 6 3.2/300 size-exclusion chromatography column (Cytiva), and 100 µl eluate fractions were collected. Aliquots of each eluate fraction were mixed 1:1 (vol/vol) with 2xLDS-PAGE loading buffer (Thermo Fisher Scientific), supplemented with DTT, boiled for 3 min at 95°C, and analyzed by SDS-PAGE.

To test the effect of deleting the Mis12c[Mtw1c] motifs that form interfaces with Knl1c on the interaction between Knl1c and Mis12c[Mtw1c], suspension cultures of High-5 insect cells were infected with one of two P3 suspension cultures, or two P3 cultures in combination, that were amplifying one of two sets of viruses. The first virus encoded ZWINT[Kre28_RWD] fused to a C-terminal TS and Knl1[Spc105_RWD] fused to an N-terminal tag comprising tandem MBP and mEGFP tags. The second set of viruses encoded either wild-type Mis12c[Mtw1c], or one of the Mis12c[Mtw1c_Mis12ΔC] or Mis12c[Mtw1c_Nnf1ΔC] mutants. All viruses encoding Mis12c[Mtw1c] contained genes that expressed Dsn1[Δ1–278]. The insect cell cultures were harvested 48 h after infection, when cell viability had dropped to 70–75%.

All subsequent steps were performed at 4°C or on ice. Cells were lysed by sonication in a lysis buffer containing 50 mM HEPES, pH 8.5, 200 mM NaCl, 8 mM benzamidine, 7.5% glycerol, 1 mM TCEP, 0.5 mM EDTA, 0.2 mM PMSF that was supplemented with cOmplete EDTA-free protease inhibitor tablets and Benzonase. The lysate was cleared by ultracentrifugation at 38,000×*g* for one h in a JA-25.50 rotor. A sample of the cleared supernatant was withdrawn and mixed 1:9 (volume/volume) with lysis buffer, mixed with 10 volume equivalents of 2xLDS-PAGE loading buffer (Thermo Fisher Scientific) supplemented with DTT, and boiled for 5 min at 95°C for analysis as an input sample by SDS-PAGE. Cleared supernatants were subsequently filtered and incubated while rolling with 0.5 ml of a 50% slurry of Strep-Tactin Superflow plus resin (Qiagen) in a wash buffer

containing 50 mM HEPES, pH 8.0, 200 mM NaCl, 5% glycerol, 0.5 mM EDTA, 1 mM TCEP for 90 min. Resin was collected by centrifugation at 500×$g$ for 3 min and washed with 10 ml of wash buffer. Resin washes were repeated, as above, a further four times. Resin was collected a final time by centrifugation, as above, and bound complexes were eluted with wash buffer containing 2.5 mM d-desthiobiotin. The resin was collected by centrifugation, as above, and 10 µl of eluate was withdrawn and mixed with an equal volume of 2xLDS-PAGE loading buffer (Thermo Fisher Scientific) that was supplemented with DTT. This sample of eluate was then boiled for 5 min and analyzed by SDS-PAGE.

### Investigation of Dsn1 N-terminal intrinsically disordered domain function in Mis12c$^{Mtw1c}$ auto-inhibition

To test the effect of deleting or mutating regions of the Dsn1 N-terminal disordered domain on Mis12c$^{Mtw1c}$ auto-inhibition, 4 µM concentrations of Knl1c:Mis12c$^{Mtw1c}$ complexes containing Knl1$^{Spc105}$_HB-RWD and different Dsn1 constructs were mixed with 4 µM concentrations of either CENP-C$^{Mif2}$_1-63 or CENP-QU in KMN buffer. Knl1c:Mis12c$^{Mtw1c}$ complexes used contained one of the following Dsn1 constructs: K$^{HB-RWD}$M, contained full-length and wild-type Dsn1; K$^{HB-RWD}$M$^{S240E,S250E}$, contained Dsn1$^{S240E,250E}$; K$^{HB-RWD}$M$^{Dsn1Δ226}$, contained Dsn1$^{Δ1-226}$; K$^{HB-RWD}$M$^{Dsn1Δ257}$, contained Dsn1$^{Δ1-257}$; K$^{HB-RWD}$M$^{Dsn1Δ226,S240E,S250E}$, contained Dsn1$^{Δ1-226,S240E,S250E}$. Protein mixtures were incubated together on ice for 30 min to attempt to reconstitute a protein complex. 100 µl of individual proteins or protein complexes or reconstitution mixtures was loaded onto a Superose 6 3.2/300 size-exclusion chromatography column (Cytiva) equilibrated in (20 mM HEPES, pH 7.5, 150 mM NaCl, 1 mM TCEP), and 100 µl eluate fractions were collected. Aliquots of each indicated eluate fraction were mixed 1:1 (volume/volume) with 2xLDS-PAGE loading buffer supplied by Thermo Fisher Scientific, supplemented with DTT, boiled for 3 min at 95°C, and analyzed by SDS-PAGE.

### Yeast strain construction

We constructed *S. cerevisiae* strains encoding the *Os*Tir1 ubiquitin ligase at the *URA3* locus and a mAID$_3$-FLAG$_5$ C-terminal tag at either the endogenous *NNF1* (encoding Nnf1) or *MTW1* (encoding *Mis12$^{Mtw1}$*) loci. Yeast strains constructed for this study are listed in Table S2. To construct strains encoding Mis12$^{Mtw1}$-mAID$_3$-FLAG$_5$ (*MTW1-mAID$_3$-FLAG$_5$* allele) or Nnf1-mAID$_3$-FLAG$_5$ (*NNF1-mAID$_3$FLAG$_5$* allele), a degron cassette encoding mAID$_3$-FLAG$_5$ and a KanMX gene that provided resistance to G418 were amplified from plasmid pST1933 (Tanaka et al., 2015) through a PCR that used Q5 polymerase and oligonucleotide primers containing 48-bp 5′ extensions with homology to the relevant locus in the yeast genome. BY26972 strain yeast cells (Table S2) from an exponentially growing suspension culture were transformed with the cassette according to the previously detailed LiAc protocol (Gietz and Schiestl, 2007). In brief, cells were harvested by centrifugation, washed thrice with sterile Millipore water (MPW), and incubated with 360 µl transformation buffer (100 mM lithium acetate, 33% PEG 3350, 100 µg salmon sperm DNA) containing the degron cassette. Cells were transformed with the cassette through a 45 min heat shock at 42°C, and allowed to recover by standing in 1 ml of YEPD medium for 3 h at 22°C. Transformants were then selected by plating yeast cells on YEPD agar containing 500 µg/ml G418 (A1720-5G; Sigma-Aldrich) and grown at 30°C for 3–4 days. Individual colonies were picked and restreaked on fresh YEPD agar containing 500 µg/ml G418 twice, successively. Integration of the cassette was then screened by PCR amplification using Q5 polymerase and two primers directed against the 3′ end of the relevant gene and an internal region of the degron cassette. PCR products obtained by this method were sequenced to validate in-frame integration of the DNA sequences encoding the mAID$_3$-FLAG$_5$ tags at the 3′ end of the relevant gene.

Strains expressing Mis12$^{Mtw1}$-mAID$_3$-FLAG$_5$ or Nnf1-mAID$_3$-FLAG$_5$ and an allele to be tested for its ability to rescue cell death were constructed as follows. Yeast strains expressing Mis12$^{Mtw1}$-mAID$_3$-FLAG$_5$ or Nnf1$^1$-mAID$_3$-FLAG$_5$ were transformed, as above, with pRS304 (encoding *MTW1* alleles) or pRS305 (encoding *NNF1* alleles) plasmids that were cleaved with a restriction enzyme at a single site within the *LEU2* (pRS304 plasmids) or *TRP1* (pRS305 plasmids) selection marker. The plasmids contained the following alleles encoding the listed gene products: *MTW1*, full-length and wild-type Mis12$^{Mtw1}$; *mtw1$^{ΔC}$*, Mis12$^{Mtw1\_Δ272-C}$, *NNF1*, full-length and wild-type Nnf1; *nnf1$^{ΔC}$*, Nnf1$^{Δ180-C}$. Transformants were selected for by plating yeast cells on leucine dropout (*MTW1* or *mtw1$^{ΔC}$* transformants) or tryptophan dropout (*NNF1* or *nnf1$^{ΔC}$* transformants) agar media containing 500 µg/ml G418. Integration of the cassette at the *LEU2* or *TRP1* locus was screened using two PCRs that amplified across the 5′ and -3′ recombination junctions. The first PCR used a pair of oligonucleotide primers targeted against a region of the genome 5′ of *LEU2* or *TRP1* and against an internal region of the integrated cassette, and the second PCR used a pair of primers targeted against a region of the genome 3′ of *LEU2* or *TRP1* and against another internal region of the cassette. Positive transformants were stored in 50% glycerol at –80°C for later use.

All strains grew well on YEPD plates in the absence of IAA, showing that the incorporation of the mAID$_3$-FLAG$_5$ tag at the endogenous MTW1 and NNF1 loci, and rescue alleles at the exogenous loci, did not adversely affect cell growth (Fig. S3 E).

### Auxin depletion assays

10 ml suspension cultures of each yeast strain were grown overnight at 30°C. The optical density at 600 nm of each suspension culture was measured in triplicate; yeast cells were subsequently diluted to OD$_{600nm}$ = 0.1, and then serially diluted 1:9 (vol/vol) over 4 further dilutions. 4 µl of each dilution was then applied to agar plates containing either YEPD medium, or YEPD medium and 0.5 mM IAA. Cells were grown at 30°C for 3 days and imaged.

### Western blotting

To validate expression levels of the gene products of the *MTW1* (Mis12$^{Mtw1}$) and *NNF1* (Nnf1) alleles integrated at ectopic sites, 40 ml of each cell culture was grown in YEPD until the culture reached an OD$_{600nm}$ between 0.7 and 2.0. At this point, 1 × 10$^7$ (*MTW1-mAID$_3$-FLAG$_5$* strains) or 1.4 × 10$^7$ (*NNF1-mAID$_3$-FLAG$_5$*

strains) cells were withdrawn from the culture and processed for western blotting as described below.

Depletion of $MTW1$-$mAID_3$-$FLAG_5$ and $NNF1$-$mAID_3$-$FLAG_5$ gene products by the addition of IAA was monitored and validated by growing 40 ml suspension cultures of each strain in YEPD medium until the culture attained an $OD_{600nm}$ of between 0.7 and 2.0. This was followed by the addition of either IAA to a final concentration of 0.5 mM, or an equal volume of phosphate-buffered saline as a negative control. A volume of suspension culture containing $1 \times 10^7$ $MTW1$-$mAID_3$-$FLAG_5$ strains or $1.4 \times 10^7$ $NNF1$-$mAID_3$-$FLAG_5$ cells was then withdrawn from each culture at the indicated time points and processed for western blotting as described below.

All subsequent steps were performed at 22°C, unless otherwise indicated. Yeast cells were collected by centrifugation at 3,000×$g$ for 3 min, resuspended in 1 ml of MPW, and pelleted by centrifugation at 12,000×$g$ for 1 min. Whole-cell extracts were then prepared by incubation of the cell pellets in 50 µl of 0.2 M NaOH for 5 min, centrifugation for 1 min at 12,000×$g$, a wash of the cell pellet with 1 ml of MPW, and a final collection of the pellet by centrifugation as above. Pellets were then boiled for 3 min at 95°C in 50 µl of 1xLDS-PAGE loading buffer supplemented with DTT. The pellet was collected by centrifugation, and 10 µl of each supernatant was applied to an SDS-PAGE gel to separate proteins through application of a constant 210 V for 45 min. Proteins were transferred from the SDS-PAGE gel onto a Trans-Blot Turbo minigel format (Bio-Rad) 0.2 µm nitrocellulose membranes using a Trans-Blot Turbo transfer system. Membranes were blocked with 3% milk suspension ($NNF1$ gene product blots) or 5% milk suspension ($MTW1$ gene product blots) in phosphate-buffered saline containing 0.2% Tween-20 (PBS-T) for 30 min at 22°C. Membranes were then probed with the primary antibody detailed below at 4°C for 16 h in PBS-T to detect the protein of interest. To detect $MTW1$-$mAID_3$-$FLAG_5$ and $NNF1$-$mAID_3$-$FLAG_5$ gene products, a 1:1,000 (volume/volume) dilution of anti-FLAG antibody (Cell Signaling, Cat#D6W5B, rabbit monoclonal) was used. To detect $NNF1$-$V5_3$ and $nnf1^{\Delta C}$-$V5_3$ or $MTW1$-$HA_3$ and $mtw1^{\Delta C}$ expressed from the ectopically integrated loci, 1:3,333 dilutions of anti-V5 (Proteintech, Cat#14440-1-AP, rabbit) or anti-HA (Abcam, Cat#ab1424, mouse monoclonal) antibodies, respectively, were used. To detect alpha tubulin as a loading control, anti-α-tubulin (MCA78G; Bio-Rad, rat monoclonal) antibody was used.

Membranes were washed four times with PBS-T before incubation with a 1:3,333 (vol/vol) dilution of HRP-conjugated secondary antibody (donkey anti-rabbit, HRP-conjugated [Cat#SA1-200; Thermo Fisher Scientific], goat anti-rat IgG-HRP (Cat#2032; Santa Cruz Biotechnology), and sheep anti-mouse, HRP-conjugated [Cat#NXA931; Cytiva]) in PBS-T by rolling at 21°C for 1.5–2.5 h. Membranes were washed a further four times with PBS-T before visualization of the secondary antibody using chemiluminescence using the ECL Prime western blotting detection kit (Amersham) and imaging using a ChemiDoc imaging system (Bio-Rad).

## Multiple sequence alignment
Multiple sequence alignments were performed with Clustal Omega using default settings and full-length, wild-type protein

sequences deposited in UniProt unless otherwise indicated (Sievers et al., 2011). Sequence alignments were visualized using the Jalview workbench using a conservation score of 10 (Waterhouse et al., 2009).

## Crosslinking mass spectrometry
Protein crosslinking reactions on the *S. cerevisiae* K$^{HB-RWD}$M complex (100 µl at 0.5 mg/ml) were carried out with sulfo-SDA (2 mM concentration). Sulfo-SDA and the protein complex were mixed and incubated on ice for 5 min before being crosslinked for 10 s with 365-nm UV radiation from a home-built UV LED setup. Crosslinking reactions were quenched with the addition of ammonium bicarbonate to a final concentration of 50 mM. The quenched solution was reduced with 5 mM DTT and alkylated with 20 mM iodoacetamide. The SP3 protocol, as described (Batth et al., 2019; Hughes et al., 2019), was used to clean up and buffer-exchange the reduced and alkylated protein. In brief, for the SP3 protocol, proteins are washed with ethanol using magnetic beads for protein capture and binding. The proteins were resuspended in 100 mM $NH_4HCO_3$ and were digested with trypsin (Promega) at an enzyme-to-substrate ratio of 1:20, and protease max 0.1% (Promega). Digestion was carried out at 37°C for 16 h. Clean-up of peptide digests was carried out with HyperSep SpinTip P-20 (Thermo Fisher Scientific) C18 columns, using 60% acetonitrile as the elution solvent. Peptides were then evaporated to dryness *via* Speed Vac Plus (Savant).

Dried peptides were resuspended in 30% acetonitrile and were fractionated *via* SEC using a Superdex 30 Increase 3.2/300 column (Cytiva) at a flow rate of 20 µl/min using 30% (vol/vol) ACN/0.1% (vol/vol) TFA as a mobile phase. Fractions are taken every 5 min, and the second to seventh fractions containing crosslinked peptides are collected. Dried peptides were suspended in 3% (vol/vol) acetonitrile and 0.1% (vol/vol) formic acid and analyzed by nano-scale capillary LC-MS/MS using an UltiMate U3000 HPLC to deliver a flow of 300 nl/min. Peptides were trapped on a C18 Acclaim PepMap100 5 µm, 0.3 µm × 5 mm cartridge (Thermo Fisher Scientific) before separation on Aurora Ultimate C18, 1.7 µm, 75 µm × 25 cm (ionOpticks). Peptides were eluted on optimized gradients of 90 min and interfaced *via* an Easy-Spray ionization source to a tribrid quadrupole Orbitrap mass spectrometer (Orbitrap Eclipse, Thermo Fisher Scientific) equipped with FAIMS. MS data were acquired in data-dependent mode with a Top-25 method; high-resolution full mass scans were carried out (R = 120,000, $m/z$ 400–1,550) followed by higher energy collision dissociation with stepped collision energy range 21, 30, 34% normalized collision energy. The tandem mass spectra were recorded (R = 60,000, isolation window $m/z$ 1, dynamic exclusion 50 s). Mass spectrometry measurements were cycled for 3-s durations between FAIMS CV -45, and –60 V.

## Crosslinking mass spectrometry data analysis
Xcalibur raw files were converted to MGF files using ProteoWizard, and crosslinks were analyzed by XiSearch (Chambers et al., 2012; Mendes et al., 2019). Search conditions used 3 maximum missed cleavages with a minimum peptide length of 5. Variable modifications used were carbamidomethylation of cysteine (57.02146 Da) and methionine oxidation (15.99491 Da).

False discovery rate was set to 5%. Crosslinking mass spectrometry data were visualized and analyzed using xiVIEW (Combe et al., 2024).

### Computational methods

The UCSF ChimeraX package was used to visualize cryo-EM density maps, AF2 structure predictions, deposited structures retrieved from the protein data bank, and crosslinking mass spectrometry data superimposed onto the structural models (Goddard et al., 2018; Pettersen et al., 2021).

### Online supplemental material

Fig. S1 shows preparation of *S. cerevisiae* KMN complex sample for Cryo-EM and validation of *in vivo* protein depletion and *in vivo* rescue allele expression. Fig. S2 shows cryo-EM data processing to reconstruct the KMN junction complex. Fig. S3 shows biochemical analysis of Mis12[Mtw1_Cterm-helix] and Nnf1[Cterm-helix] function. Fig. S4 shows cryo-EM data processing to reconstruct Mis12c[Mtw1c] head 2 domain. Fig. S5 shows modeling of contacts between Mis12cMtw1c head 1, head 2, and Dsn1[N-IDR] domains. Video 1 shows overview of the *S. cerevisiae* KMN junction complex. Video 2 shows predicted conformational change of Dsn1[AI] and head 2 accompanying *Sc*Mis12c[Mtw1c] activation by Dsn1[AI] phosphorylation.

## Data availability

Protein coordinates and cryo-EM maps were deposited at RCSB (https://www.rcsb.org/) and EMDB (https://www.ebi.ac.uk/emdb/), respectively (Table S1). The mass spectrometry proteomics data have been deposited to the ProteomeXchange Consortium *via* the PRIDE (Perez-Riverol et al., 2025) (https://www.ebi.ac.uk/pride/) partner repository with the dataset identifier PXD072662.

## Acknowledgments

We are grateful to the LMB EM Facility for help with the EM data collection; to J. Grimmett, T. Darling, and I. Clayson for scientific computing; to J. Shi and K. Turton for help with insect cell expression; and to T. Dendooven, D. Kimanius, and S. Yatskevich for discussions. We thank G.E. Ghanim and T.H.D. Nguyen for the Ulp1 protease expression vector. For the purpose of open access, the authors have applied a CC BY public copyright license to any Author Accepted Manuscript version arising.

This work was supported by UKRI/Medical Research Council MC_UP_1201/6 (to D. Barford), Cancer Research UK C576/A25675 (to D. Barford), Cancer Research UK C576/A14109 (to D. Barford), and CRUK PhD studentship grant (to N.N. Turner). Open Access funding provided by MRC Laboratory of Molecular Biology.

Author contributions: Noah N. Turner: conceptualization, data curation, formal analysis, investigation, methodology, validation, visualization, and writing—original draft. Ziguo Zhang: investigation and methodology. Jing Yang: investigation and resources. Kyle W. Muir: conceptualization, methodology, and supervision. Stephen H. McLaughlin: investigation. Tomos Morgan: investigation. David Barford: conceptualization, data curation, formal analysis, funding acquisition, investigation, methodology, project administration, resources, supervision, validation, visualization, and writing—original draft, review, and editing.

Disclosures: The authors declare no competing interests exist.

Submitted: 4 June 2025

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

# Supplemental material

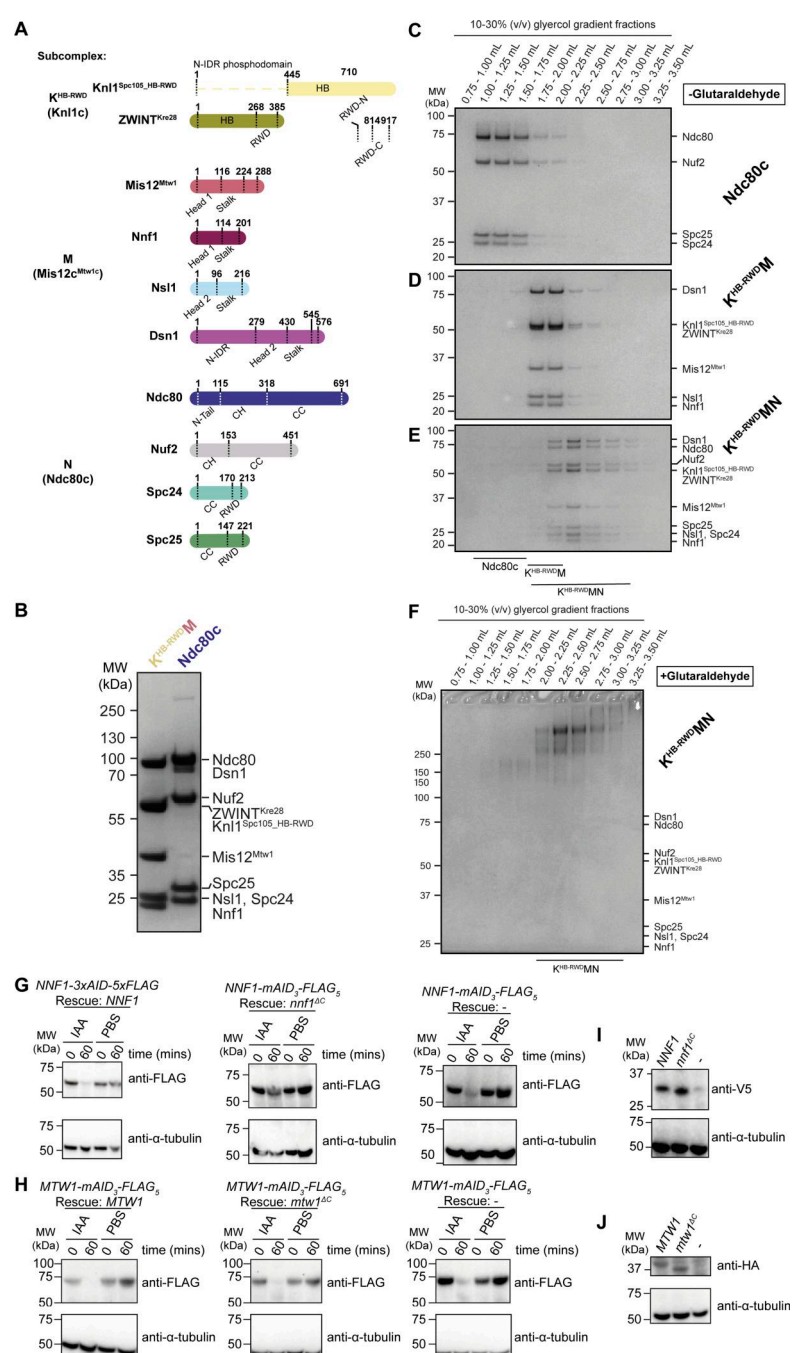

Figure S1. **Preparation of *S. cerevisiae* KMN complex sample for cryo-EM and validation of *in vivo* protein depletion and *in vivo* rescue allele expression. (A)** Subcomplex and subunit domain organization of the *S. cerevisiae* KMN complex proteins. CC, CH, HB, N-IDR, and RING-WD40-DEAD box helicase (RWD) domains are highlighted. **(B)** Coomassie brilliant blue–stained SDS-PAGE gels of the purified KMN subcomplexes. **(C–E)** Coomassie brilliant blue–stained SDS-PAGE gels of isolated (C) N and (D) K^HB-RWD^M subcomplexes, or (E) reconstituted K^HB-RWD^MN complexes after separation using 10–30% glycerol gradients. **(F)** Coomassie brilliant blue–stained SDS-PAGE gels of the reconstituted K^HB-RWD^MN complex crosslinked with a 0.0–0.2% glutaraldehyde gradient over the course of a 10–30% glycerol gradient using the GraFix methodology (Stark, 2010) (Kastner et al., 2008). Crosslinking was repeated twice with identical results. Fractions corresponding to the glycerol gradient volume 2.00–2.75 ml were pooled for cryo-EM sample preparation. Fractions corresponding to the glycerol gradient volume 2.00–2.75 ml were pooled for cryo-EM sample preparation. **(G)** Immunoblots of WCEs of the indicated yeast strains treated for the indicated duration with PBS or 0.5 mM IAA. Membranes were blotted with anti-FLAG primary antibody to detect *NNF1-mAID₃-FLAG₅* gene products (top panel) or anti-α-tubulin primary antibody as a loading control (bottom panel). **(H)** Immunoblots of WCEs of the indicated yeast strains treated for the indicated duration with PBS or 0.5 mM IAA. Membranes were blotted with anti-FLAG primary antibody to detect *MTW1-mAID₃-FLAG₅* gene products (top panel) or anti-α-tubulin primary antibody as a loading control (bottom panel). **(I)** Immunoblots of WCEs of the indicated yeast strains. Membranes were blotted with anti-V5 primary antibodies to detect the gene products expressed from the *NNF1* or *nnf1^ΔC^* variant alleles (top panel), or with anti-α-tubulin primary antibody as a loading control (bottom panel). **(J)** Immunoblots of WCEs of the indicated yeast strains. Membranes were blotted with anti-HA primary antibodies to detect the gene products expressed from the indicated *MTW1* or *mtw1^ΔC^* variant alleles (top panel), or with anti-α-tubulin primary antibody as a loading control (bottom panel). CC, coiled coil; HB, helical bundle; WCEs, whole-cell extracts. Source data are available for this figure: SourceData FS1.

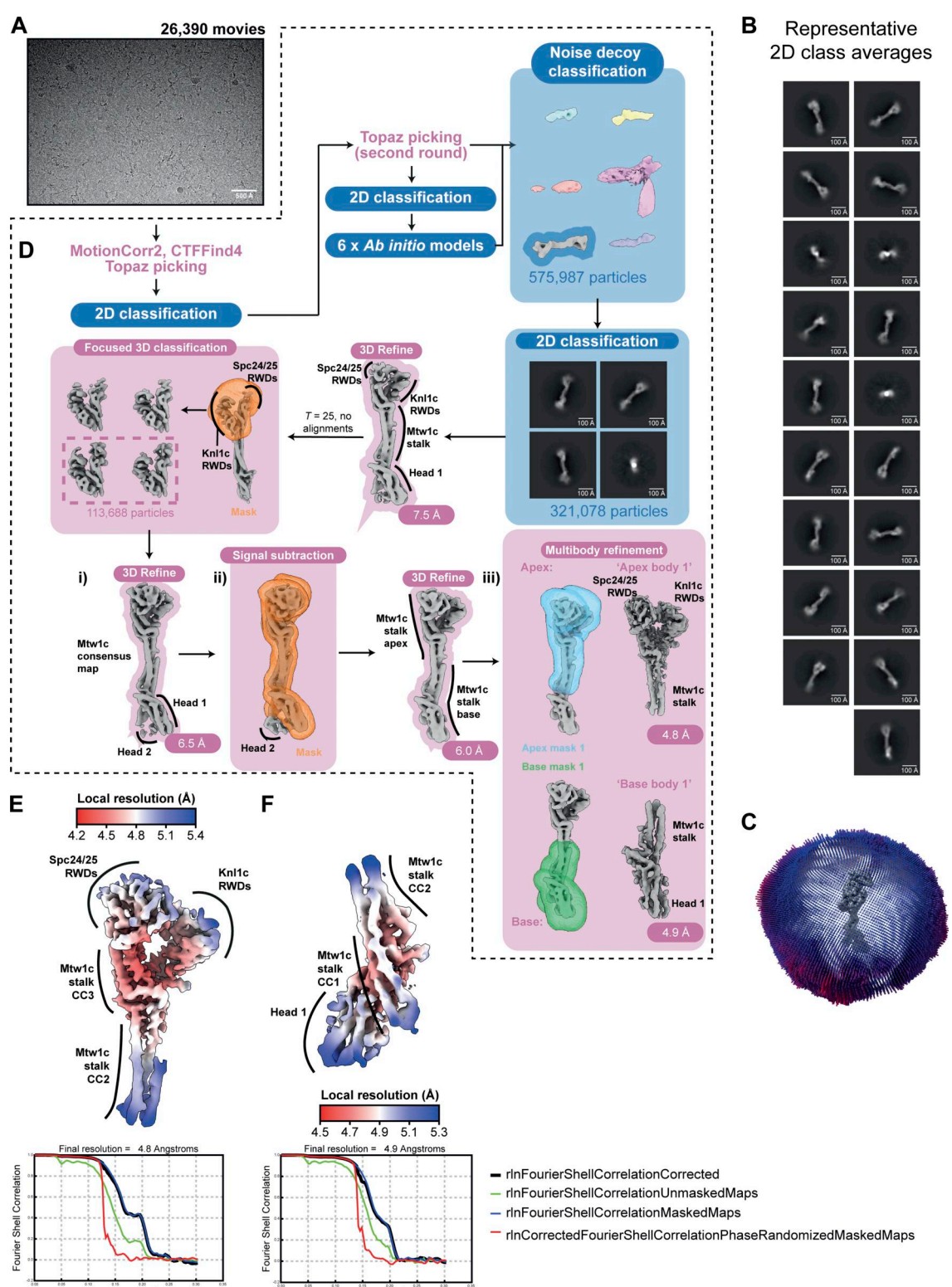

Figure S2.   **Cryo-EM data processing to reconstruct the KMN junction complex. (A)** Representative motion-corrected cryo-EM movie obtained after data acquisition. A total of 26,390 cryo-EM movies were obtained after data acquisition. **(B)** Representative 2D class averages generated using CryoSPARC during the 2D classification step following noise decoy classification. **(C)** Angular distribution of views plot of the particles that contributed to the apex body 1 and base body 1 reconstructions. **(D)** Cryo-EM data processing scheme used to reconstruct higher resolution cryo-EM density maps for the KMN junction complex, as described in the Materials and methods section. Blue and pink labeling and boxes indicates processing steps using CryoSPARC and RELION 5.0, respectively. **(E and F)** Cryo-EM density maps colored according to local resolution of the (E) apex body 1 and (F) base body 1 multibody–derived reconstructions that were used to build molecular models (top). FSC plot of the (E) apex body 1 and (F) base body 1 multibody–derived reconstructions that were used to build the model of the KMN junction complex (bottom). FSC, Fourier shell correlation.

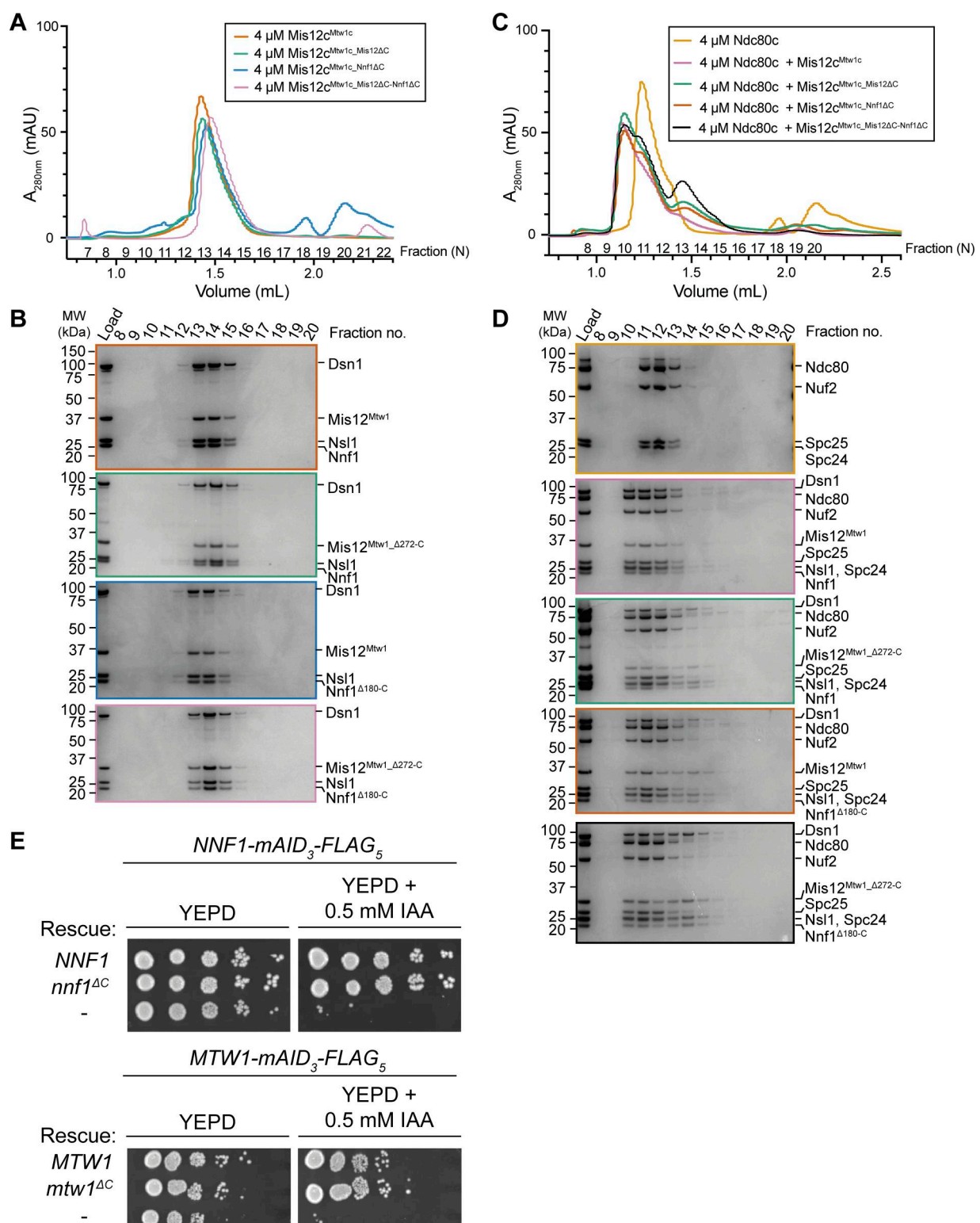

Figure S3. **Biochemical analysis of Mis12**[Mtw1_Cterm-helix] **and Nnf1**[Cterm-helix] **function. (A)** SEC elution chromatograms of Mis12c[Mtw1c] containing either full-length and wild-type proteins, or full-length and wild-type proteins apart from deletion of Mis12[Mtw1] residues 272–289 (Mis12[Mis12_Δ272-C]: Mis12c[Mtw1c_Mis12ΔC]), deletion of Nnf1 residues 180–201 (Nnf1[Δ180-C]: Mis12c[Mtw1c_Nnf1ΔC]), or deletion of both Mis12[Mtw1] residues 272–289 and Nnf1 residues 180–201 (Mis12c[Mtw1c_Mis12ΔC-Nnf1ΔC]). **(B)** Coomassie brilliant blue–stained SDS-PAGE gels of all of the experiments presented in A. **(C)** SEC elution chromatograms of the Mtw1c:Ndc80c interaction reconstitutions. Attempts at reconstituting Mis12c[Mtw1c], Mis12c[Mtw1c_Mis12ΔC], Mis12c[Mtw1c_Nnf1ΔC], and Mis12c[Mtw1c_Mis12ΔC-Nnf1ΔC] with full-length, unmodified Ndc80c are presented alongside the elution profile of Ndc80c alone. **(D)** Coomassie brilliant blue–stained SDS-PAGE gels of the experiments in C. **(E)** Analysis of mtw1[ΔC] and nnf1[ΔC] mutants in yeast strain backgrounds with endogenous MTW1 and NNF1 gene products translationally fused to mAID[3]-FLAG[5] tag. Cell growth was investigated by plating 1:9 serial dilutions onto either YEPD plates or YEPD plates supplemented with 0.5 mM IAA. - = no rescue allele was expressed. mAID = monomeric auxin-inducible degron. Source data are available for this figure: SourceData FS3.

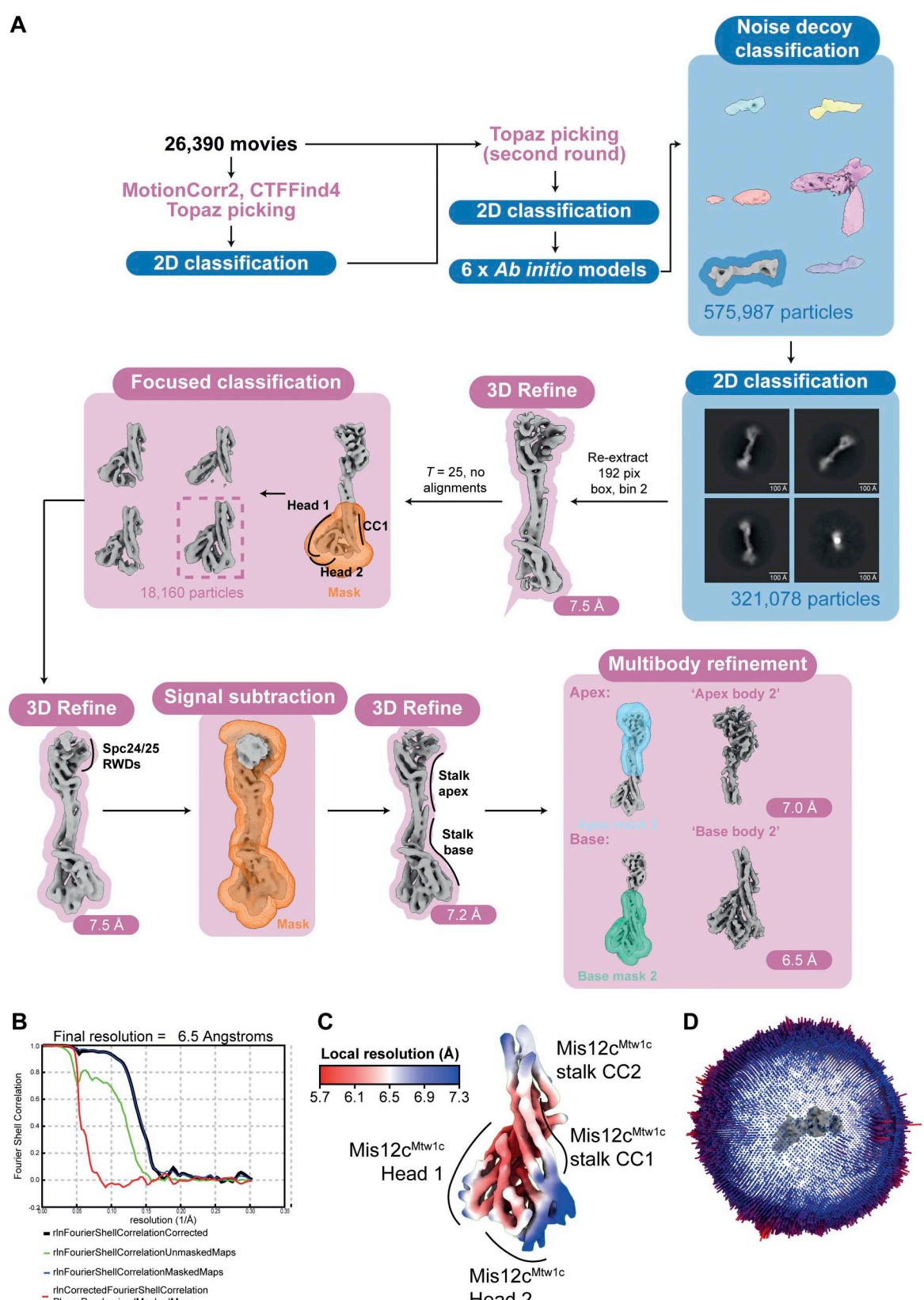

Figure S4. **Cryo-EM data processing to reconstruct Mis12c$^{Mtw1c}$ head 2 domain. (A)** Cryo-EM data processing scheme used to reconstruct cryo-EM density maps containing continuous density for the Mis12c$^{Mtw1c}$ head 2 domain, as described in the Materials and methods section. Blue and pink labeling abd boxes refers to processing steps in CryoSPARC and RELION 5.0, respectively. **(B)** FSC plot for base body 2. **(C)** Cryo-EM density map for base body 2 that was used to model the Mis12c$^{Mtw1c}$ head 2 domain colored according to local resolution. **(D)** Angular distribution of views plot for the particles that contributed to the reconstruction of base body 2. FSC, Fourier shell correlation.

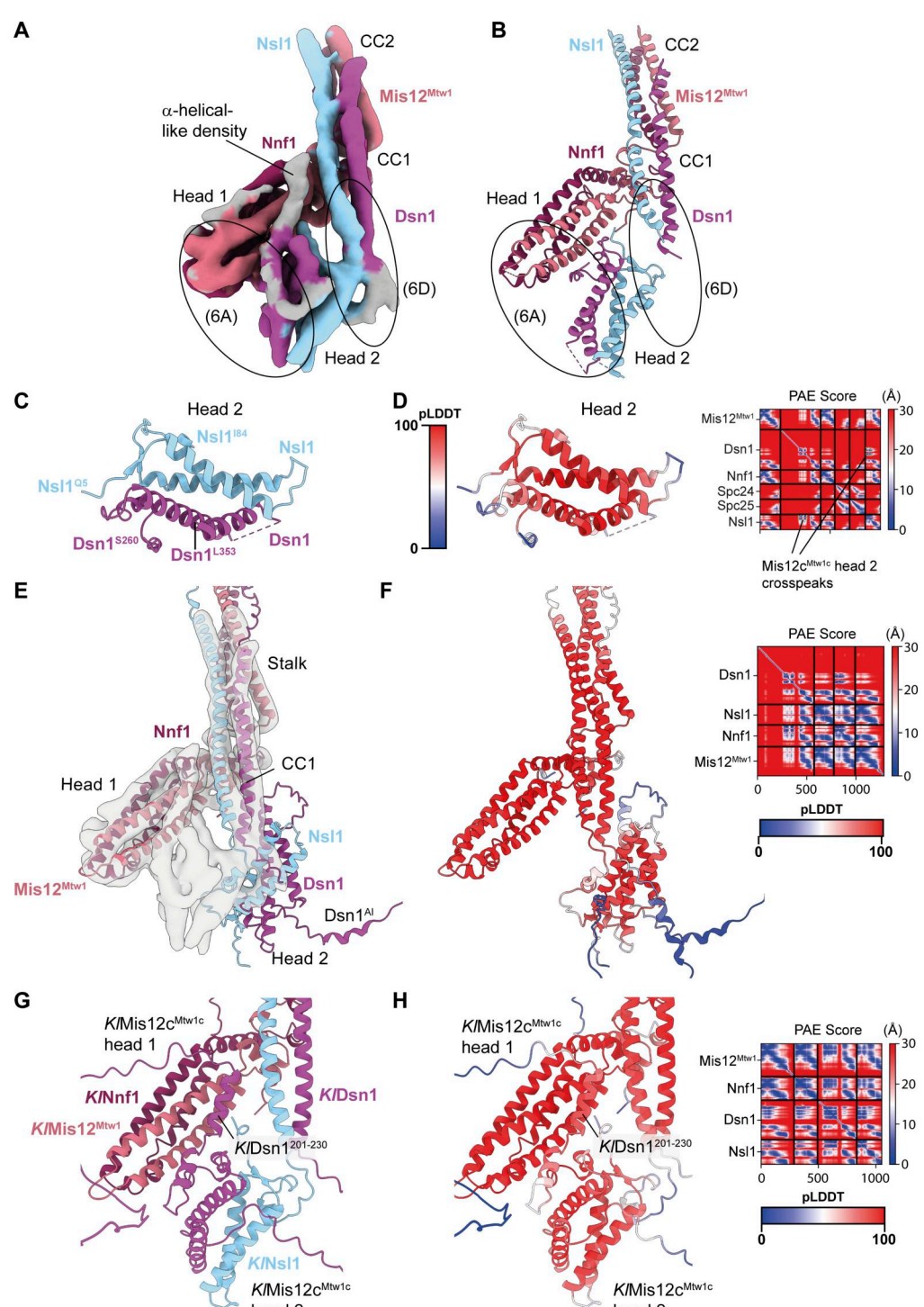

Figure S5.  **Modeling of contacts between Mis12c^Mtw1c head 1, head 2, and Dsn1^N-IDR domains. (A)** Cryo-EM density map with improved occupancy for the Mis12c^Mtw1c head 2 domain (Fig. S4) interpreted using the model of Mis12c^Mtw1c stalk and head 1 domains built in this study and the AF2 model of *S. cerevisiae* head 2 presented in C and D. (6A) and (6D) highlight regions of the map that are shown in close-up in Fig. 6, A and D, respectively. **(B)** Experimental model of *S. cerevisiae* Mis12c^Mtw1c head 1 and head 2 domains interacting at the base of the *S. cerevisiae* Mis12c^Mtw1c stalk domain. **(C)** AF2 structure prediction of the *S. cerevisiae* Mis12c^Mtw1c head 2 domain from the Mis12c^Mtw1c:Spc24:Spc25 prediction colored by chain. Mis12c^Mtw1c head 1 and stalk domains, and Spc24:Spc25 are not shown. **(D)** AF2 model presented in C colored by residue pLDDT score. The PAE score plot is shown on the right of the panel. **(E)** AF2 structure prediction of *S. cerevisiae* Mis12c^Mtw1c (*Sc*Mis12c^Mtw1c) colored by chain docked into the cryo-EM map. The prediction of the head 2 position and orientation does not fit the cryo-EM density. **(F)** AF2 model presented in E colored by residue pLDDT score. The PAE score plot is shown on the right of the panel. **(G)** AF2 structure prediction of the *K. lactis* Mis12c^Mtw1c (*Kl*Mis12c^Mtw1c) colored by chain. The prediction was performed using all full-length and wild-type protein sequences apart from *K. lactis* Dsn1 (*Kl*Dsn1), for which residues 1–149 were excluded to obtain predictions for the interaction between *Kl*Dsn1^201-230 and *Kl*Mis12c^Mtw1c head 1 domain. **(H)** AF2 model presented in G colored by the residue pLDDT score. The PAE score plot is shown on the right of the panel. PAE, predicted alignment error.

Video 1.  **Overview of the *S. cerevisiae* KMN junction complex.** Video shows the molecular details of the *S. cerevisiae* KMN junction complex as shown in Figs. 1, 2, 3, 4, and 5, detailing the overall architecture of the complex (Fig. 1); details of the Mis12c^Mtw1c interface with Knl1c^Spc105c (Fig. 2); and details of the Mis12c^Mtw1c interface with Ndc80c (Fig. 4).

Video 2.  **Predicted conformational change of Dsn1^Al and head 2 accompanying *Sc*Mis12c^Mtw1c activation by Dsn1^Al phosphorylation.** Video shows proposed conformational change of Dsn1^Al and head 2 caused by Aurora B^Ipl1 phosphorylation of Ser240 and Ser250 (Fig. 6, C and E). Phospho-Ser240 would cause electrostatic repulsion with the negatively charged surface of Mis12c^Mtw1c head 1, as well as steric clashes with Nsl1 of head 2. Displacement of Dsn1^Al and head 2 releases the CENP-C^Mif2–binding site on Mis12c^Mtw1c mainly mediated through head 1. The Mis12c^Mtw1c-CENP-C ^Mif2 complex is based on an AF2 model (Fig. 9 A).

**Provided online are Table S1, Table S2, and Table S3. Table S1 shows cryo-EM data collection, refinement, and validation statistics. Table S2 shows genotypes of *S. cerevisiae* strains used in this study. Table S3 shows details of DNA oligonucleotide primers used in this study.**

