## [Peer Review File · The Journal of Cell Biology]

Assembly and phospho-regulatory mechanisms of the budding yeast outer kinetochore KMN complex

Noah Turner, Ziguo Zhang, Jing Yang, Kyle Muir, Stephen McLaughlin, Tomos Morgan, and David Barford

Corresponding Author(s): David Barford, MRC Laboratory of Molecular Biology and Noah Turner, MRC Laboratory of Molecular Biology

Review Timeline:

Submission Date:	2025-06-04
Editorial Decision:	2025-08-05
Revision Received:	2026-01-12
Editorial Decision:	2026-02-12
Revision Received:	2026-02-18

Monitoring Editor: Hironori Funabiki

Scientific Editor: Dan Simon

Transaction Report:

DOI: <https://doi.org/10.1083/jcb.202506015>

August 5, 2025

Re: JCB manuscript #202506015

David Barford
MRC Laboratory of Molecular Biology

Dear Dr. Barford,

Thank you for submitting your manuscript entitled "Assembly and phospho-regulatory mechanisms of the budding yeast outer kinetochore KMN complex." The manuscript has now been assessed by three expert reviewers, whose comments are appended to this letter. As you can see, all three reviewers are supportive to publication of the study, but they made several constructive criticisms that should be addressed before publication. We invite you to submit a revision if you can address the reviewers' key concerns, as outlined here.

Reviewers #1 and #3 commented on the apparent dispensability of Mtw1 and Nnf1 C-terminal alpha-helices (Figure 2F), and suggested further validation, for example, by testing the combinatory effect.

Reviewer #2 suggests to further validate the proposed structural model for the Ame1-Mis12c Head1 interaction interface to substantiate the conclusion from Dimitrova et al (2016).

Reviewers #1 and #3 suggest several ideas to improve readability of the manuscript. I also had difficulty in going through the manuscript since the concept of Head1 and Head2 is not clearly depicted at the beginning of the manuscript. Since it has been shown that phosphorylation of Dsn1 neutralizes its autoinhibition that competes with CENP-C/Mif2 and CENP-U/Ame1, it would be helpful to present a model figure up front.

GENERAL GUIDELINES:

Text limits: Character count for an Article is < 40,000, not including spaces. Count includes title page, abstract, introduction, results, discussion, and acknowledgments. Count does not include materials and methods, figure legends, references, tables, or supplemental legends.

Figures: Articles may have up to 10 main text figures. Figures must be prepared according to the policies outlined in our Instructions to Authors, under Data Presentation, <https://jcb.rupress.org/site/misc/ifora.xhtml>. All figures in accepted manuscripts will be screened prior to publication.

*****IMPORTANT:** It is JCB policy that if requested, original data images must be made available. Failure to provide original images upon request will result in unavoidable delays in publication. Please ensure that you have access to all original microscopy and blot data images before submitting your revision. ***

Supplemental information: There are strict limits on the allowable amount of supplemental data. Articles may have up to 5 supplemental figures. Up to 10 supplemental videos or flash animations are allowed. A summary of all supplemental material should appear at the end of the Materials and methods section.

Please note that JCB now requires authors to submit Source Data used to generate figures containing gels and Western blots with all revised manuscripts. This Source Data consists of fully uncropped and unprocessed images for each gel/blot displayed in the main and supplemental figures. For assays performed using capillary electrophoresis and/or immunoassay-based detection, authors should instead provide the electropherogram graph(s) for each experiment, plotting fluorescence/chemiluminescence intensity vs. molecular weight/size. Please be sure to provide one Source Data file for each figure gels, blots, and/or capillary electrophoresis assays along with your revised manuscript files. File names for Source Data figures should be alphanumeric without any spaces or special characters (i.e., SourceDataF#, where F# refers to the associated main figure number or SourceDataFS# for those associated with Supplementary figures). For traditional gels and blots, the lanes of the gels/blots should be labeled as they are in the associated figure, the place where cropping was applied should be marked (with a box), and molecular weight/size standards should be labeled wherever possible. For capillary electrophoresis assays, each trace in the graph should be color-coded and labeled to indicate which protein, gene, or sample is being measured (please try to avoid red/green combinations to accommodate our color-blind readers).

Source Data files will be made available to reviewers during evaluation of revised manuscripts and, if your paper is eventually

published in JCB, the files will be directly linked to specific figures in the published article.

The typical timeframe for revisions is three to four months. If you anticipate any difficulties in meeting this aforementioned revision time limit, please contact us and we can work with you to find an appropriate time frame for resubmission. Please note that papers are generally considered through only one revision cycle, so any revised manuscript will likely be either accepted or rejected.

Thank you for this interesting contribution to Journal of Cell Biology. You can contact us at the journal office with any questions at cellbio@rockefeller.edu.

Sincerely,

Hironori Funabiki, PhD
Monitoring Editor
Journal of Cell Biology

Dan Simon, PhD
Scientific Editor
Journal of Cell Biology

Reviewer #1 (Comments to the Authors (Required)):

The KMN network is a centerpiece of the kinetochore architecture, combining the microtubule-binding elements of complex with the interface to the centromere-associated network (CCAN). Insights into its structural organization are therefore of great importance to the field. The current study by Turner and colleagues complements last year's work on the corresponding human KMN assemblies (Polley et al and Yatskevich et al). The paper allows an overall comparison between the human and the analogous yeast structures. It helps to rationalize the effect of Aurora/Ipl1 phosphorylation on the N-terminus of Dsn1, a conserved mechanism promoting kinetochore assembly. It also helps to explain the roles of the C-termini of Mtw1 and Nnf1 in organizing the interface with the rest of the KMN network. The biochemistry and the structure analysis is of very high quality. The structural insights into a centerpiece of the kinetochore architecture should be of interest for the field and they rationalize previous findings on a structural level. I am therefore supportive of publication in the JCB, provided the following points can be improved by the authors.

Main points

Figure 2 f: In the current form this in-vivo experiment is not terribly informative and links only poorly to the structural insights, since the mutations do not yield a phenotype. The logical approach would be to combine the two mutations in Mtw1 and Nnf1, or combine a C-terminal deletion with point mutants in the other subunit. Given the reagents the authors already have prepared, this should be possible (e.g. combine the two aid alleles and express two transgenes). Alternatively, the single mutations might have more of an effect in an otherwise sensitized strain background.

Related to this point: line 243.. not sure this is unexpected: The authors describe multiple binding interfaces linking Knl1c to Mis12c, either disruption of either one alone seems to be tolerated.

Figure 4f and 5b: I was confused here, because the comparison between the wildtype Dsn1 peptide binding to the KM construct and the phospho-mimetic 240E 250 E version - directly reflecting the physiological phosphorylation - is split between two figures. Probably better to re-arrange to help the reader.

Figure 5D: It's very difficult to see the small Mif2 fragment in the SEC runs. Highlight it in the Coomassie or maybe blot for it, if possible.

Minor points

line 84: contribution of Knl1 to error correction by binding a Sli15 segment.. also cite Dudziak et al., 2025, EMBO J

line 97: avoid repetition of Mis12c and Mis12... gets confusing for the reader

line 204.. the Zwint RWD can be deleted without compromising viability or causing benomyl-hypersensitivity, see Dudziak et al., 2025, EMBO J. Can this be rationalized from the structure..? Which structural elements would compensate for the lack of the Zwint RWD ?

line 373: not sure that's true.. Dsn1 S240 is annotated as phosphorylated in the SGD based on multiple studies.. please check.

line 461: might be worth mentioning in this regard that the Ame1 connection to Mis12c is essential, while the Mif2 connection is not. (Maybe therefore it is more important to strengthen it?).

line 525 ff: might be better to move this section to the discussion, as the implications for checkpoint signalling are very speculative.

Minor note: I'm not sure it's helpful to refer to Nnf1 as Pmf1 in budding yeast. It's not a name that appears anywhere in the Yeast Genome Database SGD and it hasn't been used in previous publications (e.g. Dimitrova, Mis12c structure) The authors themselves label "Nnf1" in their cellular assays (e.g. 2F). I find it rather confusing during reading.

Reviewer #2 (Comments to the Authors (Required)):

Assembly and phospho-regulatory mechanisms of the budding yeast outer kinetochore KMN complex
Noah N. Turner, Ziguo Zhang, Kyle W. Muir, Stephen H. McLaughlin, Tomos Morgan and David Barford
Turner et al. describe a structural model of the reconstituted S.c. KMN network which is based on cryo-EM density maps and fitted AF2 generated models of distinct domains in low resolution areas of the EM density maps. This KMN structure reveals the differences in complex assembly based on protein sequence divergencies in comparison to the human KMN network. The integrated AF2 models were instrumental for the structural - functional interpretation, in particular, of the head1 and head2 domains and of the mode of the autoinhibition of the Mif2 and Ame1 interactions by the Dsn1 autoinhibitory segment. Overall, the S.c. KMN structure provides an excellent structural framework for clarifying the recruitment of Spc105c and Ndc80c to Mtw1c. In particular, showing the interactions of the different C-terminal helices and CC3 of the Mtw1c to Knl1c and Ndc80c discerns the contributions of the individual motifs to a cooperative binding mode. Due to the lack of density for the coiled-coil domains of Ndc80c and Knl1c their correct positioning and possible involvement in establishing KMN complex formation is only discussed based on AF2 derived models.

The structural model of Mtw1c head1 interacting with the Dsn1 autoinhibitory segment in comparison to the structural models of head1 interacting with the N-termini of either Mif2 or Ame1 provides further insights into the regulation of Mif2 and Ame1 interaction. In particular, indicating that the interaction of head1 with Ame1 N-terminus strongly resembles the head1 - Dsn1-AI complex which is significantly different to its interaction with Mif2 N-terminus extends the current knowledge of how these crucial interactions may be regulated.

Major comment:

As the structural model of head1 - Ame1 N-terminus suggests that Ame1 residues 28-35 are responsible for the high-affinity interaction of Ame1 with Mtw1c head1 which provides insights beyond a previous study by Dimitrova et al. (2016), I recommend to address this finding based on structural models by additional experiments like mutant analysis in in vitro binding assays or structural analysis.

Given the significant technical challenges to obtain a comprehensive structural model of the KMN network and the importance of such a structural framework for understanding kinetochore assembly, SAC signaling, error correction and cell cycle regulation in general I recommend publication of this manuscript in the Journal of Cell Biology upon addressing my major comment.

Minor comments:

Line 131: the original reference of the GraFix method is Kastner et al. 2008 Nature Methods

Line 336: a Glu84 - Ser240 crosslink is a possible but not typical predominant crosslink involving NHS-ester containing crosslinker like sulfo-SDA, thus, I recommend to upload the MS raw data to a public data repository

Line 413: a 1.5-fold increase in binding affinity is extremely subtle and usually makes it very difficult to draw any conclusions. If possible, any further evidence would help to strengthen this observation.

Reviewer #3 (Comments to the Authors (Required)):

The Knl1-Mis12-Ndc80 complex, commonly known as the KMN network, is an essential kinetochore subcomplex that mediates the physical attachment of chromosomes to the spindle microtubules. Although the core intermolecular contacts stabilising the KMN assembly and regulation involve conserved subunits of the KMN network, specific molecular details underpinning the KMN assembly appear to exhibit noticeable variations between humans and budding yeast, in line with the amino acid divergence of individual protein subunits. For example, in humans, the Spc24/25 RWD domains interact with the alpha helices of Dsn1 and Nsl1. The corresponding Nsl1 alpha-helical motif is missing in budding yeast. Instead, an alpha helix from Mis12 cooperatively binds with the Dsn1 alpha helix. As far as the Knl1-Mis12 interaction is concerned, Knl1 RWD domains interact with the extreme C-terminus of Nsl1 in humans, but the identity of the subunit mediating the corresponding Knl1 RWD interaction is not yet known. Likewise, the autoinhibition of Dsn1 and its regulation by Aurora B kinase is also not well understood in budding yeast. Using a combination of cryoEM structural analysis (although only with low-resolution reconstructions and AlphaFold modelling) and biochemistry, the authors fill this knowledge gap by providing interesting structural insights into the budding yeast KMN assembly and regulation.

Points to consider during revision:

The first two subsections of the 'Results' section refer to the human KMN structure very often. Including the human KMN structure in Figure 1 will help readers readily appreciate the similarities and differences. In addition, whenever human and *S. cerevisiae* structures are compared (such as Mis12c interaction with Knl1c and Ndc80c), close-up views of the human and *S. cerevisiae* structures should be shown side by side, or where beneficial structural superpositions are required. As one of the main objectives of the study is to understand the structural similarities and differences between human and budding yeast KMN complexes, the figure panels highlighting the structural comparisons merit inclusion in the main figures themselves.

Showing the relative orientations (how much rotation and around which axes) of the close-up views of the interaction of Mis12c with Knl1c, shown in panels Figure 2C and 2D, with respect to the view shown in Figure 2B, would be useful. Likewise, in Figure 3, relative orientations of Figure 3C and 3D with respect to the view shown in Figure 3B need to be shown.

When describing interactions, often a large number of figure panels are referred to. While this is in principle correct, it doesn't really help the reader. For example:

"First, conserved aliphatic residues in Dsn1 and the C-terminal α -helix of Pmf1Nnf1 (Pmf1Nnf1_Cterm-helix), part of the Mis12cMtw1c CC3 region, engage a hydrophobic patch on the C-terminal RWD domain of Knl1Spc105 (Knl1Spc105_RWD-C) (Figures 1C, D and 2A-C and S3A and Video S1)"

'Conserved aliphatic residues in Dsn1' - including which specific figure panel (which I think is Figure 2C) one has to look at to see this interaction, immediately next to this statement, will help, also explicitly including the residue numbers/names that you are referring to in parentheses. Please make such changes wherever they might be helpful.

Evaluation of the Mis12c-Knl1 interface: deletion of the C-terminal helix of either Mtw1 or Nnf1 reduced the Mis12c interaction with Knl1c, but did not completely abolish the interaction. This aligns with the outcome of the rescue experiments. Does this mean that the structure does not completely capture all key interactions between Mis12c and Knl1? This needs to be addressed explicitly experimentally. If this is not possible for any technical reason, it needs to be discussed.

Regarding the suggestion that Aurora B (Ipl) -mediated phosphorylation of Dsn1 S240 and Dsn1 S250, located within the autoinhibitory segment of Dsn1, enhances CENP-C(mif2) binding by weakening the autoinhibitory interaction, experimentally validating this using biochemical (pull down) and/or biophysical (ITC/SPR) analysis of the interaction between the Aurora B-phosphorylated *Sc* Mis12 complex and the N-terminal fragment of *Sc* CENP-C (mif2) with and without non-phosphorylatable Dsn1 S240A and S253A mutants will strengthen the manuscript.

The use of full-length *Sc* CENP-QU in binding studies with the *Sc* KM complex, being a plausible explanation for the observed higher affinity compared to that observed for the N-terminal fragment of *Kl* CENP-U, can be strengthened by experimental validation with truncation mutants.

Considering that *Sc* KM, containing the Dsn1 autoinhibitory segment, forms a robust complex with *Sc* CENP-QU, the physiological significance of the Dsn1 autoinhibitory segment in influencing CENP-QU binding remains less convincing to this reviewer (although the phosphomimetic KM binds CENP-QU strongly than wild-type KM) without additional cellular studies.

Minor points:

"...and SEC-MALS analysis (data not shown) of KHB-RWDM showed that the purified...." - Please include the data or remove the statement.

"We prepared grids and collected a large cryo-EM dataset on this sample" - Include the number of micrographs in parentheses. Without this information, 'large' does not mean much.

MRC Laboratory
of Molecular
Biology

Dr David Barford FRS, FMedSci

Phone: +44 (0)1223 267075

Fax: +44 (0)1223 268305

Email: dbarford@mrc-lmb.cam.ac.uk

10th January 2026

Dr. Hironori Funabiki, Monitoring Editor

Dr. Dan Simon, Scientific Editor

Journal of Cell Biology.

Dear Hiro and Dan,

Re: Turner et al. #202506015

Thank you for your e.mail of 5th August 2025 with reviewers' comments on our manuscript. We thank the reviewers for carefully reviewing the manuscript, and for their thoughtful comments and suggestions to improve the manuscript. We have revised the manuscript according to their suggestions. We apologise for the delay in submitting our revised manuscript.

Our responses are in blue.

Editorial comments

Reviewers #1 and #3 commented on the apparent dispensability of Mtw1 and Nnf1 C-terminal alpha-helices (Figure 2F), and suggested further validation, for example, by testing the combinatory effect.

Unfortunately we were unable to address the effects of combining deletions of the Mtw1 and Nnf1 C-terminal alpha-helices in yeast cells due to a failure to isolate yeast strains with combined deletions of Mtw1 and Nnf1 C-terminal α -helices in the context of wild type AID-tagged Mtw1 and Nnf1.

Reviewer #2 suggests to further validate the proposed structural model for the Ame1-Mis12c Head1 interaction interface to substantiate the conclusion from Dimitrova et al (2016).

Using isothermal titration calorimetry we determined the K_{DS} of peptides modelled on the N-termini of CENP-C^{Mif2} and CENP-U^{Ame1} for the unphosphorylated and phosphorylated (phospho-mimetic) Knl1c-Mis12c (KM) complex. These data agree well with data from Dimitrova et al (2016) and indicate residues 26-33 of CENP-U^{Ame1} contribute to CENP-U^{Ame1} binding to the KM complex.

Reviewers #1 and #3 suggest several ideas to improve readability of the manuscript. I also had difficulty in going through the manuscript since the concept of Head1 and Head2 is not clearly depicted at the beginning of the manuscript. Since it has been shown that phosphorylation of Dsn1 neutralizes its autoinhibition that competes with CENP-C/Mif2 and CENP-U/Ame1, it would be helpful to present a model figure up front.

We have moved the schematic originally in Fig. 8 to Fig.1F, G. This includes an indication of the Dsn1 auto-inhibitory segment (Dsn1^{AI}) and the binding sites for CENP-C^{Mif2} and CENP-U^{Ame1} that are blocked by Dsn1^{AI}.

The character count is <40,000. Figures have been re-organised according to JCB policies: ten main figures and five Supplementary figures. Source data were uploaded.

Reviewer #1:

Remarks to the Author:

The KMN network is a centerpiece of the kinetochore architecture, combining the microtubule-binding elements of complex with the interface to the centromere-associated network (CCAN). Insights into its

structural organization are therefore of great importance to the field. The current study by Turner and colleagues complements last year's work on the corresponding human KMN assemblies (Polley et al and Yatskevich et al). The paper allows an overall comparison between the human and the analogous yeast structures. It helps to rationalize the effect of Aurora/Ipl1 phosphorylation on the N-terminus of Dsn1, a conserved mechanism promoting kinetochore assembly. It also helps to explain the roles of the C-termini of Mtw1 and Nnf1 in organizing the interface with the rest of the KMN network. The biochemistry and the structure analysis is of very high quality. The structural insights into a centerpiece of the kinetochore architecture should be of interest for the field and they rationalize previous findings on a structural level. I am therefore supportive of publication in the JCB, provided the following points can be improved by the authors.

We thank the reviewer for carefully and critically reading the manuscript, and for suggesting important clarifications and corrections that have improved the manuscript.

Main points

1. Figure 2 f: In the current form this in-vivo experiment is not terribly informative and links only poorly to the structural insights, since the mutations do not yield a phenotype. The logical approach would be to combine the two mutations in Mtw1 and Nnf1, or combine a C-terminal deletion with point mutants in the other subunit. Given the reagents the authors already have prepared, this should be possible (e.g. combine the two aid alleles and express two transgenes). Alternatively, the single mutations might have more of an effect in an otherwise sensitized strain background.

This is an interesting suggestion. We attempted the experiment of combined deletions of the C-terminal α -helices of Mtw1 and Nnf1. Unfortunately, we were unable to isolate yeast strains with combined deletions of Mtw1 and Nnf1 C-terminal α -helices in the context of wild type AID-tagged Mtw1 and Nnf1. This suggested lethality. We are unclear of the reason for this. This result is mentioned on lines 212-214.

2. Related to this point: line 243.. not sure this is unexpected: The authors describe multiple binding interfaces linking Knl1c to Mis12c, either disruption of either one alone seems to be tolerated.

We have altered this sentence and state that the *in vivo* data of single mutations are consistent with the *in vitro* results (lines 209-211).

3. Figure 4f and 5b: I was confused here, because the comparison between the wildtype Dsn1 peptide binding to the KM construct and the phospho-mimetic 240E 250 E version - directly reflecting the physiological phosphorylation - is split between two figures. Probably better to re-arrange to help the reader.

This is a good point. Figure 4f and 5b are now shown in Fig. 6G.

4. Figure 5D: It's very difficult to see the small Mif2 fragment in the SEC runs. Highlight it in the Coomassie or maybe blot for it, if possible.

Now Figure 7C: In the SDS PAGE gels, we have outlined the CENP-C^{Mif2} band using red boxes.

Minor points

1. line 84: contribution of Knl1 to error correction by binding a Sli15 segment.. also cite Dudziak et al., 2025, EMBO J.

Thank you for this. Unfortunately, due to the need to reduce the character count, we truncated some sections of the Introduction including discussion of error correction.

2. line 97: avoid repetition of Mis12c and Mis12... gets confusing for the reader

This sentence was deleted. We agree the terminology is not ideal. Occasionally we do use Mis12c^{Mtw1c} and Mis12^{Mtw12} in the same sentence (eg lines 103-105).

3. line 204.. the Zwint RWD can be deleted without compromising viability or causing benomyl-hypersensitivity, see Dudziak et al., 2025, EMBO J. Can this be rationalized from the structure..? Which structural elements would compensate for the lack of the Zwint RWD ?

This is an interesting point that we had neglected to discuss. The structure shows that the ZWINT^{Kre28_RWD} domain forms only a few contacts with Knl1^{Spc105} (to the α -helix immediately N-terminal of the Knl1^{Spc105_RWD} domains). It does not contribute directly to contacting the Mis12c^{Mtw1c} subunits. Although the ZWINT^{Kre28_RWD} may indirectly stabilise ZWINT^{Kre28_ α 1-helix} through its contact with the bridging Knl1^{Spc105} α -helix, the absence of a direct contact to Mis12c^{Mtw1c} would be consistent with the absence of a phenotype of ZWINT^{Kre28_RWD} deletion. We have discussed this point (lines 180-183) (Fig. 2B).

4. line 373: not sure that's true.. Dsn1 S240 is annotated as phosphorylated in the SGD based on multiple studies.. please check.

Thank you for this suggestion.

The papers cited in the SGD do not show Ser240 phosphorylation. Ser240 is not listed on the Biogrid data base. <https://thebiogrid.org/35002/protein> (YIR010W)

However, a recent paper does include Ser240 as a phosphorylation site (Lanz et al., PMID 33491328) – although not confirmed as an Aurora B site. We have therefore modified the sentence to: ‘ScDsn1^{S240} and its equivalent residue, K/Dsn1^{S213}, conform to the Aurora B^{pl1} substrate consensus sequence. Although not reported to be phosphorylated by Aurora B^{pl1}, ScDsn1^{S240} is phosphorylated *in vivo* (Lanz et al., 2021). Lines 336-338.

5. line 461: might be worth mentioning in this regard that the Ame1 connection to Mis12c is essential, while the Mif2 connection is not. (Maybe therefore it is more important to strengthen it?).

This is a helpful suggestion. We now discuss this point in the Discussion (lines 505-510).

6. line 525 ff: might be better to move this section to the discussion, as the implications for checkpoint signalling are very speculative.

As suggested we have moved this section to the Discussion, starting with ‘Our hypothetical model of the entire *S. cerevisiae* KMN complex including the flexible tethering of other Knl1c and Ndc80c domains to the RWD domains of the KMN junction could have functional consequences for spindle assembly checkpoint (SAC) signalling (Fig. 10).’ Lines 480-483.

7. Minor note: I'm not sure it's helpful to refer to Nnf1 as Pmf1 in budding yeast. It's not a name that appears anywhere in the Yeast Genome Database SGD and it hasn't been used in previous

publications (e.g. Dimitrova, Mis12c structure) The authors themselves label "Nnf1" in their cellular assays (e.g. 2F). I find it rather confusing during reading.

We agree with this useful suggestion.

Reviewer #2 (Comments to the Authors (Required)):

Turner et al. describe a structural model of the reconstituted S.c. KMN network which is based on cryo-EM density maps and fitted AF2 generated models of distinct domains in low resolution areas of the EM density maps. This KMN structure reveals the differences in complex assembly based on protein sequence divergencies in comparison to the human KMN network. The integrated AF2 models were instrumental for the structural - functional interpretation, in particular, of the head1 and head2 domains and of the mode of the autoinhibition of the Mif2 and Ame1 interactions by the Dsn1 autoinhibitory segment.

Overall, the S.c. KMN structure provides an excellent structural framework for clarifying the recruitment of Spc105c and Ndc80c to Mtw1c. In particular, showing the interactions of the different C-terminal helices and CC3 of the Mtw1c to Knl1c and Ndc80c discerns the contributions of the individual motifs to a cooperative binding mode. Due to the lack of density for the coiled-coil domains of Ndc80c and Knl1c their correct positioning and possible involvement in establishing KMN complex formation is only discussed based on AF2 derived models.

The structural model of Mtw1c head1 interacting with the Dsn1 autoinhibitory segment in comparison to the structural models of head1 interacting with the N-termini of either Mif2 or Ame1 provides further insights into the regulation of Mif2 and Ame1 interaction. In particular, indicating that the interaction of head1 with Ame1 N-terminus strongly resembles the head1 - Dsn1-AI complex which is significantly different to its interaction with Mif2 N-terminus extends the current knowledge of how these crucial interactions may be regulated.

We thank the reviewer for carefully and critically reading the manuscript, and for suggesting important clarifications and corrections that have improved the manuscript.

Major comment:

1. As the structural model of head1 - Ame1 N-terminus suggests that Ame1 residues 28-35 are responsible for the high-affinity interaction of Ame1 with Mtw1c head1 which provides insights beyond a previous study by Dimitrova et al. (2016), I recommend to address this finding based on structural models by additional experiments like mutant analysis in in vitro binding assays or structural analysis.

This and the point 4 below are insightful points. We address both here. On performing new surface plasmon resonance (SPR) experiments to address the role of residues 26-33 of CENP-U^{Ame1} in binding unphosphorylated KM (Knl1c-Mis12c^{Mtw1c}) and the KM^{2E} phosphomimetic mutant, we found considerable truncation of Dsn1 comprising the N-terminus and Dsn1^{AI} of sample that had been stored at -80 °C. This resulted in similar K_{DS} of CENP-QU^{Mut} binding to KM and KM^{2E}. We therefore concluded that the similar K_{DS} of CENP-C^{Mif2_1-63} to KM and KM^{2E} (K_D of 0.40 μ M for KM and 0.26 μ M for KM^{2E}) were due to truncation of Dsn1 in the KM sample used in this experiment.

Using ITC and synthetic peptides of the N-termini of CENP-C^{Mif2} (residues 1-38) and CENP-U^{Ame1} (residues 1-33) and freshly prepared KM and KM^{2E} samples (assessed for Dsn1 degradation), we obtained data very consistent with the K_{DS} determined for the association of CENP-C^{Mif2} and CENP-U^{Ame1} with *K. lactis* Mis12c^{Mtw1c} as reported by Dimitrova et al. (2016) using fluorescence polarisation. Specifically, the K_D for a CENP-U^{Ame1_1-33} peptide binding to KM^{2E} was 0.24 μ M. For a CENP-U^{Ame1_1-33-Mut} peptide (with mutations in residues 26-33 to disrupt binding to head 1 of Mis12c^{Mtw1c} involving

residues 26-33), the K_D was 1.31 μM . Dimitrova et al. (2016) reported a K_D of 1.48 μM for CENP-U^{Ame1} residues 1-25. These new data show that residues 26-33 of CENP-U^{Ame1} contribute to binding to Mis12c^{Mtw1c} and explain the higher affinity of CENP-U^{Ame1_1-33} to *S. cerevisiae* Mis12^{Mtw1C} (our data) compared to CENP-U^{Ame1_1-25} to *K. lactis* Mis12^{Mtw1C} reported by Dimitrova et al. (2016). Also similar to Dimitrova et al. (2016), using ITC we found no binding of CENP-U^{Ame1_1-33} to the unphosphorylated KM. These results are presented in lines 391-403 and Fig. 8G.

We cannot account for the large difference between the 0.6 nM affinity of CENP-QU for KM^{2E} measured by SPR compared with the 0.24 μM affinity of the CENP-U^{Ame1_1-33} peptide for KM^{2E} measured by ITC. Using ITC, we determined an approximate K_D of 0.1 μM for CENP-QU binding to KM^{2E}, but the data have a large error value (concentration of 8 μM CENP-QU and 16 μM KM^{2E}). We were unable to measure an affinity of CENP-QU for non-phospho-KM due to limited concentrations of CENP-QU - we could achieve a maximum concentration of 93 μM (CENP-QU) and 16 μM (non-phospho-KM). Therefore, for consistency we removed all SPR data from the manuscript. In the future we will attempt to investigate differences between the SPR and ITC data for KM and KM^{2E} binding to CENP-QU and the CENP-U^{Ame1_1-33} peptide.

Related to point 4, for CENP-C, the peptide CENP-C^{Mif2_1-38} did not bind unphosphorylated KM, but bound to KM^{2E} with a K_D of 0.98 μM . These affinities are very similar to those reported by Dimitrova et al (2016): no binding of a CENP-C^{Mif2} peptide (residues 1-41) to unphosphorylated Mis12c and binding with a K_D of 1.75 μM to Mis12c with Asp mutants of S213/S223. These results are presented in lines 360-368 and Fig. 7D.

We thank the reviewer for this useful suggestion.

Given the significant technical challenges to obtain a comprehensive structural model of the KMN network and the importance of such a structural framework for understanding kinetochore assembly, SAC signaling, error correction and cell cycle regulation in general I recommend publication of this manuscript in the Journal of Cell Biology upon addressing my major comment.

Minor comments:

2. Line 131: the original reference of the GraFix method is Kastner et al. 2008 Nature Methods

Reference added (line 121), thank you.

3. Line 336: a Glu84 - Ser240 crosslink is a possible but not typical predominant crosslink involving NHS-ester containing crosslinker like sulfo-SDA, thus, I recommend to upload the MS raw data to a public data repository.

Thank you for this suggestion. We have deposited the CLMS data in PRIDE with accession number: PXD072662.

4. Line 413: a 1.5-fold increase in binding affinity is extremely subtle and usually makes it very difficult to draw any conclusions. If possible, any further evidence would help to strengthen this observation.

Addressed together with point 1 above.

Reviewer #3 (Comments to the Authors (Required)):

The Knl1-Mis12-Ndc80 complex, commonly known as the KMN network, is an essential kinetochore

subcomplex that mediates the physical attachment of chromosomes to the spindle microtubules. Although the core intermolecular contacts stabilising the KMN assembly and regulation involve conserved subunits of the KMN network, specific molecular details underpinning the KMN assembly appear to exhibit noticeable variations between humans and budding yeast, in line with the amino acid divergence of individual protein subunits. For example, in humans, the Spc24/25 RWD domains interact with the alpha helices of Dsn1 and Nsl1. The corresponding Nsl1 alpha-helical motif is missing in budding yeast. Instead, an alpha helix from Mis12 cooperatively binds with the Dsn1 alpha helix. As far as the Knl1-Mis12 interaction is concerned, Knl1 RWD domains interact with the extreme C-terminus of Nsl1 in humans, but the identity of the subunit mediating the corresponding Knl1 RWD interaction is not yet known. Likewise, the autoinhibition of Dsn1 and its regulation by Aurora B kinase is also not well understood in budding yeast. Using a combination of cryoEM structural analysis (although only with low-resolution reconstructions and AlphaFold modelling) and biochemistry, the authors fill this knowledge gap by providing interesting structural insights into the budding yeast KMN assembly and regulation.

Points to consider during revision:

1. The first two subsections of the 'Results' section refer to the human KMN structure very often. Including the human KMN structure in Figure 1 will help readers readily appreciate the similarities and differences. In addition, whenever human and *S. cerevisiae* structures are compared (such as Mis12c interaction with Knl1c and Ndc80c), close-up views of the human and *S. cerevisiae* structures should be shown side by side, or where beneficial structural superpositions are required. As one of the main objectives of the study is to understand the structural similarities and differences between human and budding yeast KMN complexes, the figure panels highlighting the structural comparisons merit inclusion in the main figures themselves.

We appreciate this very helpful suggestion. We have moved the schematic comparison of human and *S. cerevisiae* KMN from Figure 8 to Figure 1F, G with improved labelling, and added an overview ribbons-style figure of human KMN to Figure 1E. We have added the following comparisons (1) Fig. 1D, E: A comparison of the inactive *S. cerevisiae* and human KMN junction complexes. (2) Fig. 2E, F: A comparison at the Mis12c interface with Knl1:ZWINT. (3) Fig. 4E, F: A comparison at the Mis12c interface with Spc24:Scp25. (4) Fig. 9G, H: A comparison of the mechanism of Dsn1^{AL}-mediated inhibition of Mis12c.

2. Showing the relative orientations (how much rotation and around which axes) of the close-up views of the interaction of Mis12c with Knl1c, shown in panels Figure 2C and 2D, with respect to the view shown in Figure 2B, would be useful. Likewise, in Figure 3, relative orientations of Figure 3C and 3D with respect to the view shown in Figure 3B need to be shown.

This is a useful suggestion and has been added to Fig. 2C, D and Fig. 4C, D (formally Figure 3).

3. When describing interactions, often a large number of figure panels are referred to. While this is in principle correct, it doesn't really help the reader. For example:
 "First, conserved aliphatic residues in Dsn1 and the C-terminal α -helix of Pmf1Nnf1 (Pmf1Nnf1_Cterm-helix), part of the Mis12cMtw1c CC3 region, engage a hydrophobic patch on the C-terminal RWD domain of Knl1Spc105 (Knl1Spc105_RWD-C) (Figures 1C, D and 2A-C and S3A and Video S1)"

We have inserted '(Figure 2C)' immediately after 'First, conserved aliphatic residues in Dsn1 and the C-terminal α -helix of Nnf1 (Nnf1^{Cterm-helix})' (line 167). Hopefully this will guide the reader to the correct figure panel.

4. 'Conserved aliphatic residues in Dsn1' - including which specific figure panel (which I think is Figure 2C) one has to look at to see this interaction, immediately next to this statement, will help, also explicitly including the residue numbers/names that you are referring to in parentheses. Please make such changes wherever they might be helpful.

Inserting '(Figure 2C)' (line 167) will help. We are not sure a long list of residues names and numbers in the text is useful. Such information can be difficult to remember, especially if they are not referred to again.

5. Evaluation of the Mis12c-Knl1 interface: deletion of the C-terminal helix of either Mtw1 or Nnf1 reduced the Mis12c interaction with Knl1c, but did not completely abolish the interaction. This aligns with the outcome of the rescue experiments. Does this mean that the structure does not completely capture all key interactions between Mis12c and Knl1? This needs to be addressed explicitly experimentally. If this is not possible for any technical reason, it needs to be discussed.

This is an interesting question and related to point 1 of referee 1. We attempted the experiment of combined deletions of the C-terminal α -helices of Mtw1 and Nnf1. Unfortunately, we were unable to isolate yeast strains with combined deletions of Mtw1 and Nnf1 C-terminal α -helices, in the context of wild type AID-tagged Mtw1 and Nnf1. This suggested lethality. We are unclear of the reason for this. This result is mentioned on lines 212-214.

6. Regarding the suggestion that Aurora B (Ipl) -mediated phosphorylation of Dsn1 S240 and Dsn1 S250, located within the autoinhibitory segment of Dsn1, enhances CENP-C(mif2) binding by weakening the autoinhibitory interaction, experimentally validating this using biochemical (pull down) and/or biophysical (ITC/SPR) analysis of the interaction between the Aurora B-phosphorylated Sc Mis12 complex and the N-terminal fragment of Sc CENP-C (mif2) with and without non-phosphorylatable Dsn1 S240A and S253A mutants will strengthen the manuscript.

We performed ITC experiments to measure the binding constant for CENP-C^{Mif2} to unphosphorylated KM (Knl1c-Mis12c^{Mtw1c}) and a KM^{2E} phosphomimetic mutant (Glu substitutions of Ser240 and Ser250). Figure 7D (lines 360-368). Analytical size exclusion chromatography is shown in Fig. 7B, C. These data show that CENP-C (either residues 1-38 or 1-63) did not bind unphosphorylated KM. CENP-C (residues 1-63) bound KM^{2E} (as assessed by SEC: Fig. 7B, C) (lines 346-359) and CENP-C (residues 1-38) bound to KM^{2E} with a K_D of 0.98 μ M (Fig. 7Dii) (lines 360-368). Dimitrova (2016) reported a K_D of 1.75 μ M for a CENP-C^{Mif2_1-41} peptide binding to *K. lactis* Mis12c^{Mtw1} (using fluorescence polarisation).

7. The use of full-length Sc CENP-QU in binding studies with the Sc KM complex, being a plausible explanation for the observed higher affinity compared to that observed for the N-terminal fragment of KI CENP-U, can be strengthened by experimental validation with truncation mutants.

This is an interesting point and related to point 1 of referee 2. To assess the role of residues 26-33 of CENP-U^{Ame1} in contributing to the affinity of CENP-QU for KM and KM^{2E}, we substituted alanines for residues of CENP-U^{Ame1} that are predicted from the AF2 model to bind Mis12c^{Mtw1c} (Fig. 8A) and measured the affinity for KM and KM^{2E} using SPR. In performing these experiments, we found that the N-terminus of Dsn1 of the non-phosphorylated KM samples had been truncated (including Dsn1^{AI}). Thus, although the SPR data showed a reduced affinity of the mutant CENP-QU (CENP-QU^{Mut}) for KM^{2E} compared with wildtype CENP-QU, the affinity was the same for both KM and KM^{2E}. We conclude that Dsn1 truncation accounted for the very similar K_D s of CENP-C^{Mif2_1-63} for KM and KM^{2E} measured by SPR (K_D of 0.26 μ M for KM^{2E} and 0.40 μ M for KM).

Using ITC and mutagenesis, and freshly prepared KM and KM^{2E} sample and CENP-U^{Ame1} peptides, we tested the contribution of residues 26-33 of CENP-U^{Ame1} to bind KM^{2E} (not present in the peptide (residues 1-25) used by Dimotrova et al 2016 for determining CENP-U binding to *K. lactis* Mis12c^{Mtw1c}). These data indicated that mutating residues in this region of CENP-U decreased binding affinity 6-fold (0.24 μ M CENP-U^{Ame1_1-33} compared with 1.31 μ M for the mutant CENP-U^{Ame1_1-33-Mut} peptide (Figure 8A, G)). Using fluorescence polarization, Dimotrova et al. (2016) reported a K_D of 1.48 μ M for CENP-U^{Ame1} residues 1-25 binding to *K. lactis* Mis12c^{Mtw1c}. These new data are discussed in the text (lines 391-403).

We cannot account for the large difference between the 0.6 nM affinity of CENP-QU for KM^{2E} measured by SPR compared with the 0.24 μ M affinity of the CENP-U peptide for KM^{2E} measured by ITC. Using ITC, we determined an approximate K_D of 0.1 μ M for CENP-QU binding to KM^{2E}, but the data have a large error value (concentration of 8 μ M CENP-QU and 16 μ M KM^{2E}). We were unable to measure an affinity of CENP-QU for non-phospho-KM due to limited concentrations of CENP-QU - we could achieve a maximum concentration of 93 μ M (CENP-QU) and 16 μ M (non-phospho-KM). Therefore, for consistency we removed all SPR data from the manuscript. In the future we will attempt to investigate differences between the SPR and ITC data for KM and KM^{2E} binding to CENP-QU and the CENP-U^{Ame1_1-33} peptide.

The SEC data do show that CENP-QU bound non-phospho-KM (Fig. 8F), whereas a region of the CENP-C N-terminus (residues 1-63) did not (Fig. 7C).

8. Considering that Sc KM, containing the Dsn1 autoinhibitory segment, forms a robust complex with Sc CENP-QU, the physiological significance of the Dsn1 autoinhibitory segment in influencing CENP-QU binding remains less convincing to this reviewer (although the phosphomimetic KM binds CENP-QU strongly than wild-type KM) without additional cellular studies.

Minor points:

9. "...and SEC-MALS analysis (data not shown) of KHB-RWDM showed that the purified...." - Please include the data or remove the statement.

We have deleted this statement.

10. "We prepared grids and collected a large cryo-EM dataset on this sample" - Include the number of micrographs in parentheses. Without this information, 'large' does not mean much.

Agreed, 'large' was removed.

With best wishes,

David Barford

Lanz, M.C., K. Yugandhar, S. Gupta, E.J. Sanford, V.M. Faca, S. Vega, A.M.N. Joiner, J.C. Fromme, H. Yu, and M.B. Smolka. 2021. In-depth and 3-dimensional exploration of the budding yeast phosphoproteome. *EMBO reports*. 22:e51121.

February 12, 2026

RE: JCB Manuscript #202506015R

David Barford
MRC Laboratory of Molecular Biology

Dear Dr. Barford,

Thank you for submitting your revised manuscript entitled "Assembly and phospho-regulatory mechanisms of the budding yeast outer kinetochore KMN complex." We would be happy to publish your paper in JCB pending final revisions necessary to meet our formatting guidelines (see details below).

A. MANUSCRIPT ORGANIZATION AND FORMATTING:

1) Text limits: Character count for Articles is < 40,000, not including spaces. Count includes title page, abstract, introduction, results, discussion, and acknowledgments. Count does not include materials and methods, figure legends, references, tables, or supplemental legends.

2) Figure formatting: Articles may have up to 10 main text figures. Scale bars must be present on all microscopy images, including inset magnifications. Molecular weight or nucleic acid size markers must be included on all gel electrophoresis. Please add scale bars to figure 2A/B and consider enlarging the PAE plots to improve visibility.

Also, please avoid pairing red and green for images and graphs to ensure legibility for color-blind readers. If red and green are paired for images, please ensure that the particular red and green hues used in micrographs are distinctive with any of the colorblind types. If not, please modify colors accordingly or provide separate images of the individual channels.

3) Statistical analysis: Error bars on graphic representations of numerical data must be clearly described in the figure legend. The number of independent data points (n) represented in a graph must be indicated in the legend. Please indicate whether 'n' refers to technical or biological replicates (i.e. number of analyzed cells, samples or animals, number of independent experiments). If independent experiments with multiple biological replicates have been performed, we recommend using distribution-reproducibility SuperPlots (please see Lord et al., JCB 2020) to better display the distribution of the entire dataset, and report statistics (such as means, error bars, and P values) that address the reproducibility of the findings.

Statistical methods should be explained in full in the materials and methods. For figures presenting pooled data the statistical measure should be defined in the figure legends. Please also be sure to indicate the statistical tests used in each of your experiments (both in the figure legend itself and in a separate methods section) as well as the parameters of the test (for example, if you ran a t-test, please indicate if it was one- or two-sided, etc.). Also, if you used parametric tests, please indicate if the data distribution was tested for normality (and if so, how). If not, you must state something to the effect that "Data distribution was assumed to be normal but this was not formally tested."

4) Materials and methods: Should be comprehensive and not simply reference a previous publication for details on how an experiment was performed. Please provide full descriptions (at least in brief) in the text for readers who may not have access to referenced manuscripts. The text should not refer to methods "...as previously described." Please also indicate the type of membrane used for immunoblotting.

5) For all cell lines, vectors, strains, constructs/cDNAs, etc. - all genetic material: please include database / vendor ID (e.g. Addgene, ATCC, etc.) or if unavailable, please briefly describe their basic genetic features, even if described in other published work or gifted to you by other investigators (and provide references where appropriate). Please be sure to provide the sequences for all of your oligos: primers, si/shRNA, RNAi, gRNAs, etc. in the materials and methods. You must also indicate in the methods the source, species, and catalog numbers/vendor identifiers (where appropriate) for all of your antibodies, including secondary. If antibodies are not commercial, please add a reference citation if possible.

6) Microscope image acquisition: The following information must be provided about the acquisition and processing of images:
a. Make and model of microscope
b. Type, magnification, and numerical aperture of the objective lenses
c. Temperature

- d. Imaging medium
- e. Fluorochromes
- f. Camera make and model
- g. Acquisition software
- h. Any software used for image processing subsequent to data acquisition. Please include details and types of operations involved (e.g., type of deconvolution, 3D reconstitutions, surface or volume rendering, gamma adjustments, etc.).

7) References: There is no limit to the number of references cited in a manuscript. References should be cited parenthetically in the text by author and year of publication. Abbreviate the names of journals according to PubMed.

8) Supplemental materials: Articles may have up to 5 supplemental figures and 10 videos. Please also note that tables, like figures, should be provided as individual, editable files. A summary of all supplemental material should appear at the end of the Materials and methods section. Please include one brief sentence per item.

9) Video legends: Should describe what is being shown, the cell type or tissue being viewed (including relevant cell treatments, concentration and duration, or transfection), the imaging method (e.g., time-lapse epifluorescence microscopy), what each color represents, how often frames were collected, the frames/second display rate, and the number of any figure that has related video stills or images.

10) eTOC summary: A ~40-50 word summary that describes the context and significance of the findings for a general readership should be included on the title page. The statement should be written in the present tense and refer to the work in the third person. It should begin with "First author name(s) et al..." to match our preferred style.

11) Conflict of interest statement: JCB requires inclusion of a statement in the acknowledgements regarding competing financial interests. If no competing financial interests exist, please include the following statement: "The authors declare no competing financial interests." If competing interests are declared, please follow your statement of these competing interests with the following statement: "The authors declare no further competing financial interests."

12) A separate author contribution section is required following the Acknowledgments in all research manuscripts. All authors should be mentioned and designated by their first and middle initials and full surnames. We encourage use of the CRediT nomenclature (<https://casrai.org/credit/>).

13) ORCID IDs: ORCID IDs are unique identifiers allowing researchers to create a record of their various scholarly contributions in a single place. Please note that ORCID IDs are required for all authors. At resubmission of your final files, please be sure to provide your ORCID ID and those of all co-authors.

14) JCB requires authors to submit Source Data used to generate figures containing gels and Western blots with all revised manuscripts. This Source Data consists of fully uncropped and unprocessed images for each gel/blot displayed in the main and supplemental figures. For assays performed using capillary electrophoresis and/or immunoassay-based detection, authors should instead provide the electropherogram graph(s) for each experiment, plotting fluorescence/chemiluminescence intensity vs. molecular weight/size. Since your paper includes cropped gel and/or blot images, please be sure to provide one Source Data file for each figure gels, blots, and/or capillary electrophoresis assays along with your revised manuscript files. File names for Source Data figures should be alphanumeric without any spaces or special characters (i.e., SourceDataF#, where F# refers to the associated main figure number or SourceDataFS# for those associated with Supplementary figures). For traditional gels and blots, the lanes of the gels/blots should be labeled as they are in the associated figure, the place where cropping was applied should be marked (with a box), and molecular weight/size standards should be labeled wherever possible. For capillary electrophoresis assays, each trace in the graph should be color-coded and labeled to indicate which protein, gene, or sample is being measured (please try to avoid red/green combinations to accommodate our color-blind readers).

Source Data files will be directly linked to specific figures in the published article. Source Data Figures should be provided as individual PDF files (one file per figure). Authors should endeavor to retain a minimum resolution of 300 dpi or pixels per inch. Please review our instructions for export from Photoshop, Illustrator, and PowerPoint here: <https://rupress.org/jcb/pages/submission-guidelines#revised>.

15) Journal of Cell Biology now requires a data availability statement for all research article submissions. These statements will be published in the article directly above the Acknowledgments. The statement should address all data underlying the research presented in the manuscript. Please visit the JCB instructions for authors for guidelines and examples of statements at (<https://rupress.org/jcb/pages/editorial-policies#data-availability-statement>).

B. FINAL FILES:

Thank you for your attention to these final processing requirements. Please revise and format the manuscript and upload materials within 7 days. If you need an extension for whatever reason, please let us know and we can work with you to determine a suitable revision period.

Thank you for this interesting contribution, we look forward to publishing your paper in Journal of Cell Biology.

Sincerely,

Hironori Funabiki, PhD
Monitoring Editor
Journal of Cell Biology

Dan Simon, PhD
Scientific Editor
Journal of Cell Biology

Reviewer #1 (Comments to the Authors (Required)):

Turner and colleagues present a revised version of their study on the organization of the budding yeast KMN network. Using a combination of cryoEM and structure predictions, the authors compare yeast and human KMN studies and provide new insights into the regulation of Mif2 and Ame1 binding to Mtw1c, via relieving competition with an autoinhibitory segment in the Dsn1 N-terminus.

The authors have addressed my comments regarding the previous version. They have improved presentation and readability and added biochemical experiments to strengthen insights into Ipl1-regulated binding to Mtw1c (Figures 7 and 8). They have also attempted additional genetic analysis of the structural predictions, even if these experiments did not yield tangible results, I applaud them for making the effort. Overall, I am supportive of publication.

Remaining points for final production:

Figure1: Consider making panel A a bit larger for better readability.

Figure 1F: I think the authors mean - CENP-U, not CENP-Q binding site (lower left part of the graph)

Reviewer #2 (Comments to the Authors (Required)):

My comments and suggestions on the first submission have been sufficiently addressed, thus I recommend publication of the manuscript in Journal of Cell Biology.

Reviewer #3 (Comments to the Authors (Required)):

In this reviewer's opinion, the authors have addressed the reviewer's comments well (although for various reasons, not all concerns were addressed experimentally, and some open questions remain). I am happy to support the publication.

MRC Laboratory
of Molecular
Biology

Dr David Barford FRS, FMedSci

Phone: +44 (0)1223 267075

Fax: +44 (0)1223 268305

Email: dbarford@mrc-lmb.cam.ac.uk

17th February 2026

Dr. Hironori Funabiki, Monitoring Editor

Dr. Dan Simon, Scientific Editor

Journal of Cell Biology.

Dear Hiro and Dan,

Re: Turner et al. #202506015

Thank you for your e.mail of 12th February 2026 with a decision to accept our manuscript. We are delighted with this news. We thank you for editing this manuscript, and the reviewers' for their comments on the revised manuscript. We have revised the manuscript according to JCB instructions and made changes as detailed below.

Our responses are in blue.

Editorial comments

2) Figure formatting: Articles may have up to 10 main text figures. Scale bars must be present on all microscopy images, including inset magnifications. Molecular weight or nucleic acid size markers must be included on all gel electrophoresis. Please add scale bars to figure 2A/B and consider enlarging the PAE plots to improve visibility.

Scale bars added to Supplementary Fig. S2A,B and Supplementary Fig. S4A.

PAE plots are enlarged.

Reviewer #1

Remaining points for final production:

Figure1: Consider making panel A a bit larger for better readability.

Panel Fig. 1A is enlarged.

Figure 1F: I think the authors mean - CENP-U, not CENP-Q binding site (lower left part of the graph)

Correction made. Thank you for spotting that.

With best wishes,

David Barford